# Evaluating and Explaining Prompt Sensitivity of LLMs Using Interactions

**Ruiyang Qin** [1]  **Qingzhuo Wang** [1]  **Tian Wang** [1]  **Zhihua Wei** [1]  **Wen Shen** [1] [†]

## Abstract

The remarkable capabilities of large language models (LLMs) are often undermined by their instability. Even subtle and semantically irrelevant changes in prompts can cause dramatic fluctuations in performance, a phenomenon known as prompt sensitivity. Previous studies typically evaluate prompt sensitivity by comparing the LLM's final outputs when prompts change. However, such coarse-grained metrics fail to explain the internal reasons for prompt sensitivity. In this paper, we introduce interactions as a fine-grained tool to analyze prompt sensitivity of LLMs. Specifically, we decompose the output score of the LLM into a set of interactions. Each interaction represents a nonlinear relationship involving a set of input variables. We discover that subtle changes to prompts can trigger severe instability in interactions, even when the outputs of the LLM remain the same. To this end, we propose an Interaction-based Prompt Sensitivity (IPS) metric by quantifying changes in interactions when we introduce subtle changes to prompts. We apply the IPS metric to 50 open-source LLMs and uncover four factors that reduce the prompt sensitivity of LLMs, including supervised fine-tuning, increased model scales, dense architectures, and few-shot learning. More crucially, we discover a common mechanism by which these four factors reduce prompt sensitivity: all four factors tend to reduce the prompt sensitivity of low-order interactions (*i.e.*, interactions involving few input variables).

## 1. Introduction

LLMs have demonstrated exceptional proficiency in numerous natural language processing tasks (Srivastava et al., 2023; Zhou et al., 2023; Annepaka & Pakray, 2024; Chang

---

[1]School of Computer Science and Technology, Tongji University, Shanghai, China. [†]Correspondence to: Wen Shen <wenshen@tongji.edu.cn>.

*Proceedings of the 43rd International Conference on Machine Learning*, Seoul, South Korea. PMLR 306, 2026. Copyright 2026 by the author(s).

et al., 2024), a success largely driven by the effectiveness of prompting. However, this power is undermined by prompt sensitivity. That is, semantically unimportant changes to the prompt can result in divergent outputs (Sclar et al., 2024). Current research (Sclar et al., 2024; Chatterjee et al., 2024; Lu et al., 2024; Alzahrani et al., 2024; Errica et al., 2025; Razavi et al., 2025) on evaluating prompt sensitivity only focuses on the LLM's final output. These output-based metrics typically measure changes in performance, such as task accuracy or output consistency. As coarse-grained measures, these metrics only reveal the consequences of prompt sensitivity (*e.g.*, the LLM's prediction changes from one answer to another one) but fail to explain its underlying reasons.

In this paper, we aim to evaluate the prompt sensitivity of LLMs from a fine-grained perspective and investigate the underlying reasons why certain factors can decrease the prompt sensitivity of LLMs. Recent research (Chen et al., 2024; Zhou et al., 2024; Ren et al., 2025; 2024b) has utilized interactions to explain the fine-grained inference logic of deep neural networks (DNNs). Inspired by these studies, we introduce the interactions framework to fine-grainedly analyze the prompt sensitivity of LLMs.

Specifically, given a sentence $x$ with $n$ input variables (*e.g.*, words or tokens) indexed by $N = \{1, 2, \ldots, n\}$, an interaction represents an intricate nonlinear relationship associated with a specific combination of input variables. Consider the sentence $x = $ *"He is a green hand."* In this context, the idiom "green hand" carries the meaning of "beginner". The joint presence of the input variables in the set $S = \{green, hand\} \subseteq N$ triggers a special interaction effect. This interaction effect, denoted as $I_S$, pushes the network's inference towards the semantic meaning of *"beginner."* Li & Zhang (2023) have mathematically demonstrated that the scalar output $v(x)$ of a DNN is always equivalent to the output of an interaction-based logical model $\phi(x) = \sum_{S \subseteq N} I_S$. That is, $v(x) = \phi(x) = \sum_{S \subseteq N} I_S$. Thus, the inference logic of a DNN can be explained by a set of interactions.

***Coarse-grained analysis vs. fine-grained analysis.*** Traditional analysis of prompt sensitivity can only coarsely reflect whether the LLM's prediction alters when the prompt changes. Beyond this, we introduce an interaction-based logical model $\phi(x)$ as a fine-grained analytical tool to analyze the stable and unstable interactions encoded by the LLM for

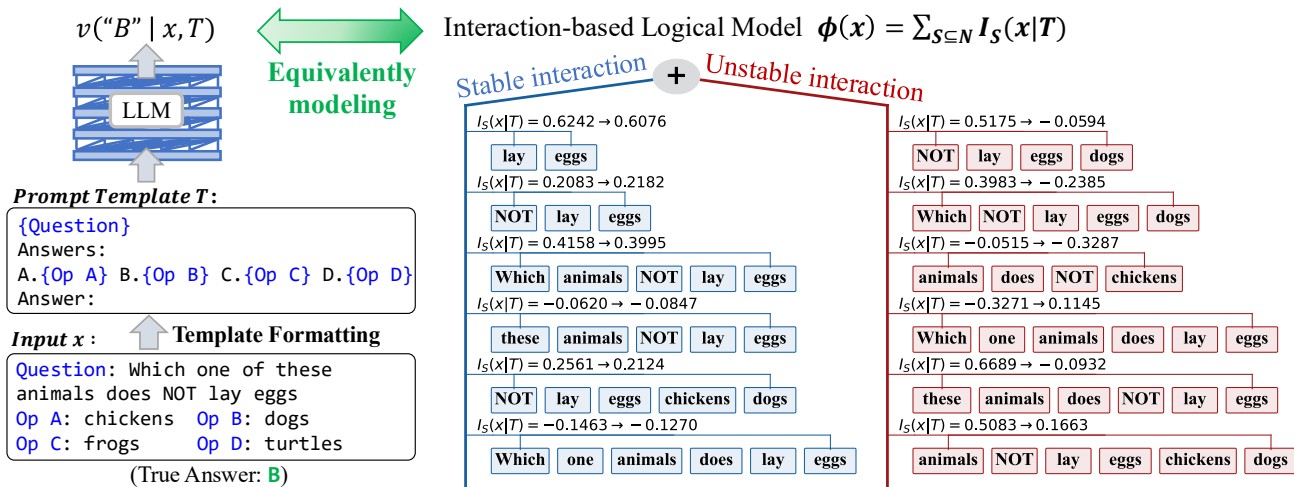

*Figure 1.* Using interactions for fine-grained analysis of prompt sensitivity. Given an input $x$ and a prompt template $T$, the LLM's output score $v("B"|x,T)$ is equivalent to the output of an interaction-based logical model $\phi(x)$, i.e., $v("B"|x,T) = \phi(x) = \sum_{S \subseteq N} I_S$. In this way, we can uncover the underlying reasons for the prompt sensitivity of LLMs by analyzing detailed interaction patterns. Specifically, we divide interactions into those that are stable to prompt changes (*i.e.* stable interaction) and those that are not (*i.e.* unstable interaction).

each specific sample. As shown in Figure 1, when we keep the same input $x$ but introduce semantically equivalent alterations to the prompt template $T$ (*e.g.*, change "Answers" to "ANSWERS"), traditional coarse-grained analysis can only reveal that the LLM's prediction changes from the correct answer "B" to the incorrect answer "A" but fails to explain why or how this change occurs. In contrast, our interaction-based analysis precisely identifies unstable interactions that may contribute to the prompt sensitivity of the LLM. An interaction is considered stable if its effect changes minimally (e.g., $I_{\{lay, eggs\}}$ shifts from 0.6242 to 0.6076). Conversely, an interaction is deemed unstable if its effect fluctuates dramatically (e.g., $I_{\{NOT, lay, eggs, dogs\}}$ shifts from 0.5175 to -0.0594). Strikingly, we find that unstable interactions exist even when the LLM's final output remains the same. This indicates that our fine-grained analysis reveals potential instability that is entirely invisible to traditional, output-level metrics.

Building on these findings, we leverage the interaction framework to propose a fine-grained metric to evaluate the prompt sensitivity of LLMs, which is termed **I**nteraction-based **P**rompt **S**ensitivity (**IPS**). This metric quantifies changes in interaction patterns when LLMs process different prompts. We apply the proposed IPS metric to evaluate the prompt sensitivity of 50 open-source LLMs. However, drawing conclusions about which LLM families are more or less sensitive from this ranking is challenging, as an LLM's prompt sensitivity stems from multiple, intertwined factors.

To this end, we conduct a series of comparative experiments to disentangle these factors, discovering four factors that reduce the prompt sensitivity of LLMs: **(1)** Supervised fine-tuning reduces the prompt sensitivity. Instruct/chat models (with supervised fine-tuning) exhibit lower prompt sensi-

tivity than base models. **(2)** LLMs with larger parameter numbers exhibit lower prompt sensitivity. **(3)** Dense models are generally less sensitive than mixture-of-experts (MoE) models. **(4)** Few-shot learning considerably reduces prompt sensitivity compared to 0-shot learning.

More crucially, we explore and uncover a common underlying mechanism that explains how the four aforementioned factors reduce prompt sensitivity: they primarily reduce the instability of low-order interactions (i.e., interactions involving a small number of input variables). This finding is counterintuitive, as our experiments demonstrate that high-order interactions (i.e., interactions involving many input variables) tend to exhibit the highest sensitivity, while low-order interactions are inherently less sensitive. Unexpectedly, these factors further stabilize the already stable low-order interactions, yet remain ineffective in addressing the more pronounced sensitivity of high-order interactions.

## 2. Related Work

**Prompt sensitivity of LLMs.** Previous studies (Sun et al., 2024; Zhu et al., 2024a; Sclar et al., 2024; Alzahrani et al., 2024) demonstrated that LLMs are highly sensitive to minor perturbations or semantically unimportant alterations to prompts, which can lead to significant performance variation. Such prompt sensitivity presents a considerable risk to the reliability of LLMs. Existing metrics (Sclar et al., 2024; Chatterjee et al., 2024; Lu et al., 2024; Zhuo et al., 2024; Cao et al., 2024; Errica et al., 2025; Razavi et al., 2025) for evaluating the prompt sensitivity of LLMs typically measure shifts in final outputs, such as task accuracy or output consistency. However, these metrics are coarse-grained and fail to probe the LLM's internal logic. In this paper, we propose

a fine-grained metric that evaluates prompt sensitivity based on interactions. This framework enables us to uncover the underlying mechanisms of prompt sensitivity in LLMs.

**Using game-theoretic interactions to explain DNNs.** Traditional methods for explanations (Tenney et al., 2020; Zhang et al., 2020) often lack mathematical guarantees of faithfulness, meaning their outputs may not accurately represent the internal logic employed by the DNN. To this end, Ren et al. (2023a) proposed to use interactions between input variables to explain DNNs and provided a series of theoretical guarantees for the method's validity. Furthermore, it has been empirically discovered (Li & Zhang, 2023) and theoretically proven (Ren et al., 2024a) that a DNN typically encodes only a sparse set of interactions. At the application level, the interaction framework has proven effective in a wide range of complex tasks, including adversarial transferability (Deng et al., 2024), model generalization (Chen et al., 2024; Zhou et al., 2024), model training process (Ren et al., 2025; 2024b), overfitting (Ren et al., 2023b) and other tasks (Shen et al., 2024; Li et al., 2025; Wen et al., 2026). In this paper, we use the interaction framework to analyze the prompt sensitivity of LLMs.

# 3. Interaction-Based Analysis of the Prompt Sensitivity of LLMs

## 3.1. Preliminaries: Interactions

This subsection introduces the definition of interactions, as well as the mathematical guarantees of interaction-based explanation. Given a DNN $v$ and an input sentence $\boldsymbol{x}$ with $n$ input variables (*e.g.*, words) indexed by $N = \{1, 2, \ldots, n\}$, let $v(\boldsymbol{x}) \in \mathbb{R}$ denote the scalar output of the DNN. Here we set $v(\boldsymbol{x}) = \log \frac{p(y=y^*|\boldsymbol{x})}{1-p(y=y^*|\boldsymbol{x})} \in \mathbb{R}$, where $p(y = y^*|\boldsymbol{x})$ represents the probability of generating the ground truth token $y^*$ given the input $\boldsymbol{x}$. We define a surrogate logical model $\phi(\boldsymbol{x})$ to match the scalar output $v(\boldsymbol{x})$ of the DNN. Recent studies (Li & Zhang, 2023; Ren et al., 2023a) have proven Theorem 3.1, which shows that the output score $v(\boldsymbol{x})$ on any randomly masked[1] input $\boldsymbol{x}_T$ can be accurately calculated by the following surrogate logical model $\phi(\cdot)$.

$$\phi(\boldsymbol{x}_T) \triangleq \phi(\boldsymbol{x}_\emptyset) + \sum_{S \subseteq N} \mathbb{1}(S \mid \boldsymbol{x}_T) \cdot I_S, \qquad (1)$$

where the AND trigger function $\mathbb{1}(S \mid \boldsymbol{x}_T) \in \{0, 1\}$ represents an **AND relationship** between input variables in $S$, which can also be termed **AND interaction pattern**. The scalar weight $I_S$ quantifies the effect of an AND relationship, which can also be termed **interaction effect**. An AND relationship is activated only by the joint presence of all input variables in the set $S$, *i.e.*, all input variables in

---

[1]It is common to use a specific token or embedding to mask input variables of a DNN, *e.g.*, replacing the target token with a specific [MASK] token. Please see Appendix A for details.

$S$ are not masked. For instance, given the input sentence $\boldsymbol{x}$ = *"He is a green hand,"* the co-occurrence of the input variables in the set $S = \{green, hand\}$ contributes a numerical effect $I_S$ that pushes the surrogate logical model's inference towards the semantic meaning of *"beginner."* If an AND interaction $S$ is triggered, *i.e.*, $\mathbb{1}(S \mid \boldsymbol{x}_T) = 1$, the corresponding interaction effect $I_S$ is added to the output of the logical model. Otherwise, if any word in $S$ is masked and the AND interaction is not triggered, *i.e.*, $\mathbb{1}(S \mid \boldsymbol{x}_T) = 0$, its corresponding interaction effect $I_S$ is not added to the output of the logical model. $\boldsymbol{x}_\emptyset$ represents that all input variables in $N$ are masked.

**Theorem 3.1** (Universal matching property, proven in Appendix B). *Given an input $\boldsymbol{x}$ with $n$ input variables, we randomly mask any combinations of input variables to generate $2^n$ masked inputs $\{\boldsymbol{x}_T \mid T \subseteq N\}$. For every masked input $\boldsymbol{x}_T$, when the scalar weight $I_S$ in the logical model $\phi(\cdot)$ are set to $I_S = \sum_{S' \subseteq S}(-1)^{|S|-|S'|} \cdot v(\boldsymbol{x}_{S'})$, $\phi(\boldsymbol{x}_\emptyset) = v(\boldsymbol{x}_\emptyset)$, the output of the logical model $\phi(\cdot)$ can always match the DNN's output score $v(\cdot)$.*

$$\forall T \subseteq N, \quad v(\boldsymbol{x}_T) = \phi(\boldsymbol{x}_T) \qquad (2)$$

In addition, **the sparsity property also provides a theoretical guarantee for the faithfulness of interaction-based explanation**. The sparsity property shows that a DNN only encodes a sparse set of interactions with salient effects. That is, only a small subset of all $2^n$ interactions in Theorem 3.1, termed salient interactions, have a significant impact on the logical model's output. In contrast, the majority of interactions have negligible effects and are considered noise patterns. The sparsity property of AND interactions has been proven by Ren et al. (2024a). We follow Li & Zhang (2023) to extract AND-OR interactions[2] from input variables. Such a technique has proven effective in pursuing higher sparsity of interactions, supported by both theoretical proofs (Li & Zhang, 2023) and extensive empirical validation (Ren et al., 2023a; Zhou et al., 2024).

## 3.2. Verifying the Faithfulness of Considering Interactions as Inference Patterns Used by LLMs

Before utilizing the interaction framework to analyze the prompt sensitivity of LLMs, we need to **theoretically prove and experimentally validate** the faithfulness of using interactions to explain LLMs. **In theory**, Theorem 3.1 guarantees that the surrogate logical model's output $\phi(\cdot)$ can always match the LLM's output $v(\cdot)$ for all $2^n$ masked samples. Since the logical model's output $\phi(\cdot)$ is composed entirely of the sum of all interaction effects as defined in Eq. (1), **we can consider interactions as the detailed inference patterns that constitute the LLM's internal logic**.

---

[2]The OR interaction is proved to be a specific AND interaction. Please see Appendix C for proof and Appendix D for how to extract OR interactions.

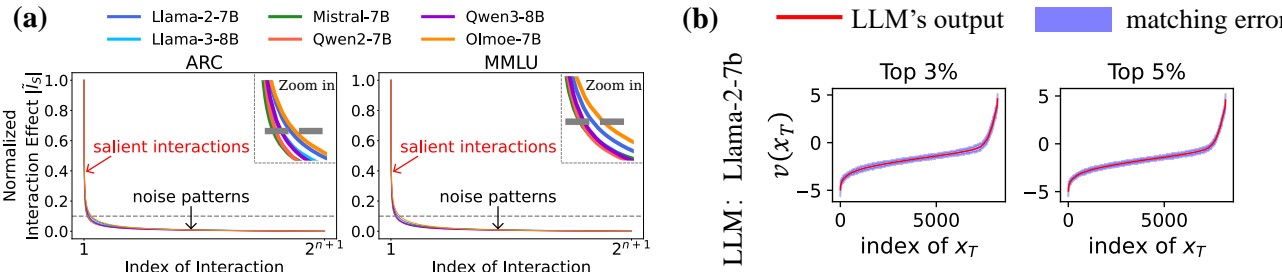

*Figure 2.* (a) Verifying the sparsity of interactions. We show absolute values of normalized interactions in a descending order. LLMs all encode a small number of salient interactions, while most of the interaction effects are negligible. (b) Verifying the quality of universal matching for any $2^n$ masked inputs. The red line plots outputs of the LLM in an ascending order.

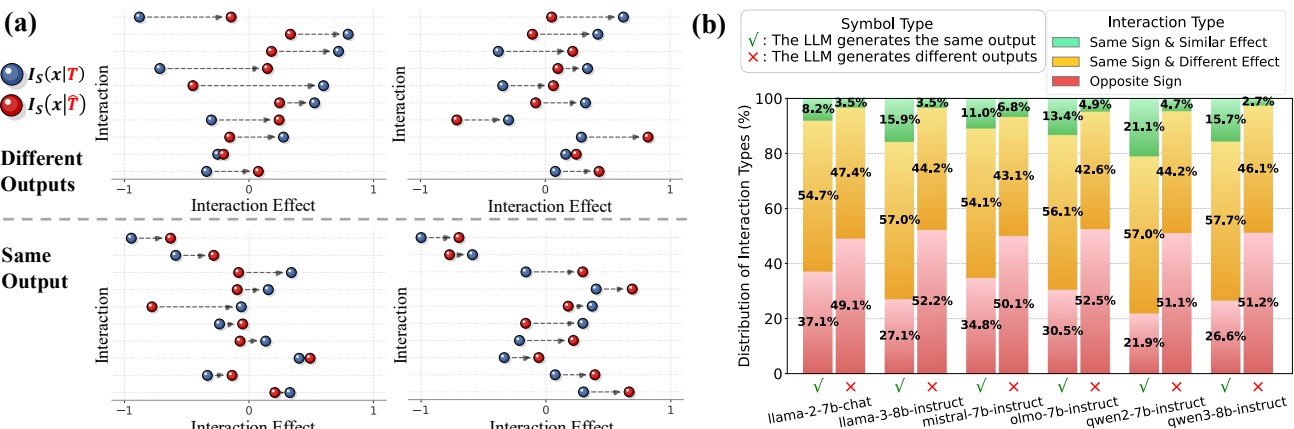

*Figure 3.* (a) Case studies of interaction-level analysis. The plots compare the interaction effects encoded by the LLM for the same input $x$ under two semantically equivalent prompt templates, $T$ (blue) and $\hat{T}$ (red). The bottom row shows examples where the LLM's output remains the same; the top row shows examples where it changes. Results show that interaction effects are highly unstable, even when the output of the LLM remains the same. (b) The distribution of salient interactions types of various LLMs.

Therefore, we can evaluate the prompt sensitivity of LLMs by measuring the instability of interaction patterns.

**In practice**, we conduct experiments to verify whether LLMs encode sparse interactions. Consider the multiple choice question (MCQ) in Figure 1 as an example. Let $Q$ denote the set of all words in the question, *e.g.*, $Q = \{$*Which, one, of, these, animals, does, NOT, lay, eggs*$\}$, $M$ denote the set of all words in the options, *e.g.*, $M = \{$*chickens, dogs, frogs, turtles*$\}$, and $T$ denote the set of all words in the prompt template, *e.g.*, $T = \{$*Answers:, A., B., C., D., Answer:*$\}$. Specifically, we use words[3] in $Q \cup M$ as input variables and compute the interaction effect $I_S$ of all interactions $S \subseteq Q \cup M$. Meanwhile, we treat words in prompt template $T$ as background context. We follow Li & Zhang (2023) to extract AND and OR interactions. Thus, given an input $x = Q \cup M$ with $n$ words, we can obtain $2^{n+1}$ interactions, including $2^n$ AND interactions and $2^n$ OR interactions. For each interaction ef-

fect $I_S$, we apply min-max normalization to it. Specifically, $\widetilde{I}_S \triangleq sgn(I_S) \cdot \frac{|I_S| - Min}{Max - Min}$, where *Min* and *Max* are the minimum and maximum absolute values of all $2^{n+1}$ interaction effects; $sgn(I_S) = \frac{I_S}{|I_S|}$ represents the sign of $I_S$. Figure 2 (a) shows the distribution of $|\widetilde{I}_S|$. Results[4] verify that only a small set of interactions have salient effects, while most of the interactions have negligible effects and can be considered as noise patterns. Figure 2 (b) compares the LLM's true output $v(x_T)$ for all $2^n$ masked inputs against the logical model using only the most salient interactions. Even when using the top 3% or top 5% of all interactions, the matching error is minimal. This empirically demonstrates that the LLM's output can be faithfully approximated by a small set of salient interactions.

### 3.3. Using Interactions as a Fine-Grained Tool to Analyze the Prompt Sensitivity of LLMs

Based on the above verification, we deploy interactions as a fine-grained analytical tool. Specifically, for an LLM, we

---

[3]We use words instead of tokens as input variables because different LLMs may divide the same word into different tokens. For example, Llama-2-7B tokenizes the word "Elements" into two tokens "Element" and "s", while Qwen3-8B treats it as one token.

[4]Results on more LLMs in Appendix F support the conclusion.

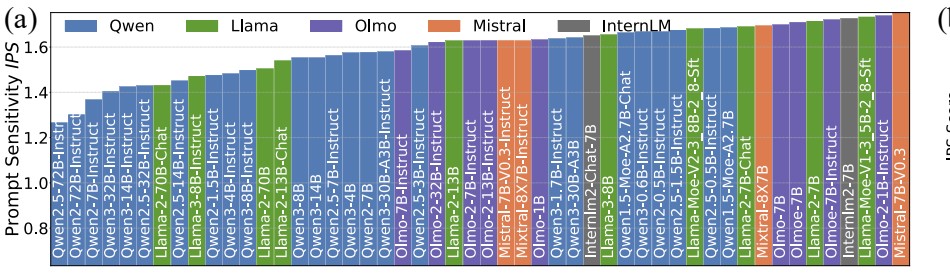
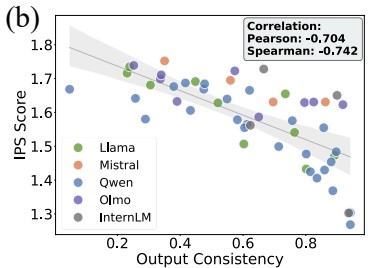

*Figure 4.* Prompt sensitivity of all the 50 open-source LLMs in an ascending order.

visualize the changes in salient interactions for the same input under a pair of prompt templates. As Figure 3 (a) shows, when the LLM's outputs are different, nearly half of the salient interactions reverse their sign (from positive to negative, or vice versa), another 30%-40% change significantly in magnitude, and only a small fraction (10%-20%) remain stable. Strikingly, even when the LLM's output remains the same, the majority (60%-80%) of the salient interactions are still unstable. This offers preliminary evidence that semantically irrelevant alterations to the prompt template can lead to significant changes in the salient interaction patterns.

To systematically analyze this instability beyond individual cases, we classify interactions into three distinct types: 1) **Opposite sign**: The interaction effect reverses its sign (e.g., from positive to negative). The sign change represents the most severe form of instability, as it can completely reverse the LLM's internal logic. 2) **Same sign & different effect**: The interaction maintains its positive or negative influence, but its magnitude changes substantially (i.e., its effect is more than doubled or less than halved). 3) **Same sign & similar effect**: The interaction's sign and magnitude both remain stable, which represents robust, stable interactions. Figure 3 (b) plots the distribution of three interaction types over all samples, conditioned on whether the LLM's final output changes or remains the same across a pair of prompts. When the LLM generates different outputs, the interaction patterns are highly unstable. *Opposite sign* interactions account for approximately 50%, meaning nearly half of salient interactions reverse the sign of their effect. The truly stable *Same sign & similar effect* interactions account for a mere 2.7%-6.8%, while the remaining 42.6%-47.4% of interactions, though maintaining their sign, change significantly in magnitude. More alarmingly, even when the final output of the LLM remains the same, the instability of interactions still exists. While the situation improves, the majority of interactions still fall into the two unstable categories..

The results demonstrate that output-level analysis is insufficient to capture the unreliable internal patterns of LLMs. Conversely, our interaction-based analysis offers a fine-grained lens to uncover latent instability of LLMs, offering a new analytical tool for quantifying the ratio of stable and unstable interactions on a per-sample basis for any LLM.

## 4. Evaluating and Analyzing the Prompt Sensitivity of LLMs

### 4.1. Evaluating Interaction-Based Prompt Sensitivity

We propose an interaction-based metric to measure the prompt sensitivity of LLMs. Given an MCQ dataset $\mathcal{D}$, for any input $\boldsymbol{x} \in \mathcal{D}$, it is composed of the question $Q$ and the options $M$, *i.e.*, $\boldsymbol{x} = Q \cup M$. Given a prompt template $T$, we make minor changes to $T$ and obtain a modified prompt template $\hat{T}$. By applying the pair of prompt templates $T$ and $\hat{T}$ to the same input $Q \cup M$, we can construct two similar prompts. The LLM is supposed to extract similar salient interactions from $Q \cup M$ when processing these two prompts because $T$ and $\hat{T}$ have the same semantic meaning. Thus, let $\Omega_{\text{salient}}(\boldsymbol{x}|T) = \{S \in \Omega(\boldsymbol{x}) \mid |\widetilde{I}_S(\boldsymbol{x}|T)| > \tau\}$ represent the set of **salient interactions** extracted from $\boldsymbol{x}$ given the prompt template $T$. Here $\tau$ is a threshold used to distinguish salient interactions from noise patterns. In the main paper, we set $\tau$ as 0.1 for all experiments. Please see Appendix F.8 for hyperparameter experiments of $\tau$. Similarly, let $\Omega_{\text{salient}}(\boldsymbol{x}|\hat{T}) = \{S \in \Omega(\boldsymbol{x}) \mid |\widetilde{I}_S(\boldsymbol{x}|\hat{T})| > \tau\}$ represent the set of **salient interactions** extracted from $\boldsymbol{x}$ given the prompt template $\hat{T}$. Therefore, on the dataset $\mathcal{D}$, we define the LLM's **Interaction-based Prompt Sensitivity** as *IPS*.

$$IPS \triangleq \mathbb{E}_{\boldsymbol{x}} \left[ \mathbb{E}_{T,\hat{T}} \left[ \frac{1}{|\Omega_{\text{union}}|} \sum_{S \in \Omega_{\text{union}}} \frac{|\widetilde{\mathcal{I}}_S(\boldsymbol{x}|T) - \widetilde{\mathcal{I}}_S(\boldsymbol{x}|\hat{T})|}{\frac{|\widetilde{I}_S(\boldsymbol{x}|T)| + |\widetilde{I}_S(\boldsymbol{x}|\hat{T})|}{2}} \right] \right],$$
(3)

where $\Omega_{\text{union}} = \Omega_{\text{salient}}(\boldsymbol{x}|T) \cup \Omega_{\text{salient}}(\boldsymbol{x}|\hat{T})$ is a unified set by taking the union of the two salient sets $\Omega_{\text{salient}}(\boldsymbol{x}|T)$ and $\Omega_{\text{salient}}(\boldsymbol{x}|\hat{T})$; the outer expectation, $\mathbb{E}_{\boldsymbol{x}}$, represents an averaging over all inputs $\boldsymbol{x} \in \mathcal{D}$, and the inner expectation, $\mathbb{E}_{T,\hat{T}}$, represents an averaging over all pairs of prompt templates $(T, \hat{T})$. This metric evaluates prompt sensitivity of LLMs by calculating the symmetric mean absolute percentage error of salient interactions over all samples in the dataset $\mathcal{D}$.

*Models and Datasets.* We conduct experiments on **50 open-source LLMs from 6 model families**. This diverse set includes 10 LLMs from the Llama family: Llama-2, Llama-3, Llama-MoE (Touvron et al., 2023; Grattafiori et al., 2024; Zhu et al., 2024b; Qu et al., 2024); 4 LLMs from the Mistral family: Mistral, Mixtral (Jiang et al., 2023; Jiang

et al., 2024); 25 LLMs from the Qwen family: Qwen2, Qwen2.5, Qwen3, Qwen1.5-MoE, Qwen3-30B-A3B (Yang et al., 2024; 2025); 9 LLMs from the OLMo family: OLMo, OLMo-2, OLMoE (Groeneveld et al., 2024; OLMo et al., 2025; Muennighoff et al., 2024); 2 LLMs from the InternLM family: InternLM2 (Cai et al., 2024). We evaluate all LLMs on **two widely used MCQ benchmarks**: ARC (Clark et al., 2018) and MMLU (Hendrycks et al., 2021). Experiments on **open-ended tasks** are shown in Section 4.3. For each input, we apply five distinct prompt templates to it. These templates maintain the same core content and differ only in minor formatting details, such as letter case and separators. For masking words in the input sentences, we follow the approach of Cheng et al. (2025) and utilize a certain [MASK] token for each LLM. A comprehensive list of all LLMs, along with the specific prompt templates, mask tokens used in experiments is provided in Appendix E.

We apply the IPS metric to evaluate the prompt sensitivity of 50 open-source LLMs. As shown in Figure 4 (a), we observe a wide variance in IPS, ranging from 1.268 (Qwen2.5-72B-Instruct) to 1.752 (Mistral-7B-v0.3). However, no LLM series or families achieve a complete victory. The distribution indicates that an LLM's prompt sensitivity is influenced by multiple underlying factors. Identifying these factors is of great significance for the robustness research of LLMs.

To validate the reliability of the IPS metric, we investigate its correlation with the traditional metrics output consistency, which is defined as the proportion of samples where the LLM generates identical predictions across different prompt templates. Figure 4 (b) shows that the IPS score exhibits a negative correlation with output consistency. This result confirms that **IPS aligns well with the output consistency while offering a more fine-grained perspective to quantify prompt sensitivity beyond output matching**.

**Robustness to the threshold $\tau$.** In the main paper, we set the threshold $\tau$ as 0.1 for all experiments. To verify that our findings are robust to the selection of $\tau$, we compute interactions across 16 distinct threshold values, ranging from 0.05 to 0.20 with a step size of 0.01. We evaluate the consistency of IPS rankings across these thresholds, observing an average Spearman's rank correlation of **0.9905** and an average Pearson correlation of **0.9957** (Appendix F.8.1). These near-perfect correlations indicate that the *relative* ranking among LLMs remains highly stable. More crucially, we conduct all the experiment with $\tau = 0.05$ and $\tau = 0.15$. As detailed in Appendix F.8.2, the main conclusions remain same as $\tau = 0.1$, proving that our findings are robust to different $\tau$.

### 4.2. Analyzing the Factors Impacting Prompt Sensitivity

In this section, we investigate four factors that might influence the prompt sensitivity of LLMs, including (1) supervised fine-tuning, (2) model scales, (3) model architectures,

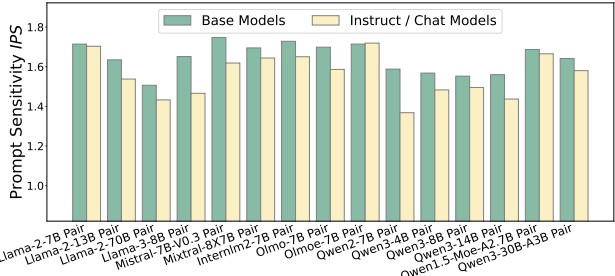

*Figure 5.* A comparison of the prompt sensitivity between instruct/chat models and base models. Results show that instruct/chat models are less sensitive than corresponding base models.

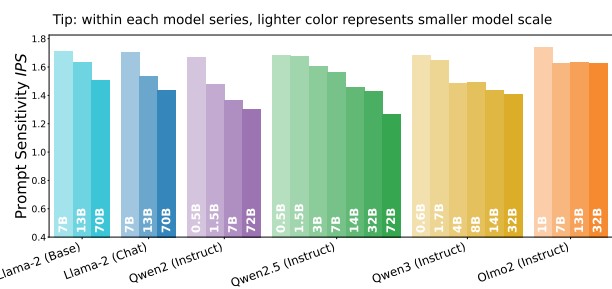

*Figure 6.* A comparison of prompt sensitivity across different model scales. As the model scale increases, the prompt sensitivity within a model series systematically decreases.

and (4) prompting methods.

**Factor 1: instruct/chat models vs. base models.** Supervised fine-tuning (*e.g.*, instruction tuning) is now a common practice to align base models with human preferences for certain tasks, yielding models often referred to as instruct or chat models (for different tasks or purposes). Therefore, we investigate the impact of supervised fine-tuning by comparing the prompt sensitivity of instruct/chat models with base models. Results[5] on the ARC dataset in Figure 5 show that almost all instruct/chat models exhibit lower prompt sensitivity than their corresponding base models. This demonstrates that supervised fine-tuning enables the LLM to encode more stable interactions. A valid explanation is that base models are pre-trained on unstructured and raw texts, but instruct/chat models are further fine-tuned on instruction-response datasets or dialogue datasets. Thus, instruct/chat models can precisely understand the function of prompt templates and focus on the task-relevant inputs.

**Factor 2: model scales.** We investigate the relationship between the model scale (*i.e.*, the number of parameters) and the prompt sensitivity. Results[5] on the ARC dataset in Figure 6 show that within the same model series, as the model scale increases, the overall prompt sensitivity systematically decreases. This demonstrates that larger LLMs

---

[5]Results on the MMLU dataset in Appendix F exhibit the same conclusion.

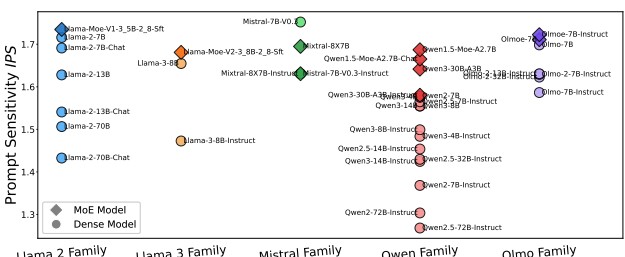

*Figure 7.* A Comparison of prompt sensitivity between MoE models and dense models. Generally, MoE models tend to be more sensitive than dense models in the same model family.

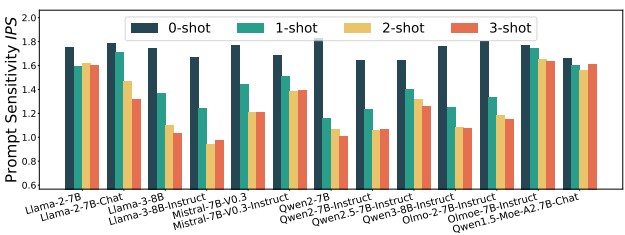

*Figure 8.* A comparison of prompt sensitivity between 0-shot learning and few-shot learning. The drop in prompt sensitivity is substantial from 0-shot to 1-shot.

encode more stable interactions, making them less susceptible to superficial changes in the prompt template.

**Factor 3: dense models vs. MoE models.** MoE models scale up model capacity with minimal computational cost by dynamically activating different subsets of *"expert"* sub-networks (Cai et al., 2025). In contrast, all parameters in dense models participate in every computation. We aim to investigate the impact of model architectures by comparing the prompt sensitivity of dense models with MoE models. Results[5] on the ARC dataset in Figure 7 show that in model families including Llama-2, Llama-3, Qwen and Olmo, all MoE models exhibit higher prompt sensitivity than dense models. This suggests that MoE models encode more unstable interactions. A potential confounding factor is the number of active parameters. Given that smaller models are more sensitive (Factor 2), one might attribute MoE instability to lower active parameters. To mitigate the influence of active parameters, we conduct controlled variable analysis. Results (Table 7, Appendix I) confirms that *MoE models remain more sensitive than dense models even with similar active parameters.* We attribute this to the dynamic routing mechanism. For different prompt templates, the gating network may route the input to different experts so that it is processed by different sub-networks, leading to different interactions encoded and thus higher sensitivity. In conclusion, while MoE models achieve impressive performance with reduced computational overhead, this benefit comes at the cost of weaker stability.

**Factor 4: few-shot learning vs. 0-shot learning.** Few-

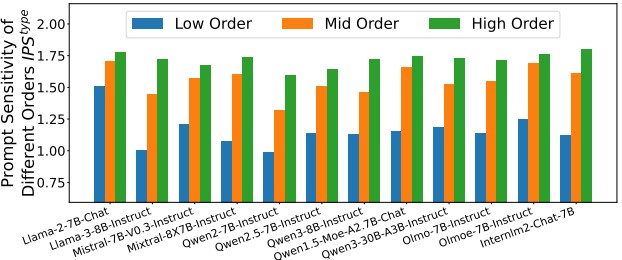

*Figure 9.* Prompt sensitivity of LLMs for different order types.

shot learning is utilized to improve LLMs' performance by providing in-context examples to better specify the task (Wang et al., 2020). We investigate the impact of prompting methods on prompt sensitivity by comparing 0-shot learning with few-shot learning[6]. Results[5] on the ARC dataset in Figure 8 show that incorporating in-context examples leads to a significant reduction in prompt sensitivity across all tested LLMs. For most of the LLMs, the most substantial drop occurs when moving from 0-shot to 1-shot, while adding more in-context examples yields slower reductions. This suggests that even a single example is sufficient to establish the LLM's understanding of the task, leading it to ignore superficial template variations and focus on the core input.

### 4.3. Explore the Underlying Mechanisms of Improved Stability for All Factors

In this section, we aim to explore **whether there exists a common reason to explain the underlying mechanisms by which the four aforementioned factors reduce the prompt sensitivity of LLMs**. Specifically, we analyze the prompt sensitivity of different types of interactions, so as to reveal the source of the LLM's prompt sensitivity. To this end, we analyze the sensitivity of interactions with different complexities, which are defined as the orders of interactions. The order of an interaction $S$ is defined as the number of input variables involved, *i.e.*, $|S|$. An interaction with high order indicates an intricate relationship including many input variables, while an interaction with low order represents a simple relationship including few input variables. We further define three types of prompt sensitivity metrics corresponding to different types of interaction orders. Specifically, we partition interactions in $\Omega_{\text{union}}$ in Eq. (3) into three distinct groups based on their orders: low-order, mid-order, and high-order. Given an input $x$ with $n$ words, $\Omega_{\text{union}}^{low} \triangleq \{S \in \Omega_{\text{union}} \mid 1 \leq |S| \leq \lfloor \frac{1}{3}n \rfloor\}$, $\Omega_{\text{union}}^{mid} \triangleq \{S \in \Omega_{\text{union}} \mid \lfloor \frac{1}{3}n \rfloor < |S| \leq \lfloor \frac{2}{3}n \rfloor\}$, $\Omega_{\text{union}}^{high} \triangleq \{S \in \Omega_{\text{union}} \mid \lfloor \frac{2}{3}n \rfloor < |S| \leq n\}$. Then we calculate prompt sensitivity for low-order, mid-order, and high-order interactions, as $IPS^{low}$, $IPS^{mid}$, $IPS^{high}$.

Figure 9 presents the prompt sensitivity of different order

---

[6]See Appendix E for the prompt templates and settings of few-shot learning.

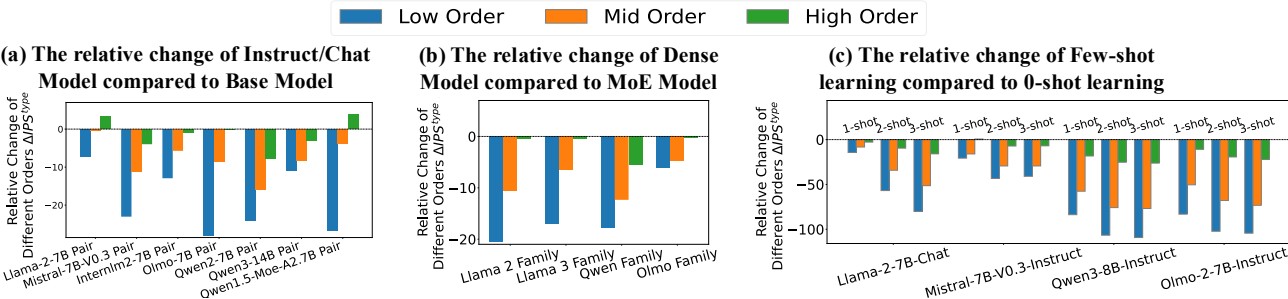

*Figure 10.* Comparing the relative change in the prompt sensitivity of low-, mid-, and high-order interactions for different factors.

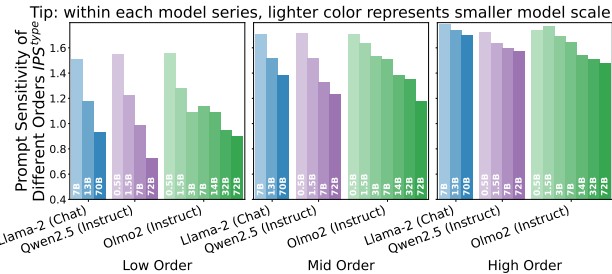

*Figure 11.* A comparison of prompt sensitivity of different order types across different model scales.

types on the ARC dataset. Results[4] show that the prompt sensitivity of low-order interactions is the lowest, followed by mid-order, while high-order interactions exhibit the highest prompt sensitivity. This indicates that low-order interactions encoded by LLMs are relatively stable when faced with subtle changes to prompt templates, *i.e.*, simple interaction patterns are more robust. Conversely, the high sensitivity of high-order interactions reveals that the LLMs' internal representation of complex patterns is highly unstable.

**Explaining why the four factors can reduce prompt sensitivity.** Inspired by the results in Figure 9, we now investigate how the four aforementioned factors influence the prompt sensitivity of low-, mid-, and high-order interactions. This order-level analysis aims to reveal the common mechanism by which these factors reduce the LLM's prompt sensitivity. For each factor, we quantify its effect on low-, mid-, and high-order interactions by computing the relative change in IPS between the LLM with the factor and its counterpart without it. Specifically, given *type* ∈ {*low, mid, high*}, the relative change is defined as $\Delta IPS^{type} = (IPS_A^{type} - IPS_B^{type})/IPS_B^{type}$. For **Factor 1** (fine-tuned vs. base), A and B are fine-tuned and base models, respectively. For **Factor 3** (dense vs. MoE), A and B are dense and MoE models, with the final $\Delta IPS^{type}$ being the average over all pairs of a specific dense model and a specific MoE model within a model family. For **Factor 4** (few-shot vs. 0-shot), A and B are x-shot ($x \in \{1, 2, 3\}$) and 0-shot learning. The relative change metric is unsuitable for **Factor 2** (model scales), as scale is a

continuous variable. Instead, we directly analyze the trend of IPS values as model parameters increase.

Results[4] shown in Figure 10 and Figure 11 converge on a common explanation for how the four factors reduce prompt sensitivity. **The most significant reduction in prompt sensitivity is consistently observed in low-order interactions.** An obvious, though less pronounced, decrease is also seen at the mid-order level. In contrast, the change of the sensitivity of high-order interactions is relatively minimal, remaining at a high level. It indicates that the stability of low-order interactions is critical to the overall robustness of LLMs.

This phenomenon is unexpected. Although results in Figure 9 show that low-order interactions are naturally more robust than other types of interactions, the four factors above still significantly reduce the sensitivity of low-order interactions. Instead, they fail to reduce the sensitivity of high-order interactions, which are inherently the most sensitive. This phenomenon indicates that stable low-order interactions are much easier for LLMs to learn, while it is difficult for LLMs to make high-order interactions more stable.

**Robustness on open-ended tasks.** To verify the robustness of our findings, we conduct additional experiments on open-ended generation tasks. We utilize the Dolly-15k dataset (Conover et al., 2023), which contains a diverse range of non-MCQ tasks, including open Q&A, classification, and others. The results of this analysis, detailed in Appendix K, consistently verify our main conclusions drawn from the MCQ experiments. This strongly suggests that the four factors and the underlying mechanisms of prompt sensitivity we have uncovered can be generalized to open-ended questions.

**Robustness to more complex prompt perturbations.** In this experimental setup, we use the Dolly-15k dataset and introduce more complex prompt perturbations, specifically semantic paraphrases and instruction reordering (detailed in Appendix L.1), to test the robustness of our conclusions. The results in Figures 12, 13, and 14 consistently affirm our conclusions drawn from the template-based experiments. This indicates that our conclusions are robust to more com-

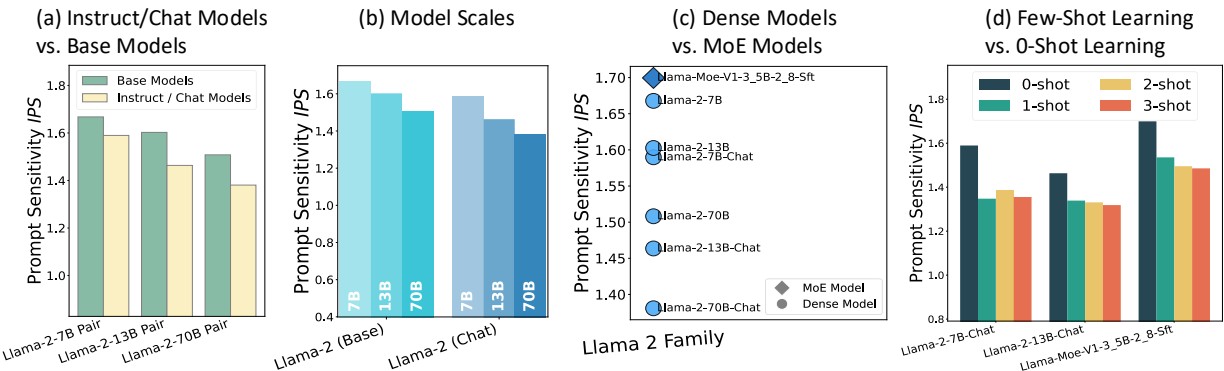

*Figure 12.* (a) A comparison of the prompt sensitivity between instruct/chat models and base models. (b) A comparison of prompt sensitivity across different model scales. (c) A comparison of prompt sensitivity between MoE models and dense models. (d) A comparison of prompt sensitivity between 0-shot learning and few-shot learning.

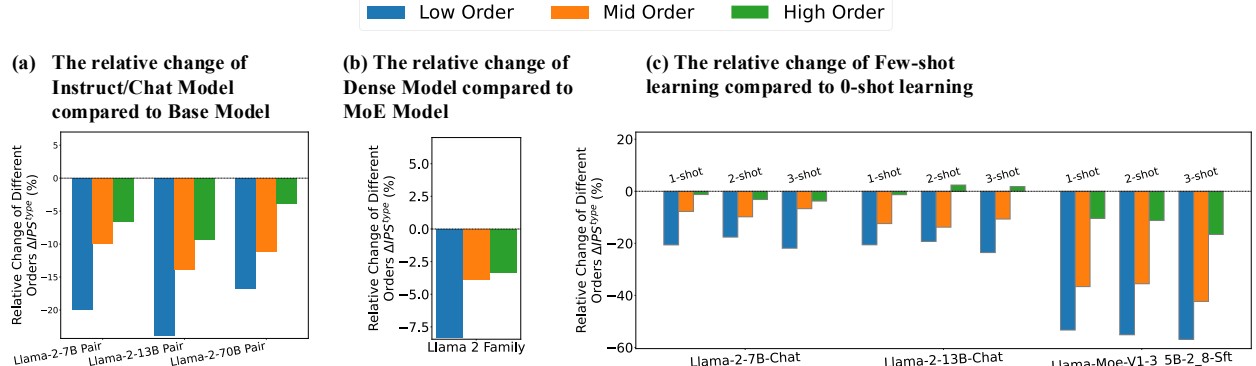

*Figure 13.* Comparing the relative change in the prompt sensitivity of low-, mid-, and high-order interactions for different factors.

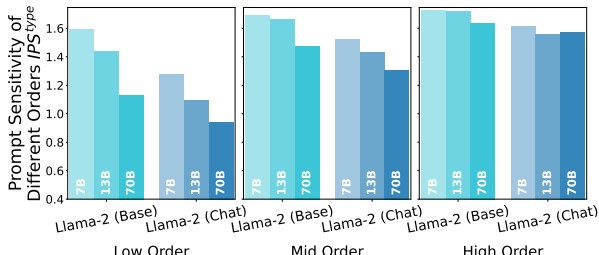

*Figure 14.* A comparison of prompt sensitivity at the order-level across different model scales.

plex prompt perturbations.

**Strategies for reducing the computational cost.** To reduce the computational cost of the interaction framework, recent studies (Chen et al., 2024; Cheng et al., 2025) used the following two strategies: *(1) Select informative words as input variables while treating uninformative ones (e.g., stop words) as fixed background context. (2) Merge related words into combined phrases as input variables*. The specific selection strategies are detailed in Appendix K.2. Results in Appendix K.3 show that using the above two strategies

on open-ended tasks yields the same conclusions. In addition to selection strategies, techniques specifically designed for efficiently computing sparse interactions to bypass exhaustive $O(2^n)$ evaluations (Kang et al., 2025; Butler et al., 2026) represent a clear path for reducing computational cost. Further details are provided in Appendix J.

## 5. Conclusion

In this paper, we propose an interaction-based metric to evaluate the prompt sensitivity of LLMs. We discover that employing supervised fine-tuning, increasing model scale, using dense over MoE architectures, and applying few-shot learning all serve to reduce the prompt sensitivity of LLMs. Our findings offer novel insights into both model designs and prompting methods for improving the robustness of LLMs. More crucially, we find that these factors achieve lower sensitivity primarily by reducing the sensitivity of low-order interactions, while the prompt sensitivity of high-order interactions remains at a relatively high level. In future studies, new training methods could be designed to increase the LLM's reliance on stable low-order interactions or, alternatively, to reduce the instability of high-order interactions.

# Acknowledgements

This work is partially supported by the Shanghai Science and Technology Commission (No. 25511102900), the National Nature Science Foundation of China (No.62376199,62576249), and the Shanghai Municipal Education Commission (No. 24CGA20).

# Impact Statement

This paper presents work whose goal is to advance the field of Machine Learning. There are many potential societal consequences of our work, none which we feel must be specifically highlighted here.

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

## A. Masking Strategies of Input Variables

In attribution method research, it is common to employ a specific token or embedding to mask the input variables of a deep neural network (DNN) (Lundberg & Lee, 2017; Ancona et al., 2019; Fong et al., 2019) and use changes in network outputs on the masked samples to estimate attributions of different input variables. The selection of a masking approach is complex, as each method has its weakness. For example, replacing input variables with the mean baseline value (the average of all samples) or the zero baseline value can introduce out-of-distribution signals, thereby providing the model with artificial information, such as uniform grey or black dots in an image (Dabkowski & Gal, 2017; Ancona et al., 2019; Sundararajan et al., 2017). Additionally, blurring image pixels using a Gaussian kernel (Fong & Vedaldi, 2017; Fong et al., 2019) as the masked state removes high-frequency signals but fails to eliminate low-frequency signals (Covert et al., 2021; Sturmfels et al., 2020).

Given these challenges, we adopt a token replacement strategy, which is standard for the text domain. This involves substituting the target input word with a dedicated [MASK] token at the embedding level. For example, to mask the word "green" in the input "He is a green hand," we would provide the LLM with the modified input "He is a [MASK] hand." This approach effectively nullifies the specific semantic contribution of the target word without introducing out-of-distribution artifacts, ensuring a clean and consistent baseline for our interaction analysis. For the specific [MASK] token for each LLM, please refer to Section 5 for details.

## B. Proof of Theorem

### B.1. Proof of Universal Matching Property

In the main body of the paper, for the sake of simplicity and clarity, we introduced the Universal Matching Property (Theorem 1) primarily through the lens of AND interactions. However, our empirical analysis and the underlying theoretical framework are built upon a more comprehensive AND-OR interaction framework. This extended framework, which incorporates both AND and OR interaction patterns, also adheres to the Universal Matching Property.

In this section, we provide the formal proof for the Universal Matching Property of the complete AND-OR interaction framework. This proof is more general and naturally subsumes the proof for the AND interaction framework presented as Theorem 1 in the main text. We will demonstrate that the output of the surrogate logical model, which is the sum of all AND-OR interaction effects, can perfectly match the output of the Deep Neural Network (DNN) for any masked sample.

The **surrogate logical model** $\phi(\cdot)$ is defined as follows:

$$\phi(\boldsymbol{x}_T) \triangleq \phi(\boldsymbol{x}_\emptyset) + \sum_{S \subseteq N, S \neq \emptyset} \mathbb{1}_{\text{AND}}(S \mid \boldsymbol{x}_T) \cdot I_S^{\text{AND}} + \sum_{S \subseteq N, S \neq \emptyset} \mathbb{1}_{\text{OR}}(S \mid \boldsymbol{x}_T) \cdot I_S^{\text{OR}}, \tag{4}$$

where the AND trigger function $\mathbb{1}_{\text{AND}}(S \mid \boldsymbol{x}_T) \in \{0, 1\}$ represents an **AND relationship** between input variables in $S$, which can also be termed **AND interaction pattern**; the OR trigger function $\mathbb{1}_{\text{OR}}(S \mid \boldsymbol{x}_T) \in \{0, 1\}$ represents an **OR relationship** between input variables in $S$, which can also be termed **OR interaction pattern**. The scalar weight $I_S^{\text{AND}}$ quantifies the effect of an AND relationship, which can also be termed **AND interaction effect**; the scalar weight $I_S^{\text{OR}}$ quantifies the effect of an OR relationship, which can also be termed **OR interaction effect**. An AND relationship is activated only by the joint presence of all input variables in the set $S$, *i.e.*, all input variables in $S$ are not masked. For instance, given the input sentence $\boldsymbol{x} =$ *"He is a green hand,"* the co-occurrence of the input variables in the set $S = \{green, hand\}$ contributes a numerical effect $I_S^{\text{AND}}$ that pushes the surrogate logical model's inference towards the semantic meaning of *"beginner."* If an AND interaction $S$ is triggered, *i.e.*, $\mathbb{1}_{\text{AND}}(S \mid \boldsymbol{x}_T) = 1$, the corresponding interaction effect $I_S^{\text{AND}}$ is added to the output of the logical model. Otherwise, if any word in $S$ is masked and the AND interaction is not triggered, *i.e.*, $\mathbb{1}_{\text{AND}}(S \mid \boldsymbol{x}_T) = 0$, the interaction effect $I_S^{\text{AND}}$ is not added to the output of the logical model. An OR relationship is activated by the presence of any of all input variables in the set $S$, *i.e.*, any input variables in $S$ are not masked. For instance, given the input sentence $\boldsymbol{x} =$ *"The service was terrible and the food was awful,"* the presence of any input variables in the set $S = \{terrible, awful\}$ contributes a numerical effect $I_S^{\text{OR}}$ that pushes the surrogate logical model's inference towards a negative sentiment classification. If an OR interaction $S$ is triggered, *i.e.*, $\mathbb{1}_{\text{OR}}(S \mid \boldsymbol{x}_T) = 1$, the corresponding interaction effect $I_S^{\text{OR}}$ is added to the output of the logical model. Otherwise, if all words in $S$ are masked and the OR interaction is not triggered, *i.e.*, $\mathbb{1}_{\text{OR}}(S \mid \boldsymbol{x}_T) = 0$, the interaction effect $I_S^{\text{OR}}$ is not added to the output of the logical model. $\boldsymbol{x}_\emptyset$ represents that all input variables in $N$ are masked.

**Definition of universal matching property for AND-OR interactions.** When the scalar weights in the surrogate logical

model $\phi(\cdot)$ are set to $I_S^{\text{AND}} = \sum_{T \subseteq S}(-1)^{|S|-|T|}v_{\text{and}}(\boldsymbol{x}_T)$ and $I_S^{\text{OR}} = -\sum_{T \subseteq S}(-1)^{|S|-|T|}v_{\text{or}}(\boldsymbol{x}_{N \setminus T})$, the output of $\phi(\cdot)$ can always match the output score of the DNN $v(\cdot)$, *i.e.*, $\forall T \subseteq N, v(\boldsymbol{x}_T) = \phi(\boldsymbol{x}_T)$. Here $v_{\text{and}}(\boldsymbol{x}_T) + v_{\text{or}}(\boldsymbol{x}_T) = v(\boldsymbol{x}_T)$.

We need to prove that given an input sample $\boldsymbol{x}$, for each masked sample $\{\boldsymbol{x}_T | T \subseteq N\}$, the network output score $v(\boldsymbol{x}_T) \in \mathbb{R}$ can be well matched by the surrogate logical model $\phi(\boldsymbol{x}_T)$. The surrogate logical model $\phi(\boldsymbol{x}_T)$ uses the sum of AND interactions and OR interactions to accurately explain/match the network output score $v(\boldsymbol{x}_T)$.

$$
\forall T \subseteq N, v(\boldsymbol{x}_T) = \phi(\boldsymbol{x}_T).
$$
$$
\phi(\boldsymbol{x}_T) = \phi(\boldsymbol{x}_\emptyset) + \sum_{S \subseteq N, S \neq \emptyset} \mathbb{1}_{\text{AND}}(S \mid \boldsymbol{x}_T) \cdot I_S^{\text{AND}} + \sum_{S \subseteq N, S \neq \emptyset} \mathbb{1}_{\text{OR}}(S \mid \boldsymbol{x}_T) \cdot I_S^{\text{OR}},
$$
$$
= \underbrace{v(\boldsymbol{x}_\emptyset) + \sum_{S \subseteq T, S \neq \emptyset} I_S^{\text{AND}}}_{v_{\text{and}}(\boldsymbol{x}_T)} + \underbrace{\sum_{S \subseteq N, S \cap T \neq \emptyset} I_S^{\text{OR}}}_{v_{\text{or}}(\boldsymbol{x}_T)} \tag{5}
$$

*Proof.* **(1) Universal matching property of AND interactions.** For all $2^n$ masked samples $\{\boldsymbol{x}_T \mid T \subseteq N\}$, what we need to prove is that the output $v_{\text{and}}(\boldsymbol{x}_T)$ of a DNN can be universally explained by all the interactions in $T \subseteq N$, *i.e.*, $\forall S \subseteq T, S \neq \emptyset, v_{\text{and}}(\boldsymbol{x}_T) = \sum_{S \subseteq T, S \neq \emptyset} I_S^{\text{AND}}(\boldsymbol{x}) = v(\boldsymbol{x}_\emptyset) + \sum_{S \subseteq T, S \neq \emptyset} I_S^{\text{AND}}$. Here, $v(\boldsymbol{x}_\emptyset) = v_{\text{and}}(\boldsymbol{x}_\emptyset)$.

According to the definition of the AND interaction, $I_S^{\text{AND}}(\boldsymbol{x}) = \sum_{L \subseteq S}(-1)^{|S|-|L|}v_{\text{and}}(\boldsymbol{x}_L)$. To simplify the computation of the sum of AND interactions $\sum_{S \subseteq T, S \neq \emptyset} I_S^{\text{AND}}(\boldsymbol{x}) = \sum_{S \subseteq T, S \neq \emptyset}\sum_{L \subseteq S}(-1)^{|S|-|L|}v_{\text{and}}(\boldsymbol{x}_L)$, we exchange the order of summation of the set $L \subseteq S \subseteq T$ and the set $S \supseteq L$. Given a set of input variables $L$, we compute all linear combinations of all sets $S$ containing $L$ with respect to the model outputs $v_{\text{and}}(\boldsymbol{x}_S)$, *i.e.*, $\sum_{S:L \subseteq S \subseteq T}(-1)^{|S|-|L|}v_{\text{and}}(\boldsymbol{x}_L)$. Then, we compute all summations over the set $L \subseteq T$ as $\sum_{S \subseteq T, S \neq \emptyset} I_S^{\text{AND}}(\boldsymbol{x}) = \sum_{L \subseteq T}\sum_{S:L \subseteq S \subseteq T}(-1)^{|S|-|L|}v_{\text{and}}(\boldsymbol{x}_L)$. Then, we can compute different cases of $L \subseteq S \subseteq T$ as follows:

(1) When $L = T = S$, $\sum_{S:L \subseteq S \subseteq T}(-1)^{|S|-|L|}v_{\text{and}}(\boldsymbol{x}_L) = (-1)^{|T|-|T|}v_{\text{and}}(\boldsymbol{x}_L) = v_{\text{and}}(\boldsymbol{x}_L)$.

(2) When $L \subseteq S \subseteq T, L \neq T$, let us consider the linear combinations of all sets $S$ with number $|S|$ for the model output $v_{\text{and}}(\boldsymbol{x}_L)$, respectively. Let $m := |S| - |L|, (0 \leq m \leq |T| - |L|)$, then there are a total of $C_{|T|-|L|}^m$ combinations of all sets $S$ of order $|S|$. Given $L$, accumulating the model outputs $v_{\text{and}}(\boldsymbol{x}_L)$ corresponding to all $S \supseteq L$, we can get $\sum_{S:L \subseteq S \subseteq T}(-1)^{|S|-|L|}v_{\text{and}}(\boldsymbol{x}_L) = v_{\text{and}}(\boldsymbol{x}_L) \cdot \underbrace{\sum_{m=0}^{|T|-|L|}C_{|T|-|L|}^m(-1)^m}_{=0} = 0$.

Considering all the cases, the complete derivation of the sum of AND interactions is as follows.

$$
\sum_{S \subseteq T, S \neq \emptyset} I_S^{\text{AND}}
$$
$$
= \sum_{S \subseteq T, S \neq \emptyset}\sum_{L \subseteq S}(-1)^{|S|-|L|}v_{\text{and}}(\boldsymbol{x}_L)
$$
$$
= \sum_{L \subseteq T}\sum_{S:L \subseteq S \subseteq T}(-1)^{|S|-|L|}v_{\text{and}}(\boldsymbol{x}_L) - v_{\text{and}}(\boldsymbol{x}_\emptyset) \tag{6}
$$
$$
= \underbrace{v_{\text{and}}(\boldsymbol{x}_T)}_{L=T} + \sum_{L \subseteq T, L \neq T}v_{\text{and}}(\boldsymbol{x}_L) \cdot \underbrace{\sum_{m=0}^{|T|-|L|}C_{|T|-|L|}^m(-1)^m}_{=0} - v_{\text{and}}(\boldsymbol{x}_\emptyset)
$$
$$
= v_{\text{and}}(\boldsymbol{x}_T) - v(\boldsymbol{x}_\emptyset)
$$

Therefore, we have proven that $\forall \emptyset \neq T \subseteq N, v_{\text{and}}(\boldsymbol{x}_T) = v(\boldsymbol{x}_\emptyset) + \sum_{S \subseteq T, S \neq \emptyset} I_S^{\text{AND}}$.

**(2) Universal matching theorem of OR interactions.** What we need to prove is that $\forall T \subseteq N, v_{\text{or}}(\boldsymbol{x}_T) = \sum_{S \in \{S:S \cap T \neq \emptyset\} \cup \{\emptyset\}} I_S^{\text{OR}} = \sum_{S:S \cap T \neq \emptyset} I_S^{\text{OR}}$. Here $I_\emptyset^{\text{OR}} = v_{\text{or}}(\boldsymbol{x}_\emptyset) = 0$.

According to the definition of the OR interaction, $I_S^{\text{OR}} := -\sum_{L \subseteq S}(-1)^{|S|-|L|}v_{\text{or}}(\boldsymbol{x}_{N \setminus L})$. To simplify the computation of the sum of OR interactions $\sum_{S:S \cap T \neq \emptyset} I_S^{\text{OR}} = \sum_{S:S \cap T \neq \emptyset}\left[-\sum_{L \subseteq S}(-1)^{|S|-|L|}v_{\text{or}}(\boldsymbol{x}_{N \setminus L})\right]$, we also exchange the order of summation of the set $L \subseteq S \subseteq N$ and the set $S : S \cap T = \emptyset$. Given a set of input variables $L$, we compute all linear combinations of all sets $S$ containing $L$ with respect to the model outputs $v_{\text{or}}(\boldsymbol{x}_{N \setminus L})$ , *i.e.*,

$\sum_{S:S\cap T\neq\emptyset,N\supseteq S\supseteq L}(-1)^{|S|-|L|}v_{\text{or}}(\boldsymbol{x}_{N\setminus L})$. Then, we compute all summations over the set $L\subseteq N$ as $\sum_{S:S\cap T\neq\emptyset}I_S^{\text{OR}}=-\sum_{L\subseteq N}\sum_{S:S\cap T\neq\emptyset,N\supseteq S\supseteq L}(-1)^{|S|-|L|}v_{\text{or}}(\boldsymbol{x}_{N\setminus L})$. Then, we can compute different cases of $L\subseteq S\subseteq N,S\cap T\neq\emptyset$ as follows:

(1) When $L=N$ (then $S=N$), $\sum_{S:S\cap T\neq\emptyset,S\supseteq L}(-1)^{|S|-|L|}v_{\text{or}}(\boldsymbol{x}_{N\setminus L})=(-1)^{|N|-|N|}v_{\text{or}}(\boldsymbol{x}_{\emptyset})=v_{\text{or}}(\boldsymbol{x}_{\emptyset})=0$, Here $I_{\emptyset}^{\text{OR}}=v_{\text{or}}(\boldsymbol{x}_{\emptyset})=0$.

(2) When $L=N\setminus T$, for all sets $S:S\supseteq L,S\cap T\neq\emptyset$ (then $S\neq N\setminus T,S\neq L$), let us consider the linear combinations of all sets $S$ with number $|S|$ for the model output $v_{\text{or}}(\boldsymbol{x}_T)$, respectively. Let $|S'|:=|S|-|L|$, $(1\leq|S'|\leq|T|)$, then there are a total of $C_{|T|}^{|S'|}$ combinations of all sets $S$ of order $|S|$. Thus, $\sum_{S:S\cap T\neq\emptyset,S\supseteq L}(-1)^{|S|-|L|}v_{\text{or}}(\boldsymbol{x}_{N\setminus L})=v_{\text{or}}(\boldsymbol{x}_T)\cdot\underbrace{\sum_{|S'|=1}^{|T|}C_{|T|}^{|S'|}(-1)^{|S'|}}_{=-1}=-v_{\text{or}}(\boldsymbol{x}_T)$.

(3) When $L\cap T\neq\emptyset,L\neq N$, for all sets $S:S\supseteq L,S\cap T\neq\emptyset$, let us consider the linear combinations of all sets $S$ with number $|S|$ for the model output $v_{\text{or}}(\boldsymbol{x}_T)$, respectively. Let us split $|S|-|L|$ into $|S'|$ and $|S''|$, i.e.,$|S|-|L|=|S'|+|S''|$, where $S'=\{i|i\in S,i\notin L,i\in N\setminus T\}$, $S''=\{i|i\in S,i\notin L,i\in T\}$ (then $0\leq|S''|\leq|T|-|T\cap L|$) and $S'+S''+L=S$. Thus, there are a total of $C_{|T|-|T\cap L|}^{|S''|}$ combinations of all sets $S''$ of order $|S''|$. Thus,

$\sum_{S:S\cap T\neq\emptyset,S\supseteq L}(-1)^{|S|-|L|}v_{\text{or}}(\boldsymbol{x}_{N\setminus L})=v_{\text{or}}(\boldsymbol{x}_{N\setminus L})\cdot\sum_{S'\subseteq N\setminus T\setminus L}\underbrace{\sum_{|S''|=0}^{|T|-|T\cap L|}C_{|T|-|T\cap L|}^{|S''|}(-1)^{|S'|+|S''|}}_{=0}=0.$

(4) When $L\cap T=\emptyset,L\neq N\setminus T$, let us split $|S|-|L|$ into $|S'|$ and $|S''|$, i.e.,$|S|-|L|=|S'|+|S''|$, where $S'=\{i|i\in S,i\notin L,i\in N\setminus T\}$, $S''=\{i|i\in S,i\in T\}$ (then $0\leq|S''|\leq|T|$) and $S'+S''+L=S$. Thus, there are a total of $C_{|T|}^{|S''|}$ combinations of all sets $S''$ of order $|S''|$. Thus, $\sum_{S:S\cap T\neq\emptyset,S\supseteq L}(-1)^{|S|-|L|}v_{\text{or}}(\boldsymbol{x}_{N\setminus L})=v_{\text{or}}(\boldsymbol{x}_{N\setminus L})\cdot$

$\sum_{S'\subseteq N\setminus T\setminus L}\underbrace{\sum_{|S''|=0}^{|T|}C_{|T|}^{|S''|}(-1)^{|S'|+|S''|}}_{=0}=0.$

Considering all the cases, the complete derivation of the sum of OR interactions is as follows.

$$\begin{aligned}
\sum_{S:S\cap T\neq\emptyset}I_S^{\text{OR}} &= \sum_{S:S\cap T\neq\emptyset}\left[-\sum_{L\subseteq S}(-1)^{|S|-|L|}v_{\text{or}}(\boldsymbol{x}_{N\setminus L})\right]\\
&= -\sum_{L\subseteq N}\sum_{S:S\cap T\neq\emptyset,N\supseteq S\supseteq L}(-1)^{|S|-|L|}v_{\text{or}}(\boldsymbol{x}_{N\setminus L})\\
&= -\left[\sum_{|S'|=1}^{|T|}C_{|T|}^{|S'|}(-1)^{|S'|}\right]\cdot\underbrace{v_{\text{or}}(\boldsymbol{x}_T)}_{L=N\setminus T}-\underbrace{v_{\text{or}}(\boldsymbol{x}_{\emptyset})}_{L=N}\\
&\quad -\sum_{L\cap T\neq\emptyset,L\neq N}\left[\sum_{S'\subseteq N\setminus T\setminus L}\left(\sum_{|S''|=0}^{|T|-|T\cap L|}C_{|T|-|T\cap L|}^{|S''|}(-1)^{|S'|+|S''|}\right)\right]\cdot v_{\text{or}}(\boldsymbol{x}_{N\setminus L})\\
&\quad -\sum_{L\cap T=\emptyset,L\neq N\setminus T}\left[\sum_{S'\subseteq N\setminus T\setminus L}\left(\sum_{|S''|=0}^{|T|}C_{|T|}^{|S''|}(-1)^{|S'|+|S''|}\right)\right]\cdot v_{\text{or}}(\boldsymbol{x}_{N\setminus L})\\
&= -(-1)\cdot v_{\text{or}}(\boldsymbol{x}_T)-v_{\text{or}}(\boldsymbol{x}_{\emptyset})-\sum_{L\cap T\neq\emptyset,L\neq N}\left[\sum_{S'\subseteq N\setminus T\setminus L}0\right]\cdot v_{\text{or}}(\boldsymbol{x}_{N\setminus L})\\
&\quad -\sum_{L\cap T=\emptyset,L\neq N\setminus T}\left[\sum_{S'\subseteq N\setminus T\setminus L}0\right]\cdot v_{\text{or}}(\boldsymbol{x}_{N\setminus L})\\
&= v_{\text{or}}(\boldsymbol{x}_T)-v_{\text{or}}(\boldsymbol{x}_{\emptyset})\\
&= v_{\text{or}}(\boldsymbol{x}_T)
\end{aligned} \tag{7}$$

Therefore, we have proven that $\forall T \subseteq N, v_{\text{or}}(\boldsymbol{x}_T) = \sum_{S:S\cap T\neq\emptyset} I_S^{\text{OR}}$.

**(3) Universal matching theorem of AND-OR interactions.** With the universal matching property of AND interactions and the universal matching property of OR interactions, we can easily get $v(\boldsymbol{x}_T) = \phi(\boldsymbol{x}_T) = v_{\text{and}}(\boldsymbol{x}_T) + v_{\text{or}}(\boldsymbol{x}_T) = v(\boldsymbol{x}_\emptyset) + \sum_{S\subseteq T,S\neq\emptyset} I_S^{\text{AND}} + \sum_{S\subseteq N,S\cap T\neq\emptyset} I_S^{\text{OR}}$, thus, we obtain the universal matching property of AND-OR interactions. $\qquad\square$

### B.2. Proof of Sparsity Property

Given all the masked samples $\{\boldsymbol{x}_T \mid T \subseteq N\}$, the surrogate logical model $\phi(\boldsymbol{x}_T)$ only utilizes a small set of salient AND interactions in $\Omega^{\text{AND}}$ and salient OR interactions in $\Omega^{\text{OR}}$ to approximate the network output score $v(\boldsymbol{x}_T)$. That is, the network's output can be well approximated by a small set of AND-OR interactions.

$$v(\boldsymbol{x}_T) = \phi(\boldsymbol{x}_T) \approx v(\boldsymbol{x}_\emptyset) + \sum_{S\subseteq T,S\neq\emptyset,S\in\Omega_{\text{AND}}} I_S^{\text{AND}} + \sum_{S\subseteq T,S\neq\emptyset,S\in\Omega_{\text{OR}}} I_S^{\text{OR}} \qquad (8)$$

*Proof.* It has been proven by Ren et al. (2024a) that under three common conditions[7], the output score $v_{\text{and}}(\boldsymbol{x}_T)$ of a well-trained DNN on all $2^n$ masked samples $\{\boldsymbol{x}_T|T\subseteq N\}$ could be universally estimated by a small number of AND interactions $T \in \Omega^{\text{AND}}$ with salient interaction effects $I_S^{\text{AND}}$, *s.t.*, $|\Omega^{\text{AND}}| \ll 2^n$, *i.e.*, $\forall T \subseteq N, v_{\text{and}}(\boldsymbol{x}_T) = \sum_{S\subseteq T,S\neq\emptyset} I_S^{\text{AND}} \approx \sum_{S\subseteq T,S\neq\emptyset,S\in\Omega^{\text{AND}}} I_S^{\text{AND}}$. According to Eq. (6), $v_{\text{and}}(\boldsymbol{x}_T) = v(\boldsymbol{x}_\emptyset) + \sum_{S\subseteq T,S\neq\emptyset} I_S^{\text{AND}}$. Therefore, $v_{\text{and}}(\boldsymbol{x}_T) \approx v(\boldsymbol{x}_\emptyset) + \sum_{S\subseteq T,S\neq\emptyset,S\in\Omega^{\text{AND}}} I_S^{\text{AND}}$.

Besides, as proven in Section C, the OR interaction can be considered as a special AND interaction. Thus, the confidence score $v_{\text{or}}(\boldsymbol{x}_T)$ of a well-trained DNN on all $2^n$ masked samples $\{\boldsymbol{x}_T|T \subseteq N\}$ could be universally estimated by a small number of OR interactions $T \in \Omega^{\text{OR}}$ with salient interaction effects $I_S^{\text{OR}}$, *s.t.*, $|\Omega^{\text{OR}}| \ll 2^n$. Similarly, $v_{\text{or}}(\boldsymbol{x}_T) = \sum_{S\subseteq T,S\neq\emptyset} I_S^{\text{OR}} \approx \sum_{S\subseteq T,S\neq\emptyset,S\in\Omega^{\text{OR}}} I_S^{\text{OR}}$

Thus, for each randomly masked sample $\boldsymbol{x}_T, T \subseteq N$, the surrogate logical model $\phi(\boldsymbol{x}_T)$ can use a small number of salient AND-OR interactions to approximate the network output score $v(\boldsymbol{x}_T)$, *i.e.*, $v(\boldsymbol{x}_T) = \phi(\boldsymbol{x}_T) = v_{\text{and}}(\boldsymbol{x}_T) + v_{\text{or}}(\boldsymbol{x}_T) \approx (\boldsymbol{x}_\emptyset) + \sum_{S\subseteq T,S\neq\emptyset,S\in\Omega_{\text{AND}}} I_S^{\text{AND}} + \sum_{S\subseteq T,S\neq\emptyset,S\in\Omega_{\text{OR}}} I_S^{\text{OR}}$.

$\qquad\square$

## C. OR Interactions Can Be Considered as Special AND Interactions

If we reverse the definition of the masked state and the unmasked state of the input variable, the OR interaction $I_S^{\text{OR}}$ can be considered as a special kind of AND interaction $I_S^{\text{AND}}$.

Given an input sample $\boldsymbol{x} \in \mathbb{R}^n$ and the output score of a DNN as $v(\cdot)$, if we randomly mask input variables in $\boldsymbol{x}$, we can get all $2^n$ masked samples. Let $\boldsymbol{x}_S$ denote the certain masked input sample when input variables in $N \setminus S$ are all masked and input variables in S are kept unchanged.

$$(\boldsymbol{x}_S)_i = \begin{cases} x_i, & i \in S \\ b_i, & i \in N \setminus S \end{cases} \qquad (9)$$

where $\mathbf{b} \in \mathbb{R}^n$ are baseline values to represent the masked state of input variables.

If we reverse the definition of the masked state and the unmasked state of an input variable, *i.e.*, we consider $\mathbf{b}$ as the input sample and consider $\boldsymbol{x}$ as the masked state, then the masked sample $\widetilde{\boldsymbol{x}}_S$ can be defined as follows.

$$(\widetilde{\boldsymbol{x}}_S)_i = \begin{cases} b_i, & i \in S \\ x_i, & i \in N \setminus S \end{cases} \qquad (10)$$

Thus, we can get $\boldsymbol{x}_{N\setminus S} = \widetilde{\boldsymbol{x}}_S$. To simplify the analysis, let us assume $v_{\text{and}}(\boldsymbol{x}_S) = v_{\text{or}}(\boldsymbol{x}_S) = 0.5v(\boldsymbol{x}_S)$, then the OR

---

[7]Here are the three conditions: (1) The DNN doesn't encode extremely high-order AND interactions. (2) The DNN performs effectively on masked samples and exhibits greater confidence as the input sample is less masked. (3) When we increase the number of masked input variables, the confidence of the DNN does not drop significantly.

interaction $I_S^{\text{OR}}$ can be regarded as a specific AND interaction $I_S^{\text{AND}}(\widetilde{x})$ as follows.

$$
\begin{aligned}
I_S^{\text{OR}}(\mathbf{x}) &= -\sum_{T \subseteq S}(-1)^{|S|-|T|} v_{\text{or}}(\boldsymbol{x}_{N \setminus T}), \\
&= -\sum_{T \subseteq S}(-1)^{|S|-|T|} v_{\text{or}}(\widetilde{\boldsymbol{x}}_T), \\
&= -\sum_{T \subseteq S}(-1)^{|S|-|T|} v_{\text{and}}(\widetilde{\boldsymbol{x}}_T), \\
&= -I_S^{\text{AND}}(\widetilde{\boldsymbol{x}}).
\end{aligned}
\tag{11}
$$

Now we have proven that OR interactions can be considered as special AND interactions.

## D. Details of Extracting the Sparsest AND-OR Interactions

We follow Li & Zhang (2023) to extract AND-OR interactions. Given a masked sample $\boldsymbol{x}_T$, the output score of the network $v(\boldsymbol{x}_T)$ can be decomposed into a combination of AND interaction and OR interaction, *i.e.*, $v(\boldsymbol{x}_T) = v_{\text{and}}(\boldsymbol{x}_T) + v_{\text{or}}(\boldsymbol{x}_T)$. Specifically, $v_{\text{and}}(\boldsymbol{x}_T) = 0.5 \cdot v(\boldsymbol{x}_T) + \gamma_T$ and $v_{\text{or}}(\boldsymbol{x}_T) = 0.5 \cdot v(\boldsymbol{x}_T) - \gamma_T$, where $\{\gamma_T \mid T \subseteq N\}$ is a set of learnable parameters. The parameters $\{\gamma_T\}$ were trained through minimizing the following LASSO-like loss to obtain sparse interactions:

$$
\min_{\{\gamma_T\}} \sum_{S \subseteq N} |I_S^{\text{AND}}(\boldsymbol{x})| + |I_S^{\text{OR}}(\boldsymbol{x})|,
\tag{12}
$$

where $I_S^{\text{AND}}(\boldsymbol{x}) = \sum_{T \subseteq S}(-1)^{|S|-|T|} v_{\text{and}}(\boldsymbol{x}_T) = \sum_{T \subseteq S}(-1)^{|S|-|T|}(0.5 \cdot v(\boldsymbol{x}_T) + \gamma_T)$ and $I_S^{\text{OR}}(\boldsymbol{x}) = -\sum_{T \subseteq S}(-1)^{|S|-|T|} v_{\text{or}}(\boldsymbol{x}_{N \setminus T}) = -\sum_{T \subseteq S}(-1)^{|S|-|T|}(0.5 \cdot v(\boldsymbol{x}_T) - \gamma_T)$. Thus, we can extract the sparsest set of AND-OR interactions.

## E. Experimental Details

### E.1. Computing Infrastructure

We conducted all our experiments on four NVIDIA Tesla V100-DGXS GPUs, each with 32 GB of VRAM. The software environment consisted of NVIDIA Driver version 570.133.07 and CUDA 12.8.

For all of the evaluated LLMs, we used a torch.float16 data type, which provides a standard level of precision for inference tasks.

### E.2. Model Details

We conduct experiments on 50 open-source LLMs from 6 major model families. A comprehensive list of all evaluated models is provided in Table 1. To facilitate a controlled analysis of the factors influencing prompt sensitivity, we group these models into specific subsets for each comparison, as detailed below.

(1) **Instruct/Chat vs. Base Models.** To investigate the impact of the alignment process, we form pairs of instruct/chat models and their corresponding base models. This comparison includes models from the Llama, Mistral, Qwen, InternLM, and Olmo families. The main LLMs used for this comparison are:

- *Llama Family:*
    - `llama-2-7b-chat` vs. `llama-2-7b`
    - `llama-2-13b-chat` vs. `llama-2-13b`
    - `llama-2-70b-chat` vs. `llama-2-70b`
    - `llama-3-8b-instruct` vs. `llama-3-8b`

- *Mistral Family:*
    - `mistral-7b-v0.3-instruct` vs. `mistral-7b-v0.3`
    - `mixtral-8x7b-instruct` vs. `mixtral-8x7b`

- *Qwen Family:*

*Table 1.* A comprehensive list and characteristics of LLMs, grouped by model family and model series.

| Model Family | Model Series | Model | Type | Architecture | Scale |
|---|---|---|---|---|---|
| Llama (10 models) | Llama 2 (6 models) | Llama-2-7b | Base | Dense | 7B |
| | | Llama-2-7b-chat | Chat | Dense | 7B |
| | | Llama-2-13b | Base | Dense | 13B |
| | | Llama-2-13b-chat | Chat | Dense | 13B |
| | | Llama-2-70b | Base | Dense | 70B |
| | | Llama-2-70b-chat | Chat | Dense | 70B |
| | Llama 3 (2 models) | Llama-3-8b | Base | Dense | 8B |
| | | Llama-3-8b-instruct | Instruct | Dense | 8B |
| | Llama MoE (2 models) | Llama-moe-v1-3_5b-2_8-sft | Instruct | MoE | 3.5B (Activated) |
| | | Llama-moe-v2-3_8b-2_8-sft | Instruct | MoE | 3.8B (Activated) |
| Mistral (4 models) | Mistral (2 models) | Mistral-7b-v0.3 | Base | Dense | 7B |
| | | Mistral-7b-v0.3-instruct | Instruct | Dense | 7B |
| | Mixtral (2 models) | Mixtral-8x7b | Base | MoE | 13B (Activated) |
| | | Mixtral-8x7b-instruct | Instruct | MoE | 13B (Activated) |
| Qwen (25 models) | Qwen 1.5 MoE (2 models) | Qwen1.5-moe-a2.7b-chat | Chat | MoE | 2.7B (Activated) |
| | | Qwen1.5-moe-a2.7b | Base | MoE | 2.7B (Activated) |
| | Qwen 2 (5 models) | Qwen2-7b | Base | Dense | 7B |
| | | Qwen2-0.5b-instruct | Instruct | Dense | 0.5B |
| | | Qwen2-1.5b-instruct | Instruct | Dense | 1.5B |
| | | Qwen2-7b-instruct | Instruct | Dense | 7B |
| | | Qwen2-72b-instruct | Instruct | Dense | 72B |
| | Qwen 2.5 (7 models) | Qwen2.5-0.5b-instruct | Instruct | Dense | 0.5B |
| | | Qwen2.5-1.5b-instruct | Instruct | Dense | 1.5B |
| | | Qwen2.5-3b-instruct | Instruct | Dense | 3B |
| | | Qwen2.5-7b-instruct | Instruct | Dense | 7B |
| | | Qwen2.5-14b-instruct | Instruct | Dense | 14B |
| | | Qwen2.5-32b-instruct | Instruct | Dense | 32B |
| | | Qwen2.5-72b-instruct | Instruct | Dense | 72B |
| | Qwen 3 (9 models) | Qwen3-0.6b-instruct | Instruct | Dense | 0.6B |
| | | Qwen3-1.7b-instruct | Instruct | Dense | 1.7B |
| | | Qwen3-4b | Base | Dense | 4B |
| | | Qwen3-4b-instruct | Instruct | Dense | 4B |
| | | Qwen3-8b | Base | Dense | 8B |
| | | Qwen3-8b-instruct | Instruct | Dense | 8B |
| | | Qwen3-14b | Base | Dense | 14B |
| | | Qwen3-14b-instruct | Instruct | Dense | 14B |
| | | Qwen3-32b-instruct | Instruct | Dense | 32B |
| | Qwen 3 A3B (2 models) | Qwen3-30b-a3b-instruct | Instruct | MoE | 3B (Activated) |
| | | Qwen3-30b-a3b | Base | MoE | 3B (Activated) |
| Olmo (9 models) | Olmo v1 (3 models) | Olmo-1b | Base | Dense | 1B |
| | | Olmo-7b | Base | Dense | 7B |
| | | Olmo-7b-instruct | Instruct | Dense | 7B |
| | Olmo v2 (4 models) | Olmo-2-1b-instruct | Instruct | Dense | 1B |
| | | Olmo-2-7b-instruct | Instruct | Dense | 7B |
| | | Olmo-2-13b-instruct | Instruct | Dense | 13B |
| | | Olmo-2-32b-instruct | Instruct | Dense | 32B |
| | OlmoE (2 models) | Olmoe-7b | Base | MoE | 1B (Activated) |
| | | Olmoe-7b-instruct | Instruct | MoE | 1B (Activated) |
| InternLM (2 models) | InternLM 2 (2 models) | Internlm2-7b | Base | Dense | 7B |
| | | Internlm2-chat-7b | Chat | Dense | 7B |

- – `qwen3-4b-instruct` vs. `qwen3-4b`
- – `qwen3-8b-instruct` vs. `qwen3-8b`
- – `qwen3-14b-instruct` vs. `qwen3-14b`
- – `qwen1.5-moe-a2.7b-chat` vs. `qwen1.5-moe-a2.7b`
- – `qwen3-30b-a3b-instruct` vs. `qwen3-30b-a3b`

- *Olmo Family:*

  - – `olmo-7b-instruct` vs. `olmo-7b`
  - – `olmoe-7b-instruct` vs. `olmoe-7b`

- *InternLM Family:*

  - – `internlm2-chat-7b` vs. `internlm2-7b`

(2) **Dense vs. MoE Models.** To analyze the effect of architecture, we compare dense and Mixture-of-Experts (MoE) models, primarily within the same model family to control for other variables. The main LLMs used for this comparison are:

- *Llama Family:*

  - – Dense: `llama-2-7b`, `llama-2-7b-chat`, `llama-2-13b`, `llama-2-13b-chat`, `llama-2-70b`, `llama-2-70b-chat`, `llama-3-8b`, `llama-3-8b-instruct`.
  - – MoE: `llama-moe-v1-3_5b-2_8-sft`, `llama-moe-v2-3_8b-2_8-sft`.

- *Mistral Family:*

  - – Dense: `mistral-7b-v0.3`, `mistral-7b-v0.3-instruct`.
  - – MoE: `mixtral-8x7b`, `mixtral-8x7b-instruct`.

- *Qwen Family:*

  - – Dense: `qwen2-7b`, `qwen2-7b-instruct`, `qwen2-72b-instruct`, `qwen2.5-7b-instruct`, `qwen2.5-14b-instruct`, `qwen2.5-32b-instruct`, `qwen2.5-72b-instruct`, `qwen3-4b`, `qwen3-4b-instruct`, `qwen3-8b`, `qwen3-8b-instruct`, `qwen3-14b`, `qwen3-14b-instruct`, `qwen3-32b-instruct`.
  - – MoE: `qwen1.5-moe-a2.7b`, `qwen1.5-moe-a2.7b-chat`, `qwen3-30b-a3b`, `qwen3-30b-a3b-instruct`.

- *Olmo Family:*

  - – Dense: `olmo-7b`, `olmo-7b-instruct`, `olmo-2-7b-instruct`, `olmo-2-13b-instruct`, `olmo-2-32b-instruct`.
  - – MoE: `olmoe-7b`, `olmoe-7b-instruct`.

(3) **Model Scale.** To study the impact of model scale, we analyze a series of LLMs from the same family and with the same training paradigm but with varying parameter counts. The main LLMs used for this comparison are:

- *Llama-2 (Base):* `7b`, `13b`, `70b`.

- *Llama-2 (Chat):* `7b`, `13b`, `70b`.

- *Qwen2 (Instruct):* `0.5b`, `1.5b`, `7b`, `72b`.

- *Qwen2.5 (Instruct):* `0.5b`, `1.5b`, `3b`, `7b`, `14b`, `32b`, `72b`.

- *Qwen3 (Instruct):* `0.6b`, `1.7b`, `4b`, `8b`, `14b`, `32b`.

- *Olmo-2 (Instruct):* `1b`, `7b`, `13b`, `32b`.

### E.3. Generation Configuration of LLMs

To ensure reproducible results, we employ a greedy search strategy for all LLMs. This is achieved by setting the "*do_sample*" parameter to "*False*" in our generation configuration. When "*do_sample=False*", the LLM selects the token with the highest probability as the next token in the sequence. By adopting this greedy approach, we eliminate the randomness inherent in sampling-based methods. The configuration ensures that for a given input, the same LLM will generate the exact same output every time, which is a critical requirement for the replicability of our experiments.

### E.4. How to Mask Input Words For Different LLMs

To compute interactions, we follow the approach of Cheng et al. (2025) and mask the words in $N \setminus S$ by replacing them with a LLM-specific [MASK] token. Our selection of this token follows a prioritized strategy: (1) We preferentially use the LLM's designated unknown (`<unk>`) token. (2) If an unknown token is not available or suitable, we use the padding (`<pad>`) token as a fallback. Since the specific token strings and their corresponding IDs vary across different LLMs, the exact mask token used for each LLM is detailed below:

- For `llama-2-7b`, `llama-2-7b-chat`, `llama-moe-v1-3.5b-2.8-sft`, `mistral-7b-v0.3`, `mistral-7b-v0.3-instruct`, `mixtral-8x7b`, `mixtral-8x7b-instruct`, `internlm2-7b`, `internlm2-chat-7b`, we use the `<unk>` token (ID: 0) to mask words.

- For `llama-2-13b`, `llama-2-13b-chat`, `llama-2-70b`, `llama-2-70b-chat`, we use the `<|pad_token|>` token (ID: 0) to mask words.

- For `llama-3-8b`, `llama-3-8b-instruct`, we use the `<|pad_token|>`/`<|reserved_special_token_250|>` token (ID: 128255) to mask words.

- For `llama-moe-v2-3.8b-2.8-sft`, we use the `<|pad_token|>`/`<|eot_id|>` token (ID: 128009) to mask words.

- For `qwen2-7b`, we use the `<|PAD_TOKEN|>` token (ID: 151646) to mask words.

- For a large group of Qwen models, including `qwen2-0.5b-instruct`, `qwen2-1.5b-instruct`, `qwen2-7b-instruct`, `qwen2-72b-instruct`, `qwen2.5` series, `qwen3-0.6b-instruct`, `qwen3-1.7b-instruct`, `qwen3-4b` series, `qwen3-32b-instruct`, `qwen1.5-moe` series, and `qwen3-30b-a3b` series, we use the `<|pad_token|>`/`<|endoftext|>` token (ID: 151643) to mask words.

- For `qwen3-8b`, `qwen3-8b-instruct`, `qwen3-14b`, `qwen3-14b-instruct`, we use the `<|pad_token|>`/`<|vision_pad|>` token (ID: 151654) to mask words.

- For the Olmo V1 series models, including `olmo-1b`, `olmo-7b`, `olmo-7b-instruct`, `olmoe-7b`, and `olmoe-7b-instruct`, we use the `<|padding|>` token (ID: 1) to mask words.

- For the Olmo V2 series models, including `olmo-2-1b-instruct`, `olmo-2-7b-instruct`, `olmo-2-13b-instruct`, and `olmo-2-32b-instruct`, we use the `<|pad_token|>`/`<|endoftext|>` token (ID: 100257) to mask words.

### E.5. Prompt Templates

To systematically evaluate the prompt sensitivity of LLMs, we designed a set of five distinct prompt templates. As illustrated in Figure 15, these templates are derived from a base prompt template (i.e., Prompt Template 1) through a series of subtle, semantically irrelevant modifications. These variations include changes in letter case, e.g., "Answers" vs. "ANSWERS" and alterations to separators, e.g., ":" vs. "::" or the format of option markers, e.g., "A." vs. "A)". Crucially, these changes only affect the superficial formatting while preserving the core semantic meaning of the prompt template.

In our experimental procedure, for a given input, which consists of a question and options, we apply each of the five prompt templates to generate five prompts. For every unique pair of these five prompts, we then calculate the prompt sensitivity by quantifying the change in the interactions among the input variables (*i.e.*, words within the question and options). This procedure allows us to precisely measure how much the LLM's interaction patterns of the core input are perturbed by superficial changes in the prompt template, thus evaluating the prompt sensitivity of LLMs.

| *Prompt Template* 1: | *Prompt Template* 2: | *Prompt Template* 3: | *Prompt Template* 4: | *Prompt Template* 5: |
|---|---|---|---|---|
| {Question}
**Answers:**
A.{Option A}
B.{Option B}
C.{Option C}
D.{Option D}
**Answer:** | {Question}
**ANSWERS:**
A.{Option A}
B.{Option B}
C.{Option C}
D.{Option D}
**ANSWER:** | {Question}
**Answers::**
A.{Option A}
B.{Option B}
C.{Option C}
D.{Option D}
**Answer::** | {Question}
**ANSWERS::**
A.{Option A}
B.{Option B}
C.{Option C}
D.{Option D}
**ANSWER::** | {Question}
**Answers:**
A){Option A}
B){Option B}
C){Option C}
D){Option D}
**Answer:** |

*Figure 15.* Different prompt templates. Red parts show the difference between the current prompt template with the first prompt template, *i.e.*, Prompt Template 1.

## E.6. Few-shot Learning Templates

For this experiment, we selected the pair of prompt templates that exhibited the highest average prompt sensitivity in the 0-shot setting, aiming to test if few-shot learning could help the most severe situation. To investigate whether few-shot learning can mitigate high prompt sensitivity, we conducted a follow-up experiment. We selected **Prompt Template 1** and **Prompt Template 4** from Figure 15 for this analysis, as this pair exhibited the highest average prompt sensitivity in our 0-shot setting. This allowed us to test the efficacy of few-shot learning in the most challenging scenario.

Based on these two base templates, we constructed few-shot learning prompts with one, two, and three in-context examples (*i.e.*, 1-shot, 2-shot, and 3-shot learning), as illustrated in Figure 16. The examples were formulated using certain questions and their corresponding answers, randomly selected from a set of datasets that are not included in the test set. The structure of each example is related to its corresponding prompt template. For instance, the first example (`Example 1`) is formatted differently for each template:

- **Example 1 For Prompt Template 1:**

```
Question: Which type of precipitation consists of frozen rain drops?
Answers:
A.sleet
B.hail
C.snow
D.fog
Answer: A
```

- **Example 1 For Prompt Template 4 (Note the different format):**

```
Question: Which type of precipitation consists of frozen rain drops?
ANSWERS::
A.sleet
B.hail
C.snow
D.fog
ANSWER:: A
```

The other two examples (`Example 2` and `Example 3`) are presented below:

- **Example 2 For Prompt Template 1:**

```
Question: Decayed prehistoric plants have helped in the formation of
Answers:
A.coal, shale, and quartz.
B.coal, oil, and gas.
C.shale, quartz, and coal.
D.oil, shale, and granite.
Answer: B
```

- **Example 2 For Prompt Template 4:**

```
Question: Decayed prehistoric plants have helped in the formation of
ANSWERS::
A.coal, shale, and quartz.
B.coal, oil, and gas.
C.shale, quartz, and coal.
D.oil, shale, and granite.
ANSWER:: B
```

- **Example 3 For Prompt Template 1:**

```
Question: Which describes a material that is not a food?
Answers:
A.It stores energy but not nutrients.
B.It does not store energy or nutrients.
C.It stores energy and nutrients.
D.It does not store energy but stores nutrients.
Answer: B
```

- **Example 3 For Prompt Template 4:**

```
Question: Which describes a material that is not a food?
ANSWERS::
A.It stores energy but not nutrients.
B.It does not store energy or nutrients.
C.It stores energy and nutrients.
D.It does not store energy but stores nutrients.
ANSWER:: B
```

## 1-shot Learning

*Prompt Template* **1 :**

```
Instruction: Read the following question and
the four options provided. Choose the single
best answer and provide only its corresponding
letter.
Example:
{Example 1}

Question: {Question}
Answers:
A.{Option A}
B.{Option B}
C.{Option C}
D.{Option D}
Answer:
```

*Prompt Template* **2 :**

```
Instruction: Read the following question and
the four options provided. Choose the single
best answer and provide only its corresponding
letter.
Example:
{Example 1}

Question: {Question}
ANSWERS::
A.{Option A}
B.{Option B}
C.{Option C}
D.{Option D}
ANSWER::
```

## 2-shot Learning

*Prompt Template* **1 :**

```
Instruction: Read the following question and
the four options provided. Choose the single
best answer and provide only its corresponding
letter.
Example:
{Example 1}

{Example 2}

Question: {Question}
Answers:
A.{Option A}
B.{Option B}
C.{Option C}
D.{Option D}
Answer:
```

*Prompt Template* **2 :**

```
Instruction: Read the following question and
the four options provided. Choose the single
best answer and provide only its corresponding
letter.
Example:
{Example 1}

{Example 2}

Question: {Question}
ANSWERS::
A.{Option A}
B.{Option B}
C.{Option C}
D.{Option D}
ANSWER::
```

## 3-shot Learning

*Prompt Template* **1 :**

```
Instruction: Read the following question and
the four options provided. Choose the single
best answer and provide only its corresponding
letter.
Example:
{Example 1}

{Example 2}

{Example 3}

Question: {Question}
Answers:
A.{Option A}
B.{Option B}
C.{Option C}
D.{Option D}
Answer:
```

*Prompt Template* **2 :**

```
Instruction: Read the following question and
the four options provided. Choose the single
best answer and provide only its corresponding
letter.
Example:
{Example 1}

{Example 2}

{Example 3}

Question: {Question}
ANSWERS::
A.{Option A}
B.{Option B}
C.{Option C}
D.{Option D}
ANSWER::
```

*Figure 16.* Prompt templates of few-shot learning.

# F. More Experimental Results

## F.1. More Results on the Verification of the Sparsity of Interactions

Here are more results on the verification of the sparsity of interactions. As illustrated in Figure 17, the results verify that only a small set of interactions have salient effects, while most of the interactions have negligible effects and can be considered as noise patterns.

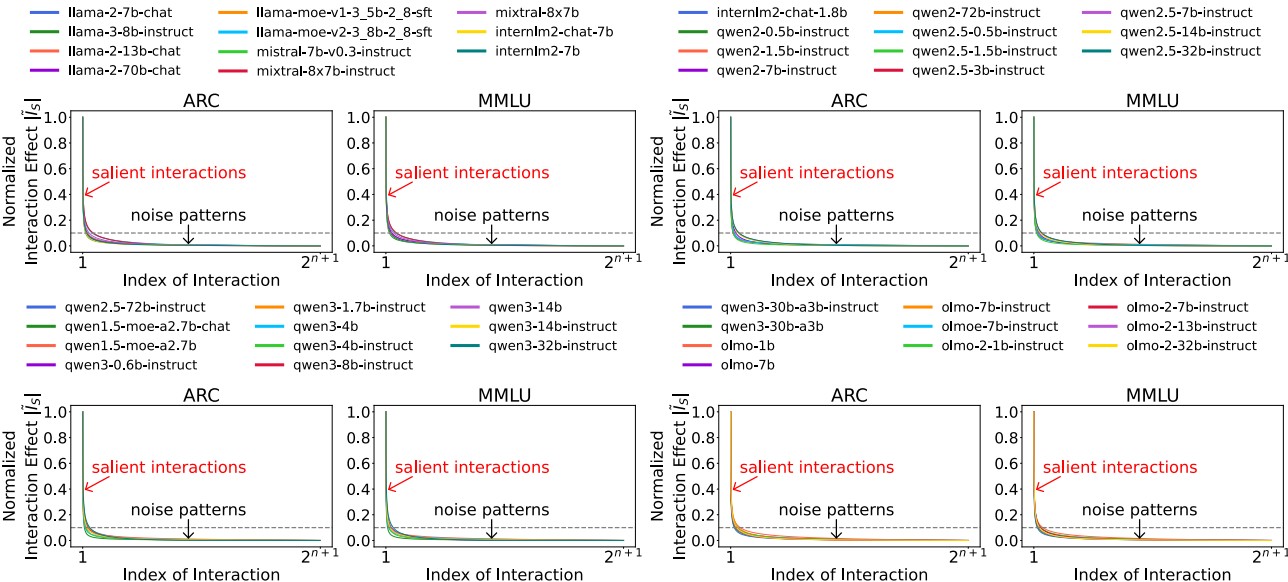

*Figure 17.* Verifying the sparsity of interactions. We show absolute values of normalized interactions in a descending order. LLMs all encode a small number of salient interactions, while most of the interaction effects are negligible.

## F.2. More Results on the Verification of the Sparsity of Interactions

Here are more results on the verification of quality of universal matching. Figure 18 compares the LLM's true output $v(\boldsymbol{x}_T)$ for all masked inputs against the logical model using only the most salient interactions. Even when using just the top 3% or top 5% of all interactions, the matching error is minimal. This empirically demonstrates that the LLM's output can be faithfully approximated by a small, sparse set of salient interactions.

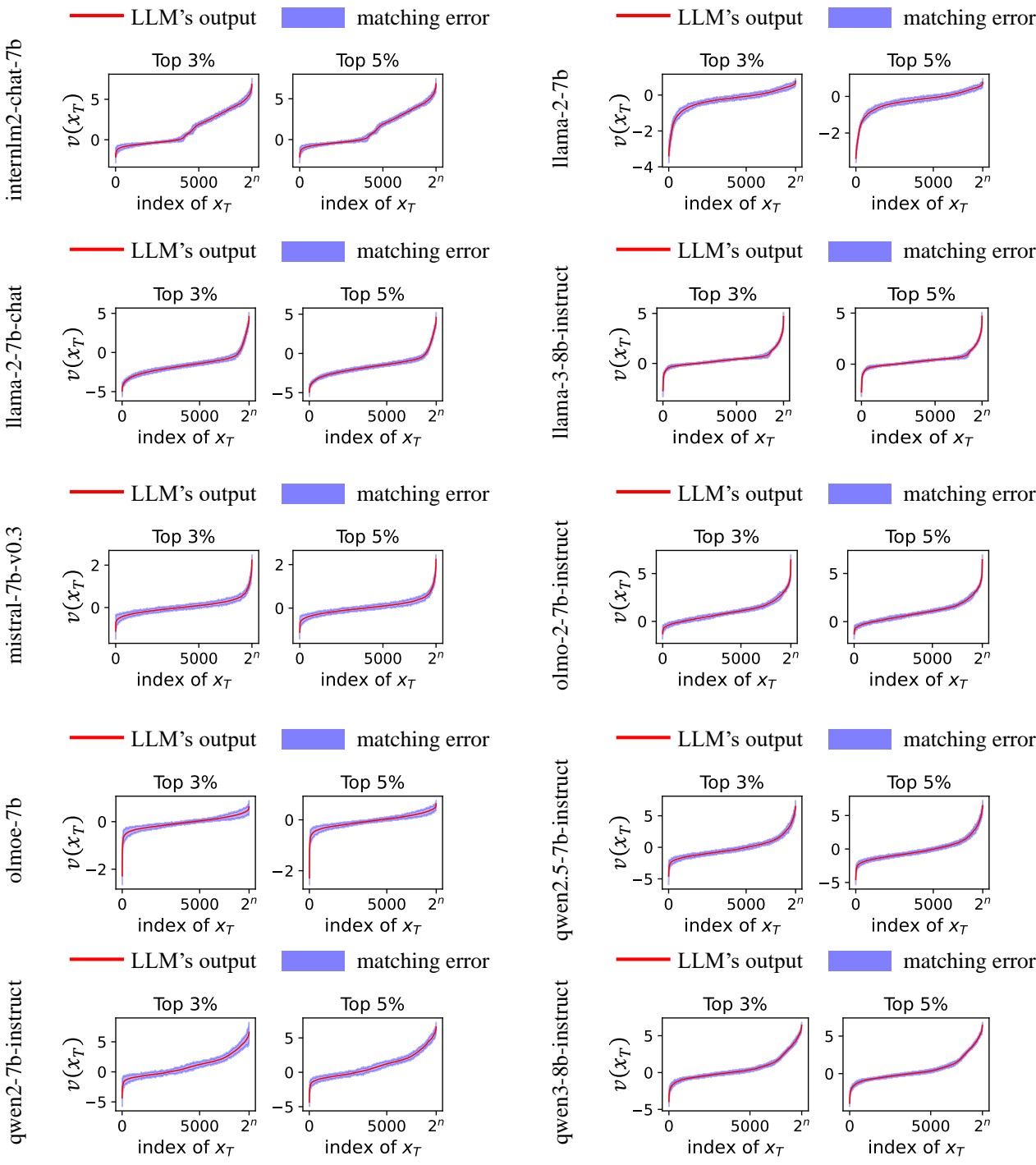

*Figure 18.* Verifying the quality of universal matching for any $2^n$ masked inputs. The red line plots outputs of the LLM in an ascending order.

## F.3. Detailed Case Study

Figure 19 is the detailed case study of how to use our interaction-based analytical tool. It offers preliminary evidence that semantically irrelevant alterations to the prompt template can lead to significant changes in the salient interaction patterns, even when the input and output remains unchanged. This reveals the existence of unstable interactions, which we propose as the underlying cause of prompt sensitivity.

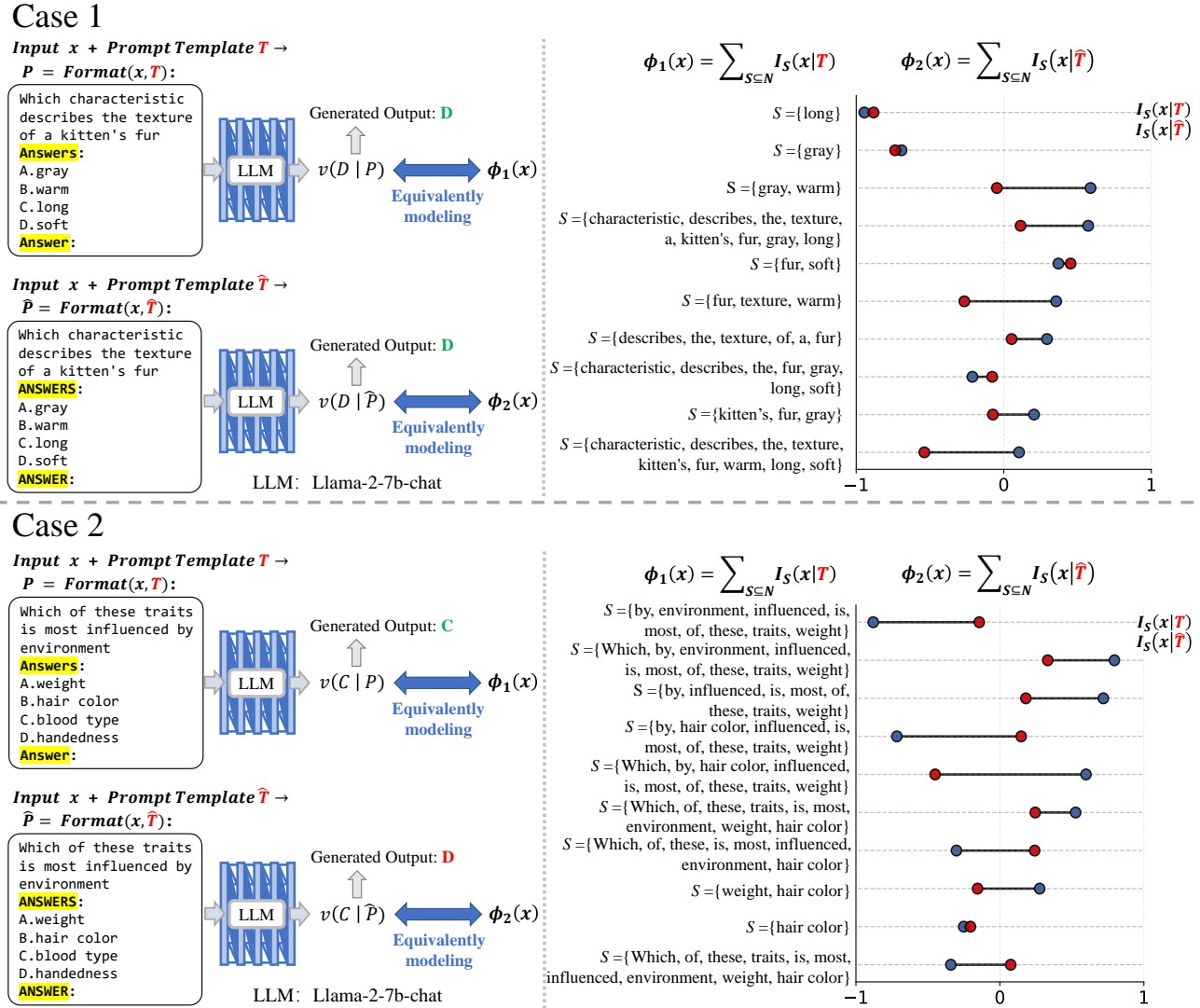

*Figure 19.* A case study of interaction-level analysis revealing latent instability. The same input $x$ is formatted with two semantically identical templates, $T$ and $\hat{T}$, differing only in letter case (e.g.,"Answer" vs. "ANSWER"). Although the LLM generates the same correct output ("D") in both cases, the composition of the interaction-based logical model $\phi(x)$ reveals significant internal divergence. Many interaction effects are highly unstable, changing in either sign or magnitude. This highlights a critical risk of prompt sensitivity that is invisible to output-level

## F.4. More Results on the Prompt Sensitivity of Different Orders

Here are more results on the prompt sensitivity of different orders on the ARC dataset. As illustrated in Figure 20, it shows that the prompt sensitivity of low-order interactions is the lowest, followed by mid-order, while high-order interactions exhibit the highest prompt sensitivity. This indicates that low-order interactions encoded by LLMs are highly stable when faced with subtle changes to prompt templates, *i.e.*, simple interaction patterns are more robust. Conversely, the high sensitivity of high-order interactions reveals that the LLMs' internal representation of complex patterns is highly unstable.

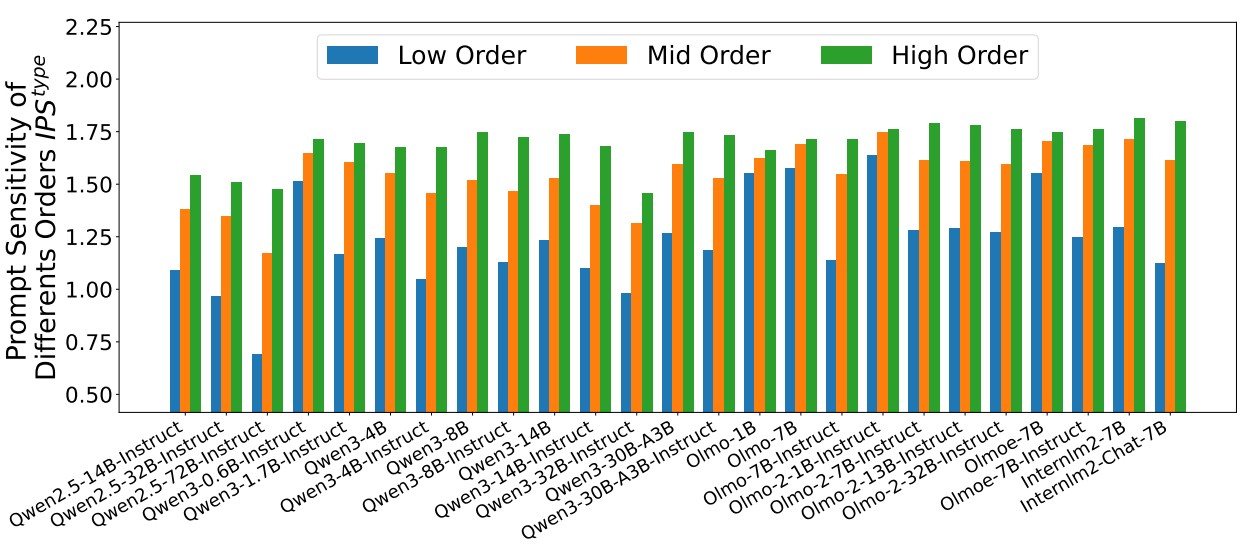

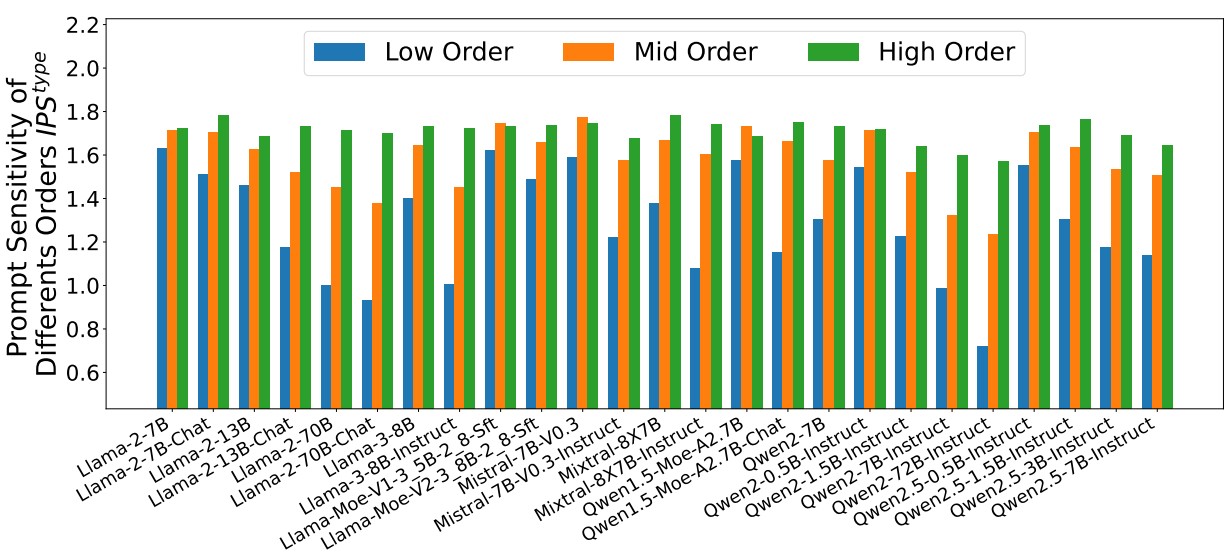

*Figure 20.* A comparison of the prompt sensitivity of three order types. Results show that low-order interactions are the least sensitive, while high-order interactions are the most sensitive.

## F.5. More Results on Relative Change in the Prompt Sensitivity of Low-, Mid-, and High-Order Interactions for Different Factors.

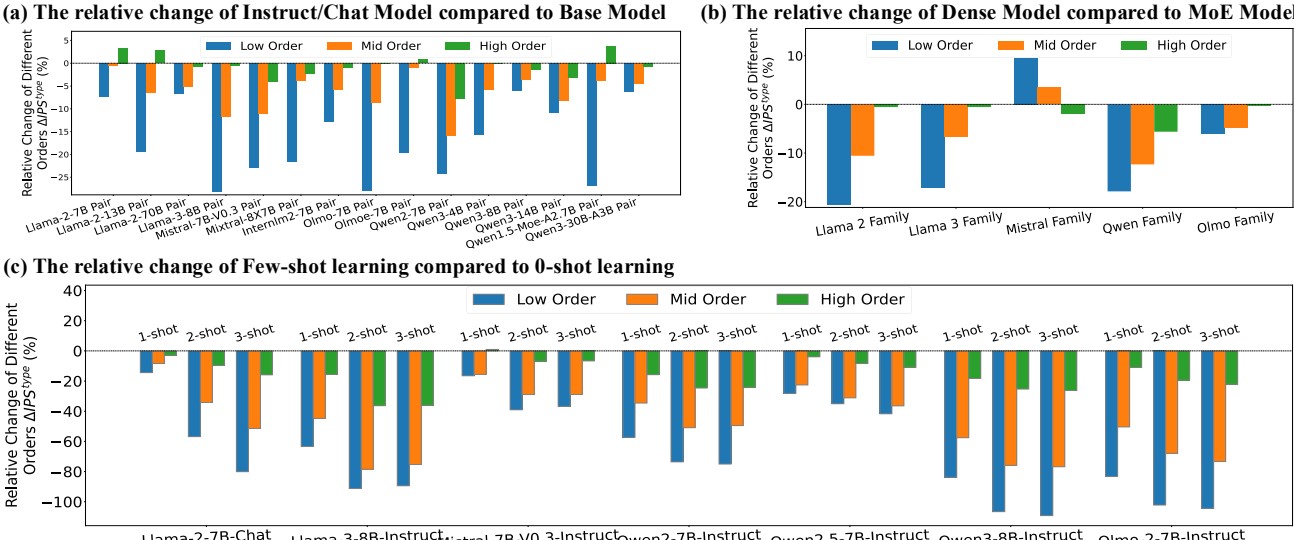

Figure 21. Comparing the relative change in the prompt sensitivity of low-, mid-, and high-order interactions for different factors.

## F.6. More Results on the Prompt Sensitivity of Different Order Types across Different Model Scales.

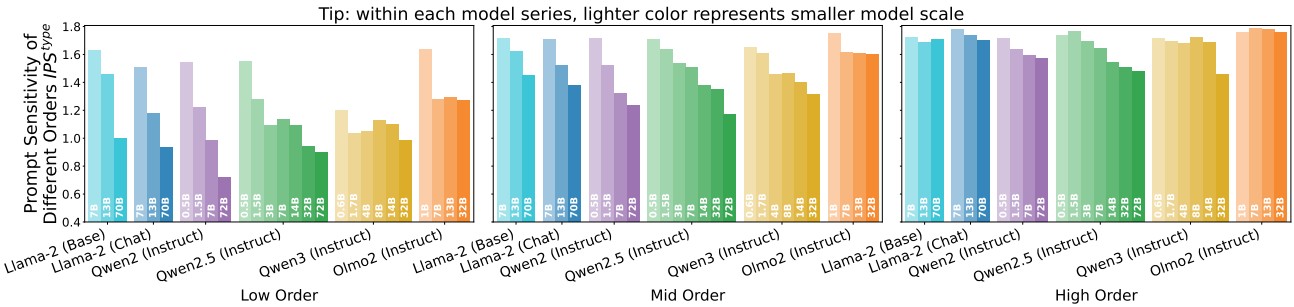

Figure 22. A comparison of prompt sensitivity of different order types across different model scales.

## F.7. More Results on the Prompt Sensitivity of Different Orders for Each Individual LLM when Applying Few-Shot Learning

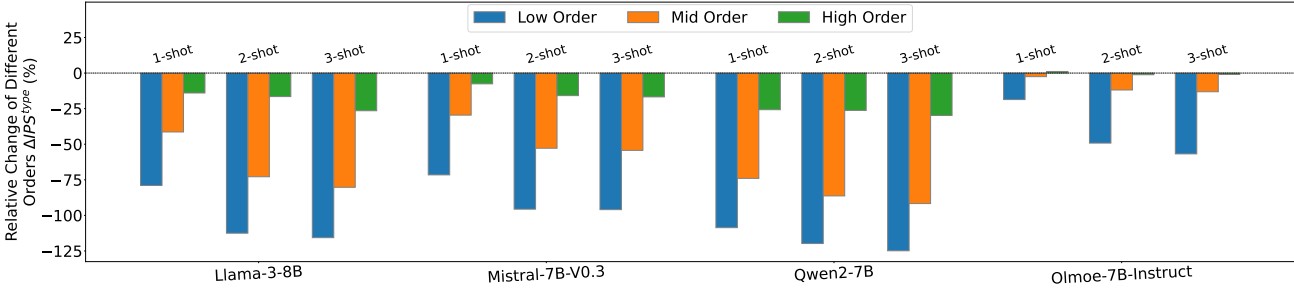

Figure 23. A comparison of prompt sensitivity of low-, mid-, and high-order interactions between 0-shot learning and few-shot learning. Prompt sensitivity at all three order levels shows an clear drop when applying few-shot learning.

## F.8. Hyperparameter Experiments of the threshold $\tau$

To rigorously evaluate the robustness of our Interaction-based Prompt Sensitivity (IPS) metric, we conducted a hyperparameter sweep on the threshold $\tau$.

### F.8.1. DETAILED MODEL RANKINGS UNDER VARYING THRESHOLDS

We aggregated the ranking and scoring consistency across all 16 thresholds using five metrics. As shown in Table 2, the high correlation coefficients and low error rates demonstrate that the IPS metric is highly robust to the choice of $\tau$.

*Table 2.* Summary of consistency metrics across 16 different $\tau$ thresholds (0.05–0.20). The high values in correlation metrics and low RMSE indicate that the relative ranking of model sensitivity remains stable regardless of the specific threshold used.

| Metric | Value | Interpretation |
|---|---|---|
| **Spearman's $\rho$** | 0.9905 | **Rank Correlation:** Measures the average similarity of the overall ranking trends. A value close to 1.0 indicates near-perfect monotonic consistency. |
| **Pearson's $r$** | 0.9957 | **Linearity:** Measures the linear correlation of the raw IPS scores, indicating that the scale of sensitivity shifts linearly across thresholds. |
| **Kendall's $\tau$** | 0.9465 | **Pairwise Consistency:** Indicates the probability that any pair of models maintains their relative order (better/worse) across different thresholds. |
| **RMSE** | 1.72 | **Ranking Stability:** On average, a model's rank fluctuates by only $\pm 1.72$ positions across different threshold settings. |
| **Top-10 Overlap** | 96.7% | **SOTA Stability:** The set of the top-10 most stable models remains 96.7% identical, ensuring reliable identification of the best-performing models. |

To provide a granular view of robustness, Table 3 details the IPS scores across 10 distinct thresholds ranging from $\tau = 0.05$ to $\tau = 0.20$. Models are sorted based on their stability at the baseline threshold $\tau = 0.05$. The data reveals that while absolute scores fluctuate, the relative ranking of model stability remains highly consistent.

*Table 3.* Detailed IPS scores for 50 LLMs across 10 different thresholds. The consistency in color gradients (implied by values) across rows confirms the robustness of the metric.

| Model Name | IPS Score ($\downarrow$) at Threshold $\tau$ | | | | | | | | | |
|---|---|---|---|---|---|---|---|---|---|---|
| | **0.05** | **0.06** | **0.07** | **0.08** | **0.09** | **0.10** | **0.12** | **0.15** | **0.18** | **0.20** |
| Qwen2.5-72B-Instruct | 1.328 | 1.312 | 1.297 | 1.286 | 1.276 | 1.268 | 1.255 | 1.240 | 1.228 | 1.222 |
| Qwen2-72B-Instruct | 1.362 | 1.346 | 1.333 | 1.322 | 1.312 | 1.304 | 1.290 | 1.278 | 1.271 | 1.265 |
| Qwen2-7B-Instruct | 1.411 | 1.400 | 1.391 | 1.383 | 1.376 | 1.368 | 1.357 | 1.341 | 1.327 | 1.318 |
| Qwen3-32B-Instruct | 1.424 | 1.419 | 1.415 | 1.411 | 1.408 | 1.406 | 1.403 | 1.401 | 1.396 | 1.395 |
| Qwen3-14B-Instruct | 1.443 | 1.438 | 1.434 | 1.429 | 1.427 | 1.425 | 1.419 | 1.415 | 1.412 | 1.407 |
| Llama-2-70B-Chat | 1.450 | 1.446 | 1.441 | 1.438 | 1.436 | 1.433 | 1.428 | 1.420 | 1.408 | 1.398 |
| Qwen2.5-32B-Instruct | 1.463 | 1.454 | 1.446 | 1.440 | 1.434 | 1.430 | 1.423 | 1.415 | 1.411 | 1.409 |
| Llama-3-8B-Instruct | 1.486 | 1.483 | 1.480 | 1.477 | 1.475 | 1.473 | 1.467 | 1.460 | 1.447 | 1.436 |
| Qwen2.5-14B-Instruct | 1.490 | 1.481 | 1.473 | 1.465 | 1.459 | 1.454 | 1.446 | 1.436 | 1.430 | 1.425 |
| Qwen3-4B-Instruct | 1.496 | 1.493 | 1.490 | 1.488 | 1.485 | 1.484 | 1.480 | 1.473 | 1.470 | 1.464 |
| Qwen2-1.5B-Instruct | 1.497 | 1.494 | 1.492 | 1.487 | 1.483 | 1.477 | 1.465 | 1.447 | 1.429 | 1.416 |
| Llama-2-70B | 1.505 | 1.504 | 1.504 | 1.505 | 1.505 | 1.507 | 1.509 | 1.516 | 1.523 | 1.527 |
| Qwen3-8B-Instruct | 1.515 | 1.511 | 1.508 | 1.505 | 1.502 | 1.499 | 1.494 | 1.491 | 1.491 | 1.490 |

**Table 3 – continued from previous page**

| Model Name | IPS Score (↓) at Threshold $\tau$ | | | | | | | | | |
|---|---|---|---|---|---|---|---|---|---|---|
| | 0.05 | 0.06 | 0.07 | 0.08 | 0.09 | 0.10 | 0.12 | 0.15 | 0.18 | 0.20 |
| Qwen3-8B | 1.553 | 1.554 | 1.554 | 1.554 | 1.554 | 1.555 | 1.557 | 1.558 | 1.557 | 1.555 |
| Qwen3-4B | 1.555 | 1.562 | 1.566 | 1.570 | 1.573 | 1.575 | 1.578 | 1.579 | 1.574 | 1.572 |
| Llama-2-13B-Chat | 1.559 | 1.555 | 1.551 | 1.547 | 1.544 | 1.541 | 1.535 | 1.528 | 1.520 | 1.514 |
| Qwen3-14B | 1.559 | 1.558 | 1.557 | 1.556 | 1.556 | 1.555 | 1.553 | 1.552 | 1.549 | 1.548 |
| Qwen2-7B | 1.570 | 1.574 | 1.576 | 1.577 | 1.578 | 1.578 | 1.579 | 1.580 | 1.574 | 1.569 |
| Qwen3-30B-A3B-Instruct | 1.575 | 1.575 | 1.576 | 1.577 | 1.579 | 1.580 | 1.582 | 1.583 | 1.583 | 1.582 |
| Qwen2.5-7B-Instruct | 1.580 | 1.575 | 1.572 | 1.569 | 1.567 | 1.565 | 1.564 | 1.562 | 1.563 | 1.563 |
| Llama-2-13B | 1.592 | 1.602 | 1.610 | 1.617 | 1.623 | 1.628 | 1.636 | 1.645 | 1.653 | 1.655 |
| Olmo-1B | 1.601 | 1.610 | 1.617 | 1.623 | 1.628 | 1.632 | 1.639 | 1.647 | 1.652 | 1.655 |
| Olmo-7B-Instruct | 1.604 | 1.599 | 1.595 | 1.592 | 1.589 | 1.587 | 1.583 | 1.579 | 1.578 | 1.576 |
| Llama-3-8B | 1.618 | 1.628 | 1.637 | 1.644 | 1.650 | 1.655 | 1.664 | 1.674 | 1.679 | 1.682 |
| Qwen2.5-3B-Instruct | 1.622 | 1.618 | 1.615 | 1.611 | 1.608 | 1.606 | 1.601 | 1.595 | 1.590 | 1.585 |
| Olmo-2-13B-Instruct | 1.625 | 1.627 | 1.628 | 1.629 | 1.630 | 1.631 | 1.632 | 1.633 | 1.632 | 1.631 |
| Olmo-2-7B-Instruct | 1.626 | 1.627 | 1.627 | 1.628 | 1.629 | 1.629 | 1.629 | 1.627 | 1.628 | 1.626 |
| Olmo-2-32B-Instruct | 1.626 | 1.626 | 1.625 | 1.624 | 1.623 | 1.623 | 1.622 | 1.620 | 1.619 | 1.618 |
| Qwen3-30B-A3B | 1.629 | 1.631 | 1.634 | 1.637 | 1.639 | 1.642 | 1.646 | 1.651 | 1.655 | 1.656 |
| Mistral-7B-v0.3-Instruct | 1.630 | 1.629 | 1.629 | 1.629 | 1.629 | 1.631 | 1.633 | 1.638 | 1.641 | 1.644 |
| InternLM2-Chat-7B | 1.634 | 1.639 | 1.641 | 1.644 | 1.647 | 1.650 | 1.656 | 1.660 | 1.664 | 1.667 |
| Qwen2-0.5B-Instruct | 1.639 | 1.650 | 1.658 | 1.663 | 1.666 | 1.669 | 1.669 | 1.665 | 1.658 | 1.651 |
| Mixtral-8x7B-Instruct | 1.641 | 1.638 | 1.636 | 1.633 | 1.632 | 1.631 | 1.629 | 1.628 | 1.629 | 1.630 |
| Qwen1.5-MoE-A2.7B | 1.645 | 1.658 | 1.668 | 1.675 | 1.682 | 1.687 | 1.696 | 1.705 | 1.712 | 1.717 |
| Qwen1.5-MoE-A2.7B-Chat | 1.648 | 1.654 | 1.657 | 1.660 | 1.663 | 1.665 | 1.667 | 1.668 | 1.664 | 1.660 |
| Qwen3-1.7B-Instruct | 1.648 | 1.645 | 1.642 | 1.641 | 1.640 | 1.639 | 1.639 | 1.637 | 1.638 | 1.637 |
| Qwen2.5-0.5B-Instruct | 1.652 | 1.663 | 1.671 | 1.677 | 1.681 | 1.684 | 1.688 | 1.689 | 1.687 | 1.685 |
| Qwen2.5-1.5B-Instruct | 1.658 | 1.663 | 1.667 | 1.671 | 1.674 | 1.676 | 1.679 | 1.684 | 1.686 | 1.689 |
| Llama-MoE-v2-3_8B-2_8-SFT | 1.665 | 1.669 | 1.673 | 1.676 | 1.678 | 1.681 | 1.685 | 1.691 | 1.696 | 1.698 |
| Llama-2-7B-Chat | 1.670 | 1.677 | 1.683 | 1.687 | 1.690 | 1.691 | 1.692 | 1.693 | 1.691 | 1.689 |
| Olmo-7B | 1.671 | 1.679 | 1.685 | 1.690 | 1.695 | 1.699 | 1.706 | 1.714 | 1.718 | 1.721 |
| Mixtral-8x7B | 1.677 | 1.682 | 1.686 | 1.689 | 1.692 | 1.695 | 1.701 | 1.709 | 1.715 | 1.720 |
| Olmoe-7B | 1.680 | 1.689 | 1.696 | 1.702 | 1.707 | 1.711 | 1.720 | 1.733 | 1.742 | 1.746 |
| Qwen3-0.6B-Instruct | 1.682 | 1.678 | 1.675 | 1.673 | 1.671 | 1.668 | 1.665 | 1.659 | 1.654 | 1.653 |
| Llama-2-7B | 1.682 | 1.692 | 1.699 | 1.705 | 1.711 | 1.716 | 1.724 | 1.733 | 1.740 | 1.745 |
| Olmo-2-1B-Instruct | 1.698 | 1.710 | 1.719 | 1.727 | 1.734 | 1.739 | 1.747 | 1.756 | 1.760 | 1.760 |
| InternLM2-7B | 1.706 | 1.711 | 1.716 | 1.720 | 1.724 | 1.728 | 1.734 | 1.743 | 1.751 | 1.758 |
| Llama-MoE-v1-3_5B-2_8-SFT | 1.715 | 1.720 | 1.724 | 1.728 | 1.732 | 1.735 | 1.743 | 1.753 | 1.761 | 1.765 |
| Olmoe-7B-Instruct | 1.716 | 1.717 | 1.719 | 1.721 | 1.722 | 1.723 | 1.724 | 1.726 | 1.730 | 1.732 |
| Mistral-7B-v0.3 | 1.720 | 1.729 | 1.737 | 1.743 | 1.748 | 1.752 | 1.760 | 1.768 | 1.775 | 1.779 |

F.8.2. VERIFYING THE GENERALIZABILITY OF THE METHODS AND CONCLUSIONS ON DIFFERENT THRESHOLD $\tau$.

In Sections 4.2 and 4.3, we set the threshold $\tau$ to 0.1 to distinguish salient interactions from noise. This threshold directly influences the proportion of interactions classified as salient interactions. A higher $\tau$ value usually generates a smaller set of salient interactions with more significant effects. Li & Zhang (2023) conducted experiments which show that conclusions are not sensitive to the choice of $\tau$. Our choice of $\tau$ is guided by the empirical sparsity of interactions. Figure 2 (a) shows a sharp "elbow" in the distribution of interaction effects, clearly separating a small set of high-magnitude salient interactions from a long tail of near-zero noise interactions. A threshold $\tau$ chosen from the range of 0.05 to 0.15 effectively captures this salient set, satisfying the sparsity assumption without being overly restrictive. To ensure the robustness of our findings, we conducted hyperparameter experiments with $\tau = 0.05$ and $\tau = 0.15$. Our main conclusions remain consistent across different threshold values.

**Here are the results for $\tau = 0.15$.**

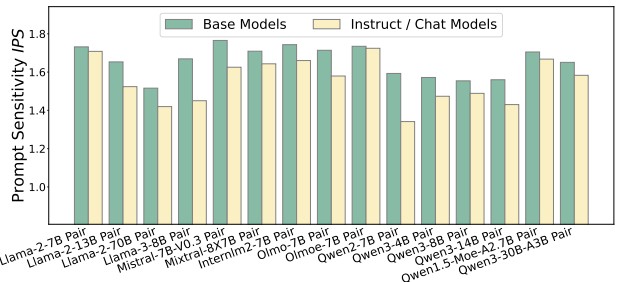

*Figure 24.* A comparison of the prompt sensitivity between instruct/chat models and base models. Results show that instruct/chat models are less sensitive than corresponding base models.

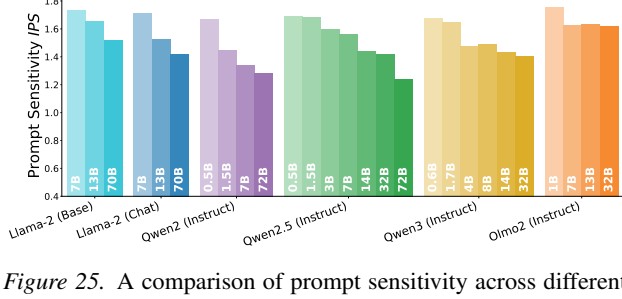

*Figure 25.* A comparison of prompt sensitivity across different model scales. As the model scale increases, the prompt sensitivity within a model series systematically decreases.

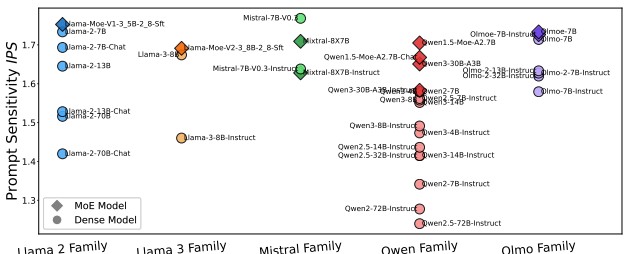

*Figure 26.* A Comparison of prompt sensitivity between MoE models and dense models. Generally, MoE models tend to be more sensitive than dense models in the same model family.

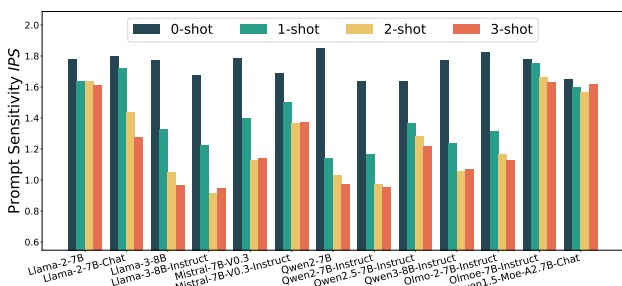

*Figure 27.* A comparison of prompt sensitivity between 0-shot learning and few-shot learning. The drop in prompt sensitivity is substantial from 0-shot to 1-shot.

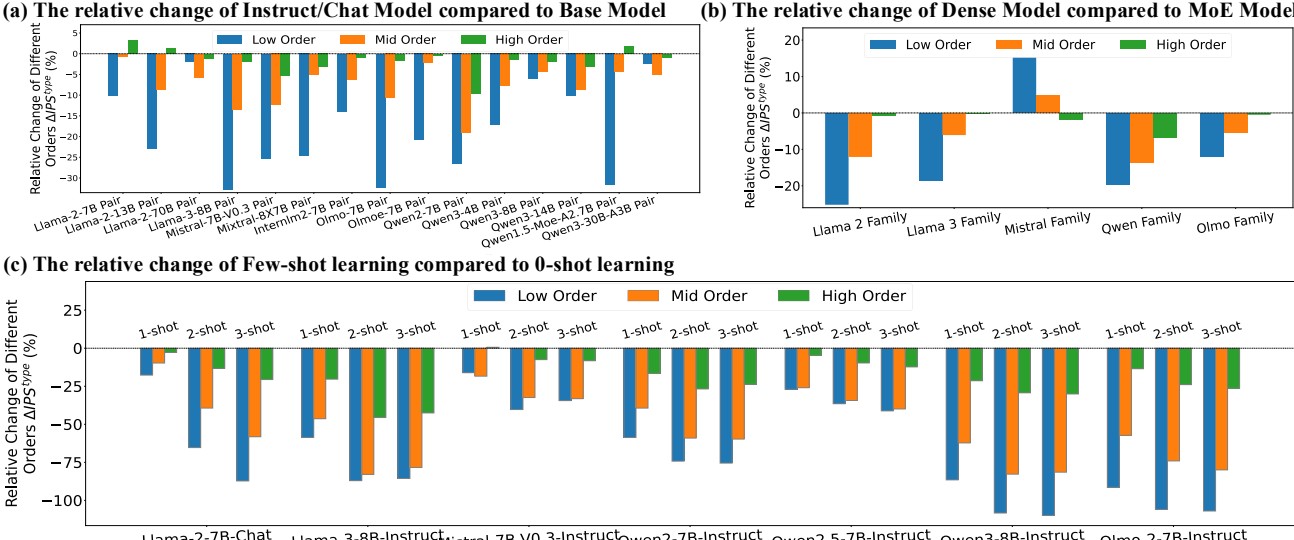

*Figure 28.* Comparing the relative change in the prompt sensitivity of low-, mid-, and high-order interactions for different factors.

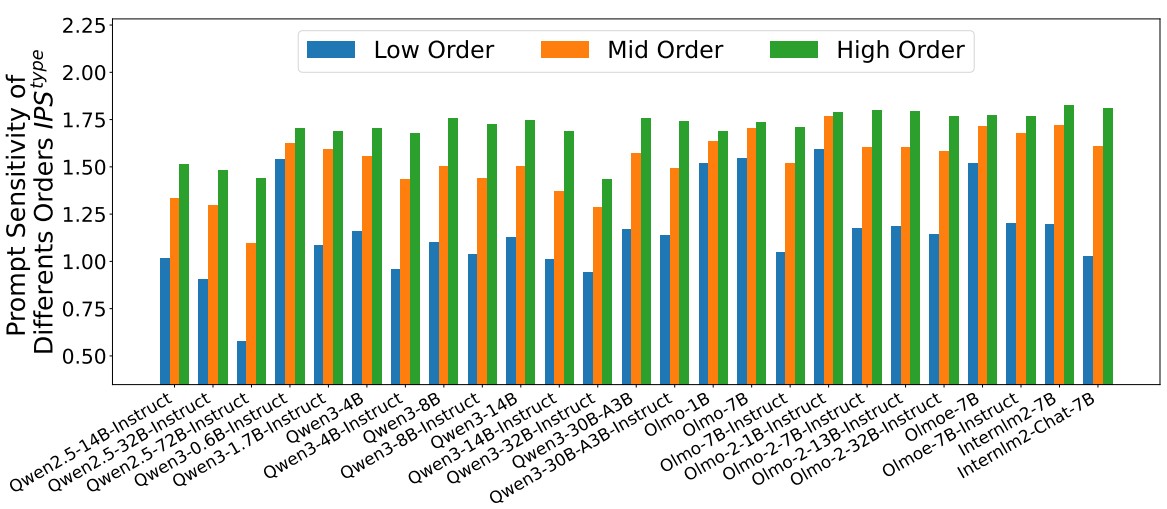

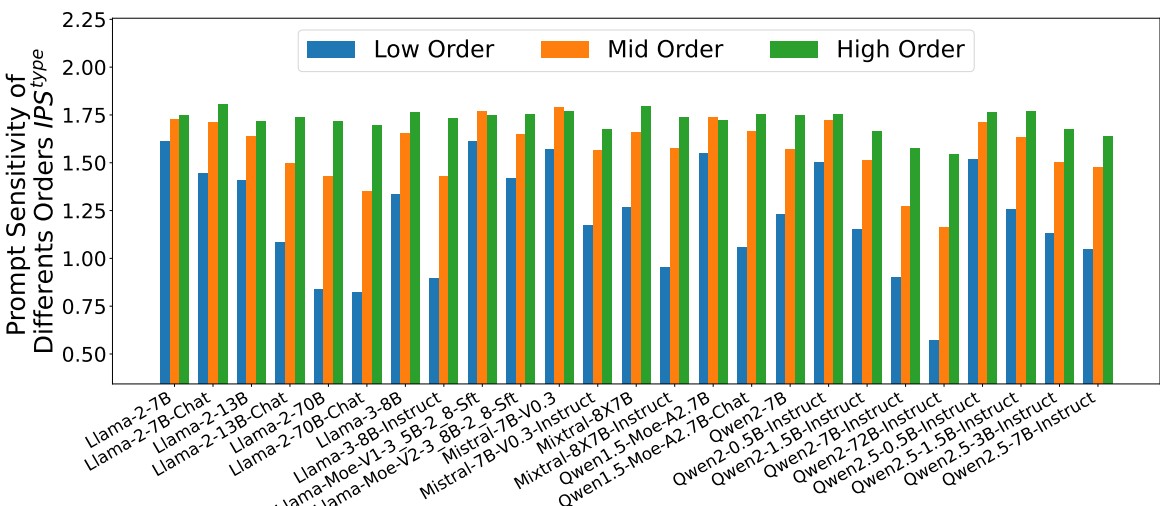

*Figure 29.* A comparison of the prompt sensitivity of three order types. Results show that low-order interactions are the least sensitive, while high-order interactions are the most sensitive.

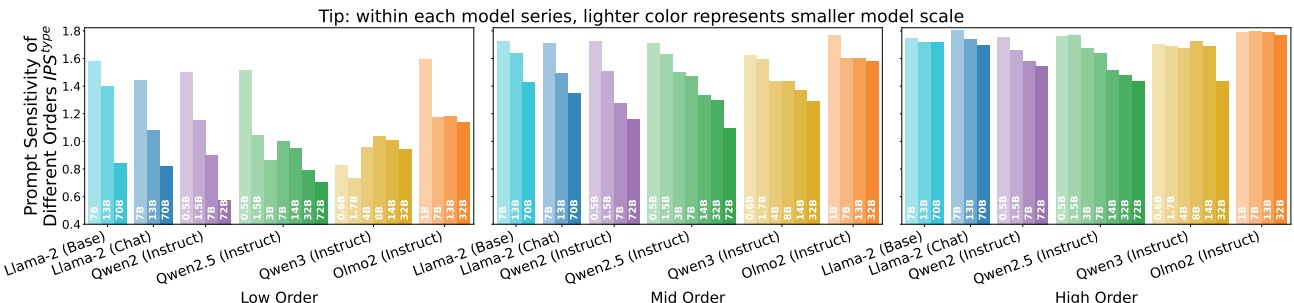

*Figure 30.* A comparison of prompt sensitivity at the order-level across different model scales.

**Here are the results for $\tau = 0.05$.**

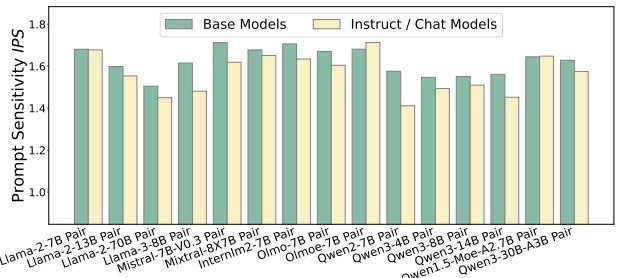

*Figure 31.* A comparison of the prompt sensitivity between instruct/chat models and base models. Results show that instruct/chat models are less sensitive than corresponding base models.

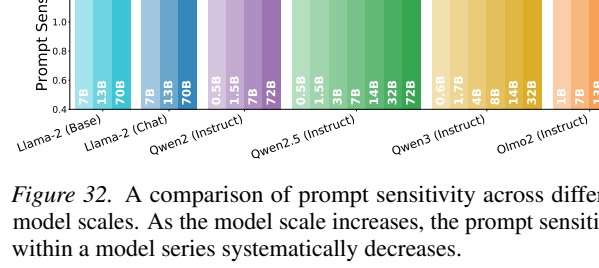

*Figure 32.* A comparison of prompt sensitivity across different model scales. As the model scale increases, the prompt sensitivity within a model series systematically decreases.

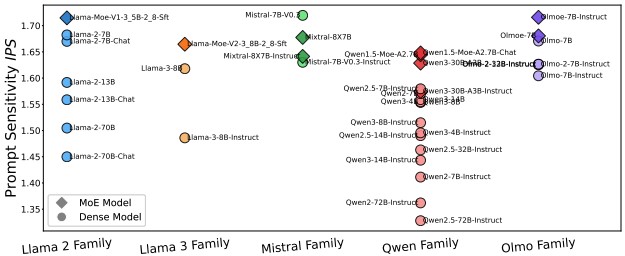

*Figure 33.* A Comparison of prompt sensitivity between MoE models and dense models. Generally, MoE models tend to be more sensitive than dense models in the same model family.

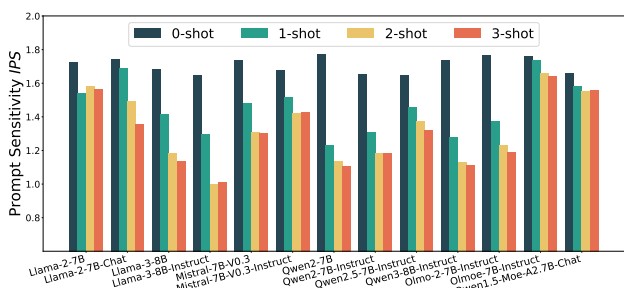

*Figure 34.* A comparison of prompt sensitivity between 0-shot learning and few-shot learning. The drop in prompt sensitivity is substantial from 0-shot to 1-shot.

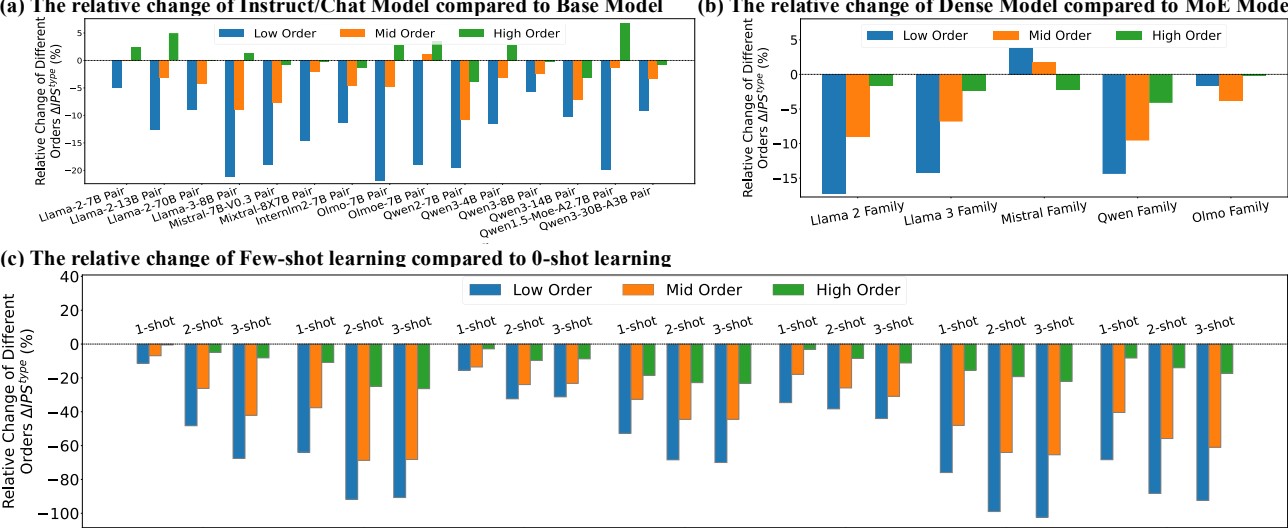

*Figure 35.* Comparing the relative change in the prompt sensitivity of low-, mid-, and high-order interactions for different factors.

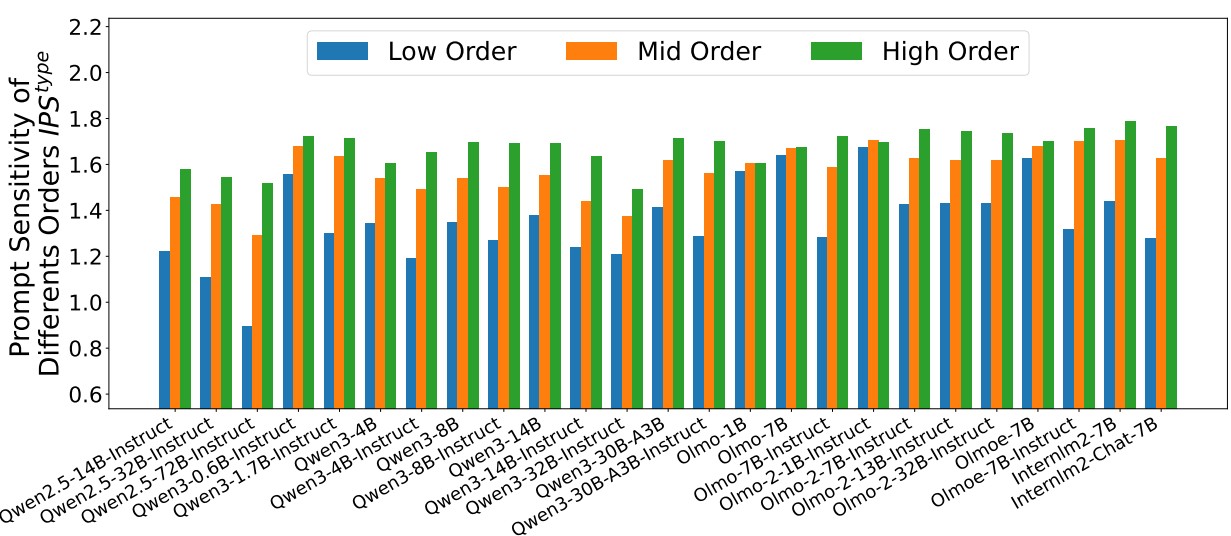

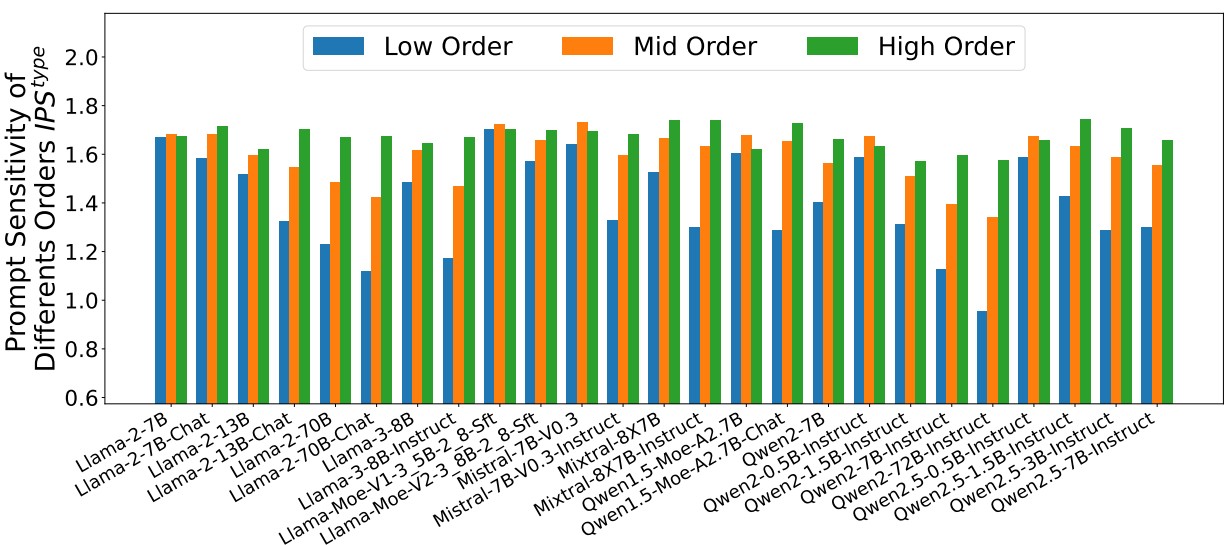

*Figure 36.* A comparison of the prompt sensitivity of three order types. Results show that low-order interactions are the least sensitive, while high-order interactions are the most sensitive.

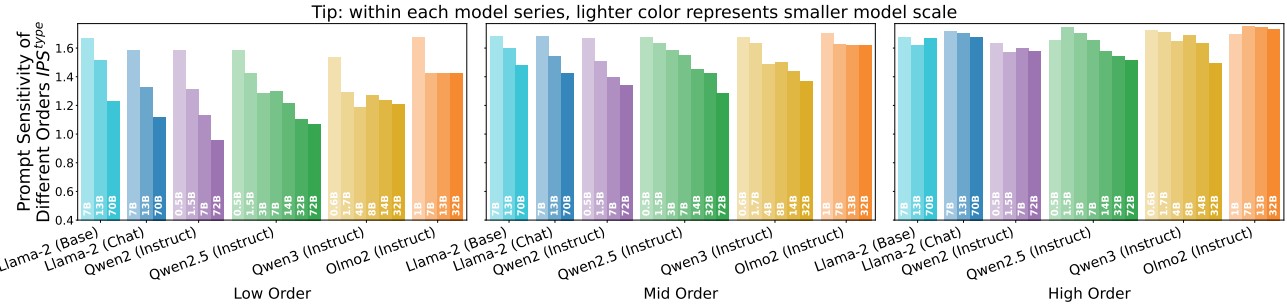

*Figure 37.* A comparison of prompt sensitivity at the order-level across different model scales.

## F.9. Results on MMLU Dataset

Here are the results on the MMLU dataset and $\tau$ is set to 0.1, we can observe the same conclusions on this dataset.

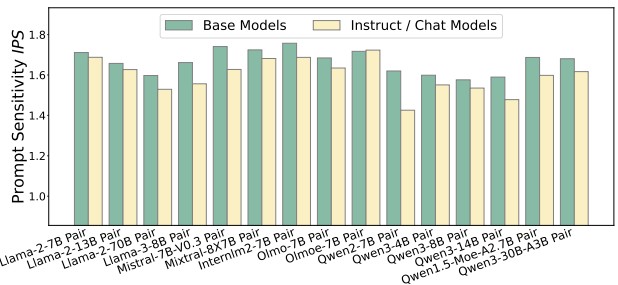

*Figure 38.* A comparison of the prompt sensitivity between instruct/chat models and base models. Results show that instruct/chat models are less sensitive than corresponding base models.

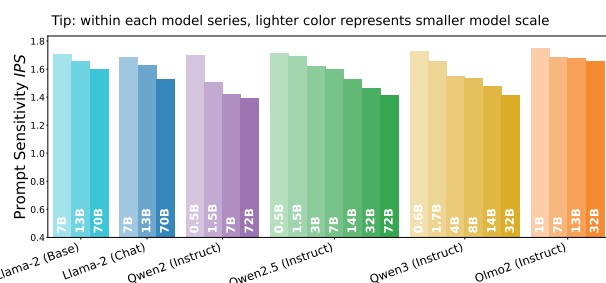

*Figure 39.* A comparison of prompt sensitivity across different model scales. As the model scale increases, the prompt sensitivity within a model series systematically decreases.

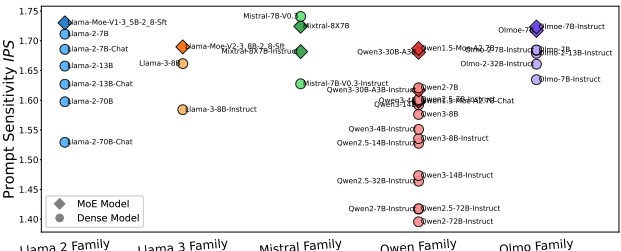

*Figure 40.* A Comparison of prompt sensitivity between MoE models and dense models. Generally, MoE models tend to be more sensitive than dense models in the same model family.

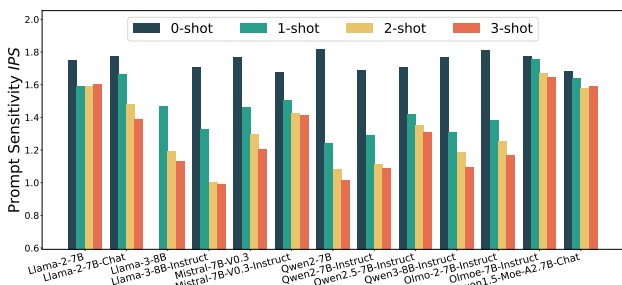

*Figure 41.* A comparison of prompt sensitivity between 0-shot learning and few-shot learning. The drop in prompt sensitivity is substantial from 0-shot to 1-shot.

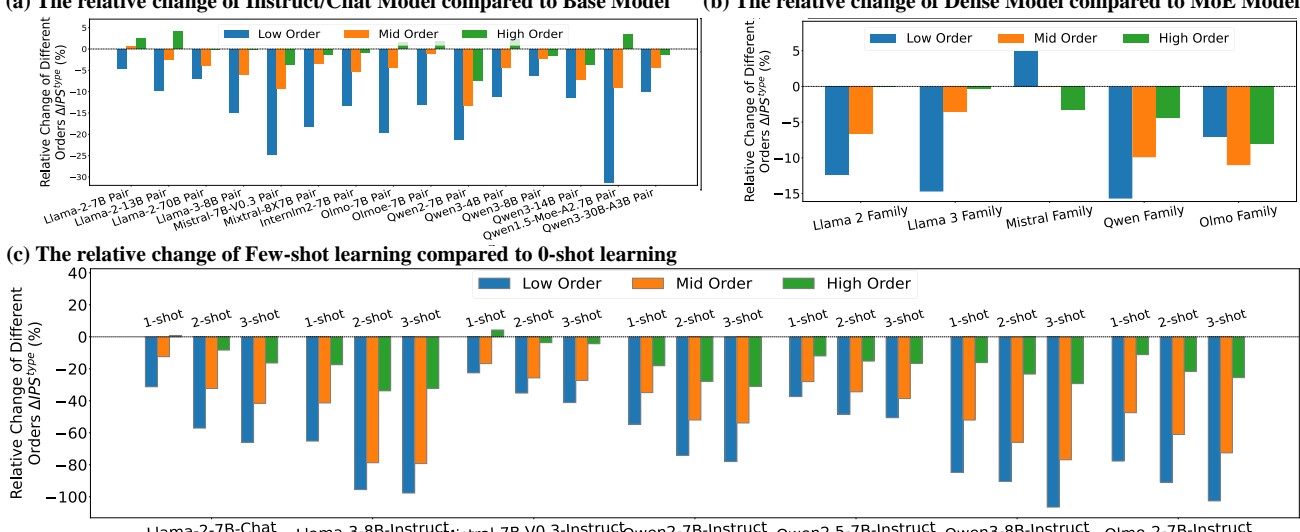

*Figure 42.* Comparing the relative change in the prompt sensitivity of low-, mid-, and high-order interactions for different factors.

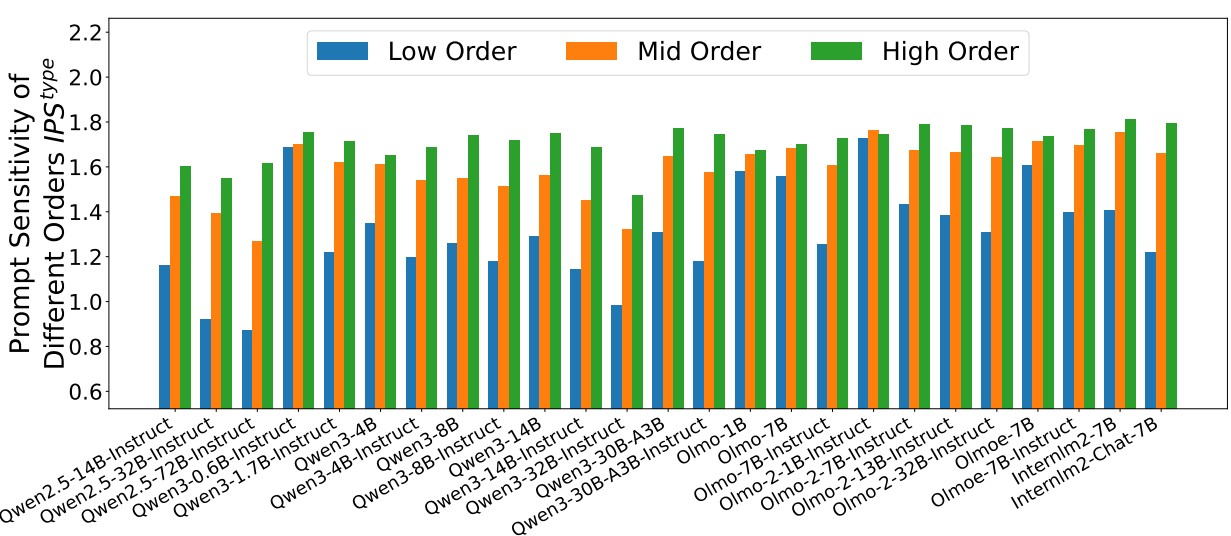

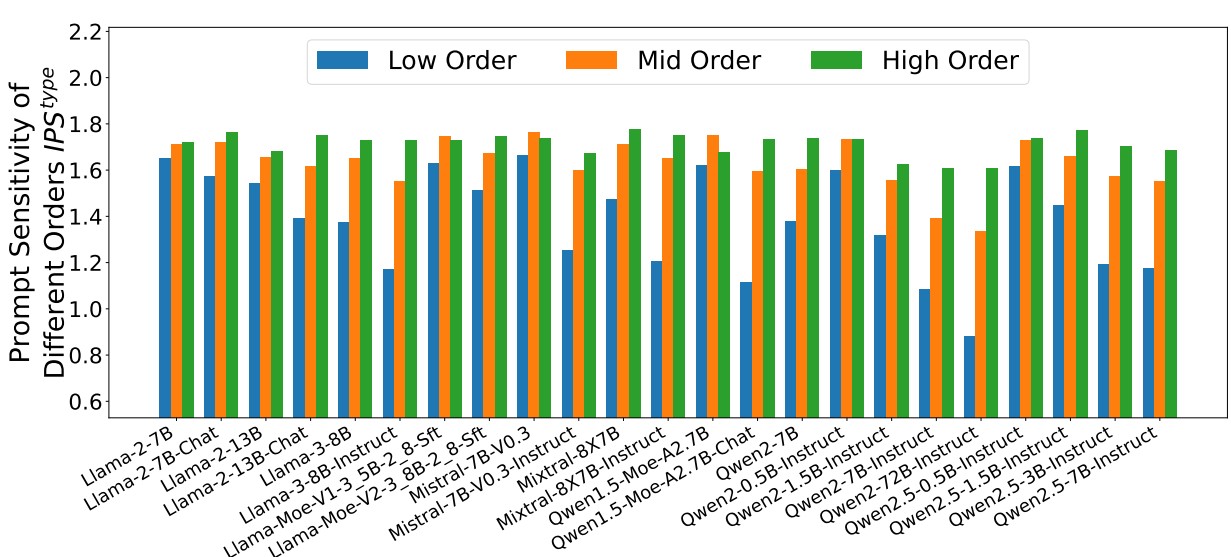

*Figure 43.* A comparison of the prompt sensitivity of three order types. Results show that low-order interactions are the least sensitive, while high-order interactions are the most sensitive.

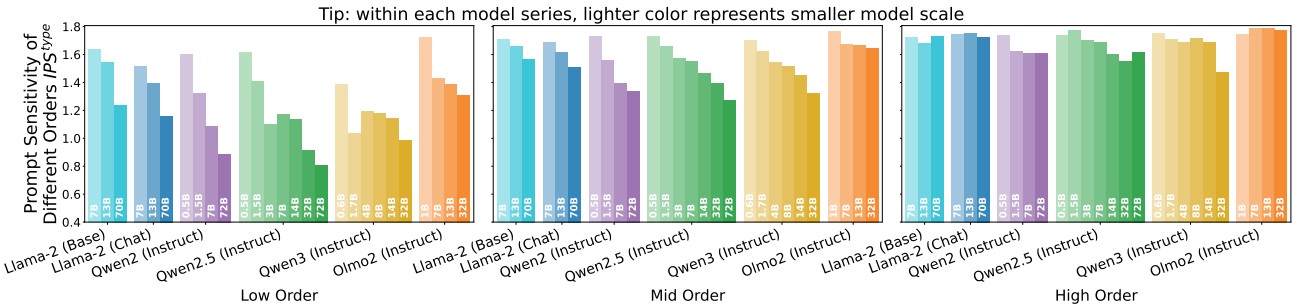

*Figure 44.* A comparison of prompt sensitivity at the order-level across different model scales.

## G. Prompt Sensitivity of altering tokens vs. adding tokens

Disaggregating prompt sensitivity by the type of perturbation offers deeper insights into model behavior. Following this direction, we conduct a fine-grained decomposition of our experimental results, comparing prompt sensitivity of two distinct categories to prompt alterations:

1. **Altering tokens:** This involves modifying the capitalization of words, such as from `"Answer"` to `"ANSWER"`.

2. **Adding tokens:** This involves adding symbolic components, for instance, changing a colon from `":"` to `"::"`.

| Dense Model | Altering Tokens | Adding Tokens | Difference |
|---|---|---|---|
| internlm2-chat-7b | 1.673 | **1.676** | +0.003 |
| llama-2-13b-chat | **1.520** | 1.433 | -0.087 |
| llama-2-7b-chat | 1.633 | **1.710** | +0.077 |
| llama-3-8b-instruct | 1.450 | **1.467** | +0.017 |
| mistral-7b-v0.3-instruct | 1.538 | **1.635** | +0.097 |
| olmo-2-7b-instruct | 1.618 | **1.655** | +0.037 |
| olmo-7b-instruct | 1.462 | **1.496** | +0.034 |
| qwen2-7b-instruct | 1.366 | **1.404** | +0.038 |
| qwen2.5-7b-instruct | 1.558 | **1.567** | +0.009 |
| qwen3-8b-instruct | 1.505 | **1.550** | +0.045 |

*Table 4.* IPS Scores of Dense Models on Different Perturbation Types. Higher scores indicate greater sensitivity. The more sensitive perturbation type for each model is highlighted in bold.

| MoE Model | Altering Tokens | Adding Tokens | Difference |
|---|---|---|---|
| llama-moe-v1-3.5b-sft | **1.736** | 1.724 | -0.012 |
| llama-moe-v2-3.8b-sft | **1.698** | 1.655 | -0.043 |
| mixtral-8x7b-instruct | **1.658** | 1.617 | -0.041 |
| olmoe-7b-instruct | **1.725** | 1.699 | -0.026 |
| qwen1.5-moe-a2.7b-chat | **1.669** | 1.634 | -0.035 |
| qwen3-30b-a3b-instruct | 1.608 | **1.638** | +0.030 |

*Table 5.* IPS Scores of MoE Models on Different Perturbation Types. The more sensitive perturbation type for each model is highlighted in bold.

Our results highlight a clear architectural divide: **(1) Dense models are more sensitive to adding tokens.** As shown in Table 4, 9 out of the 10 analyzed dense models exhibit greater sensitivity to the "adding tokens" category. This suggests a strong, consistent trend where changes to the template's structure have a more pronounced impact on the internal interactions of dense architectures. **(2) MoE models are more sensitive to altering tokens.** In stark contrast, Table 5 shows that 5 out of the 6 MoE models are more sensitive to "altering tokens". This consistent pattern suggests that MoE architectures are more susceptible to variations in the change of word capitalization.

## H. Comparison between IPS and Other Metrics

To demonstrate the unique value of our interaction-based approach, we compare the Interaction-based Prompt Sensitivity (IPS) against standard coarse-grained metrics derived from internal representations. Specifically, we measure the **Cosine Similarity** and $L_2$ **Distance** of the final-layer hidden states under prompt perturbations. While these metrics are commonly used to assess representation robustness, our analysis reveals that they fail to capture the nuanced mechanisms of prompt sensitivity in LLMs.

### H.1. Empirical Inconsistency of Representation-based Metrics

We re-evaluate **Factor 1 (Base vs. Instruct/Chat models)** using these representation-based metrics. The results, summarized in Table 6, demonstrate a significant lack of consistency compared to the robust trends observed via IPS.

As shown in Table 6, neither Cosine Similarity nor $L_2$ Distance provides a reliable proxy for prompt sensitivity:

*Table 6.* Comparison of stability metrics (Cosine Similarity and $L_2$ Distance) for Base vs. Instruct/Chat models. Unlike IPS, these metrics fail to show a consistent trend regarding the impact of Supervised Fine-Tuning.

| Model Family | Base Model | | Instruct/Chat Model | |
|---|---|---|---|---|
| | Cosine (↑) | $L_2$ Dist. (↓) | Cosine (↑) | $L_2$ Dist. (↓) |
| Llama-2-7B | 0.770 | **75.12** | **0.781** | 66.51 |
| Llama-2-13B | **0.942** | **17.35** | 0.927 | 19.69 |
| Llama-2-70B | **0.911** | **33.65** | 0.795 | 43.14 |
| Llama-3-8B | 0.856 | 79.77 | **0.880** | **73.56** |
| Mistral-7B-V0.3 | 0.811 | 207.89 | **0.843** | **154.43** |
| Mixtral-8x7B | **0.965** | 138.71 | 0.875 | **136.03** |
| InternLM2-7B | 0.898 | 142.51 | **0.987** | **75.83** |
| Olmo-7B | **0.793** | **36.88** | 0.661 | 48.33 |
| Olmoe-7B | **0.672** | **56.67** | 0.510 | 85.73 |
| Qwen2-7B | 0.864 | 138.73 | **0.888** | **136.49** |
| Qwen3-4B | **0.914** | **60.13** | 0.779 | 73.82 |
| Qwen3-8B | **0.995** | **33.95** | 0.962 | 41.42 |
| Qwen3-14B | **0.947** | 67.85 | 0.932 | **51.97** |
| Qwen1.5-MoE-A2.7B | 0.765 | 160.31 | **0.911** | **86.98** |
| Qwen3-30B-A3B | 0.864 | 81.13 | **0.932** | **39.04** |

- **Contradictions between metrics:** For models like *Mixtral-8x7B*, the two metrics contradict each other—Cosine Similarity suggests the Base model is more stable, while $L_2$ Distance favors the Instruct model.

- **Inconsistency with established trends:** While our IPS analysis (and general consensus) identifies Instruct/Chat models as more robust to prompt variations, representation metrics frequently suggest the opposite. For instance, in the *Llama-2-13B* and *Qwen3-8B* pairs, the Base models exhibit higher cosine similarity and lower $L_2$ distance than their Instruct counterparts.

- **Random fluctuations:** There is no discernible pattern across model families. For *Llama-2-7B*, the Chat version appears more stable via Cosine Similarity but less stable via $L_2$ Distance.

These contradictions indicate that global measures of hidden state changes are too coarse to serve as accurate indicators of the model's functional sensitivity.

### H.2. Superiority of the Interaction-based Framework

The empirical limitations of representation-based metrics highlight the theoretical advantages of our proposed framework. The superiority of IPS stems from two fundamental differences:

**1. Explanability ("Why" vs. "What"):**   Hidden state similarity merely measures *what* has changed—the magnitude or direction of the aggregate internal representation vector. It treats the model as a black box regarding the reasoning process. In contrast, our interaction framework explains *why* the output fluctuates. By decomposing predictions into interactions, we can pinpoint specific combinations of input tokens (inference patterns) that become unstable. This fine-grained insight allows us to distinguish between benign representation shifts and those that disrupt the model's logical coherence.

**2. Faithfulness to the Output:**   Representation metrics lack a direct mathematical link to the final prediction. A small shift in Euclidean distance can sometimes lead to a flipped prediction, while a large shift might not. Conversely, our method is grounded in the **Universal Matching Property** (Theorem 1). This theorem guarantees that the sum of all interactions perfectly reconstructs the LLM's output score. Consequently, IPS provides a faithful evaluation of the decision-making logic, ensuring that the measured sensitivity directly reflects the instability in the model's actual predictive mechanism.

# I. Detailed Discussion on the Impact of Model Architecture

We notice that in Figure 7, the behavior of Mistral family is different from the other family: the dense models are more sensitive than the MoE models at low-order and mid-order level. We attribute this to the number of activated parameters, extending our finding from Factor 2 that larger models are less sensitive. While most MoE models (e.g., in Llama and Qwen families) are more sensitive due to fewer activated parameters, the Mistral case is reversed: Mixtral-8x7B activates more parameters than its dense counterpart Mistral-7B-V0.3 (13B vs. 7B), resulting in lower prompt sensitivity.

To more rigorously isolate the influence of model architecture on the prompt sensitivity from model size or other factors, we conduct a controlled variable analysis. We select specific pairs from the Qwen and OLMo families that share the most similar model scales (*i.e.*, activated parameters) and analogous training paradigms (*i.e.*, base vs. base, instruct/chat vs. instruct/chat). This targeted comparison enables us to minimize confounding factors and focus directly on the architectural impact.

Results in Table 7 consistently show that MoE models exhibit higher prompt sensitivity than dense models. This suggests that the increased prompt sensitivity of MoE architectures is not merely a consequence of smaller number of activated parameters. Instead, it further strengthens our conclusion that the MoE architecture inherently increases the prompt sensitivity of LLMs.

*Table 7.* Controlled variable analysis of prompt sensitivity between MoE and dense models.

| Model Name | Architecture | Act. Params. | Type | IPS (ARC) $\downarrow$ | IPS (MMLU) $\downarrow$ |
|---|---|---|---|---|---|
| Qwen1.5-moe-a2.7b-chat | MoE | 2.7B | Chat | 1.665 | 1.640 |
| Qwen2.5-3b-instruct | Dense | 3B | Instruct | **1.606** | **1.625** |
| Olmoe-7b | MoE | 1B | Base | 1.714 | 1.717 |
| Olmo-1b | Dense | 1B | Base | **1.632** | **1.658** |
| Qwen3-30b-a3b | MoE | 3B | Base | 1.642 | 1.680 |
| Qwen3-4b | Dense | 4B | Base | **1.568** | **1.599** |
| Qwen3-30b-a3b-instruct | MoE | 3B | Instruct | 1.580 | 1.617 |
| Qwen3-4b-instruct | Dense | 4B | Instruct | **1.483** | **1.551** |

# J. Solutions for reducing the computational cost of the method

The limitation of our current study lies in the computational cost of the interaction framework. The method's complexity scales exponentially with the number of input variables $n$, as it requires evaluating $2^n$ masked inputs. However, applying this method to very long text inputs would demand a high computational load.

Future work can address this scalability challenge through several promising avenues. These strategies aim to reduce the effective number of input variables without fundamentally changing the faithfulness of the analysis:

(1) **Selective Input Variable Analysis.** One approach is to analyze only a subset of informative input variables (*i.e.*, words) while treating uninformative ones (e.g., stop words) as fixed background context. Previous research has demonstrated that this selection does not significantly impair the faithfulness of the interaction framework (Chen et al., 2024).

(2) **Phrase-level Aggregation.** Instead of analyzing individual words, we can operate at a coarser perspective by merging related words into combined phrasal units. This reduces the total number of input variables while preserving key semantic meaning.

For methods (1) and (2), we have put them into practice. In our experiments on long-form open-ended questions, we apply these two methods to effectively control the number of input variables. The specific selection strategies are detailed in Appendix K and Appendix L. It demonstrates that these techniques can substantially reduce computational complexity without affecting the key conclusions.

(3) **Approximation Methods.** In addition to selection strategies, techniques specifically designed for efficiently computing sparse interactions to bypass exhaustive $O(2^n)$ evaluations (Kang et al., 2025; Butler et al., 2026), represent a clear path for reducing computational cost.

These strategies represent promising directions for extending the powerful capabilities of interaction-based analysis to a wider range of long-text NLP tasks.

# K. Details of the Experiments on Open-Ended Generation Tasks

## K.1. Experimental Setup

This section illustrates the detailed setup of open-ended question-answering tasks, which more closely resemble real-world user scenarios. We employed the **Databricks Dolly-15k** dataset, an open-source collection of instruction-following records. This dataset spans multiple behavioral categories as defined in the InstructGPT paper (Ouyang et al., 2022), including brainstorming, classification, closed QA, open QA, and summarization, providing a diverse and realistic dataset for evaluating model robustness. We test the results mainly on the Llama-2 family.

## K.2. Selection of Input Variables

Given that the text length in open-ended instructions far exceeds that of MCQ tasks, a direct analysis of all words would lead to exponential computational costs. To address this challenge, we applied the optimization strategies discussed in our limitations section: **(1) Selective Input Variable Analysis** and **(2) Phrase-level Aggregation**.

Our approach is guided by a systematic procedure to choose a fixed number of key input variables from the full input. Specifically, for each input sentence, we select meaningful words or phrases to construct the set of input variables $N$. A word is considered "meaningful" if it is not an NLTK (Bird, 2006) stop word or a punctuation mark. The remaining parts of the text, such as generic instruction templates (e.g., "Below is an instruction...") and stop words, are treated as fixed background context. During the interaction analysis, only the variables within the set $N$ are masked.

For a concrete example, consider the following prompt:

```
Below is an instruction that describes a task.  Write a response that
appropriately completes the request.

### Instruction:
Identify from the following list characters from The X-Files who are bald or
balding:  Walter Skinner, John Fitzgerald Byers, Dana Scully, Melvin Frohike,
Darius Michaud, Peter Watts, Conrad Strughold, Queequeg

### Response:
```

**Selection Process for Input Variables:**

1. **Method (1) Application:** We designate the generic instruction template and functional stop words (e.g., "from", "the", "who", "are") as background context, excluding them from the input variable set.

2. **Method (2) Application:** We aggregate words forming core semantic concepts into single phrasal units, such as the key entity `"Walter Skinner"` and the critical condition `"bald or balding"`.

**Final Input Variables:**

```
[
  "Identify", "following list", "characters","X-Files",
  "bald or balding","Walter Skinner", "John Fitzgerald Byers",
  "Dana Scully","Melvin Frohike", "Darius Michaud",
  "Peter Watts","Conrad Strughold", "Queequeg"
]
```

## K.3. Experimental Results

By applying the aforementioned input variable selection strategies (Methods 1 and 2) in our experiments on the Dolly dataset, we successfully managed the analytical complexity for each long-text input, leading to a substantial reduction in computational cost.

Crucially, the experimental outcomes derived from open-ended questions and the optimized setup remained **highly consistent** with the main conclusions drawn from our MCQ-based experiments.

Here are the results and $\tau$ is set to 0.1, we can observe the same conclusions in this experiment.

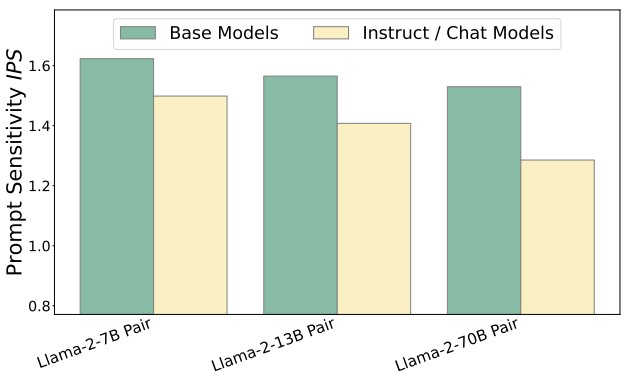

Figure 45. A comparison of the prompt sensitivity between instruct/chat models and base models. Results show that instruct/chat models are less sensitive than corresponding base models.

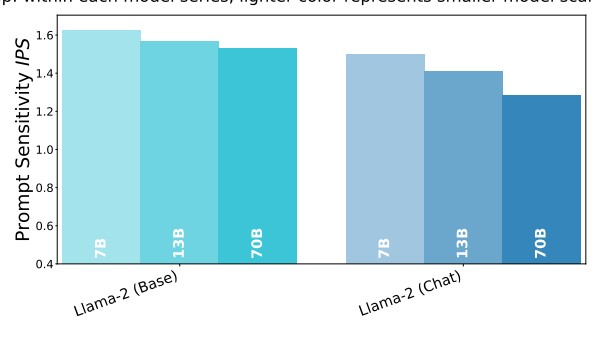

Figure 46. A comparison of prompt sensitivity across different model scales. As the model scale increases, the prompt sensitivity within a model series systematically decreases.

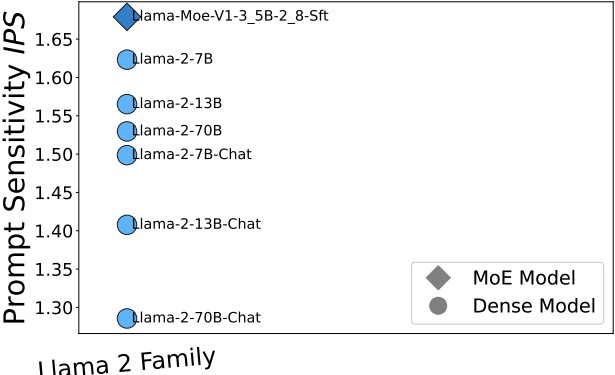

Figure 47. A Comparison of prompt sensitivity between MoE models and dense models. Generally, MoE models tend to be more sensitive than dense models in the same model family.

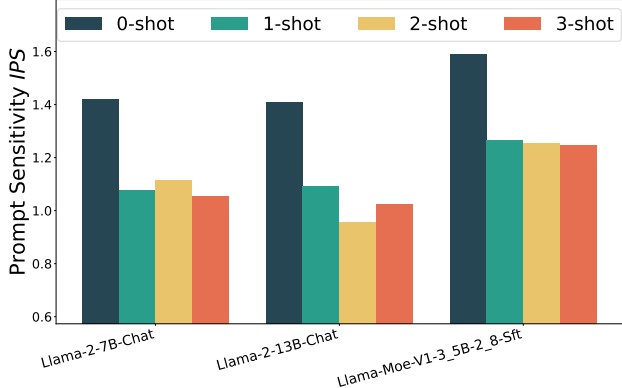

Figure 48. A comparison of prompt sensitivity between 0-shot learning and few-shot learning. The drop in prompt sensitivity is substantial from 0-shot to 1-shot.

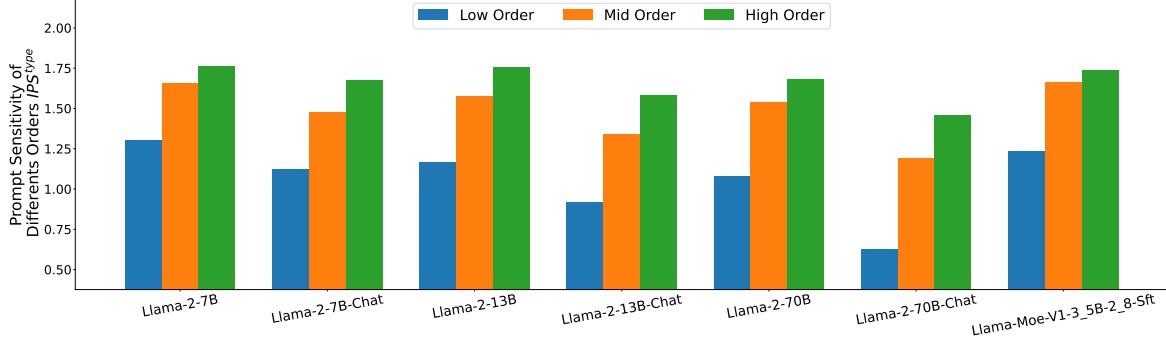

Figure 49. A comparison of the prompt sensitivity of three order types. Results show that low-order interactions are the least sensitive, while high-order interactions are the most sensitive.

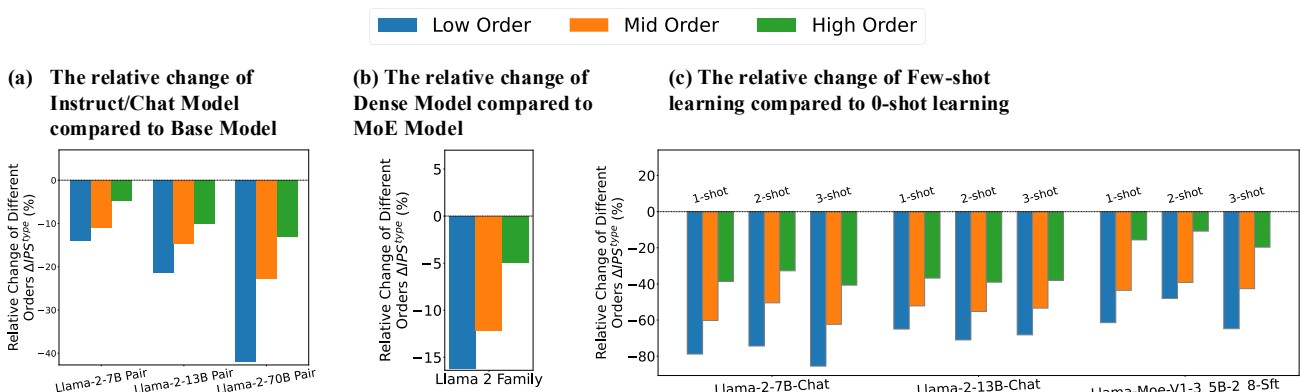

*Figure 50.* Comparing the relative change in the prompt sensitivity of low-, mid-, and high-order interactions for different factors.

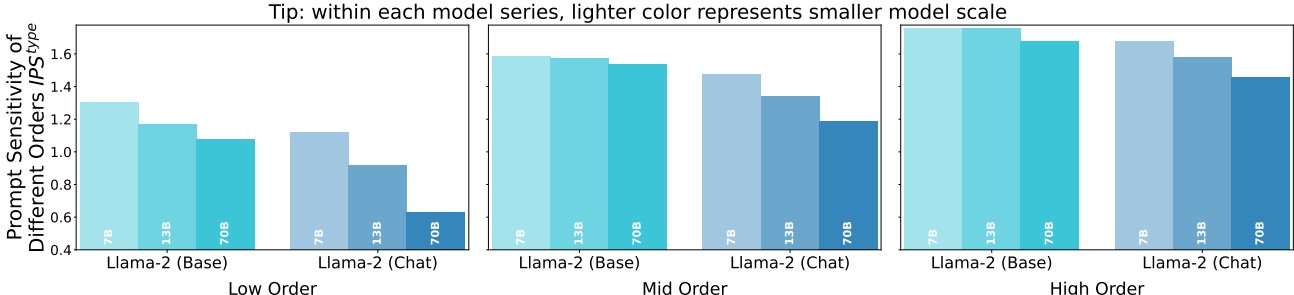

*Figure 51.* A comparison of prompt sensitivity at the order-level across different model scales.

# L. Details of the Experiments Beyond Prompt Template Modifications

## L.1. Experimental Setup

To assess the generalizability of our findings beyond superficial template modifications, we conducted an additional experiment on the **Dolly-15k** dataset. This setup introduces more realistic and complex prompt perturbations that better reflect authentic user interactions, specifically **semantic paraphrasing** and **instruction reordering**. We employed these techniques to test the robustness of our conclusions under more challenging conditions. An illustrative example of a semantic paraphrase used in our experiment is shown below:

An illustrative example of the full prompt structure and the applied perturbations is shown below.

**Original Prompt:**

```
Below is an instruction that describes a task. Write a
response that appropriately completes the request.

### Instruction:
Identify from the following list characters from The X-Files
who are bald or balding: Walter Skinner, John Fitzgerald
Byers, Dana Scully, Melvin Frohike, Darius Michaud,
Peter Watts, Conrad Strughold, Queequeg

### Response:
```

**Perturbation 1: Semantic Paraphrase:**

```
Below is an instruction that describes a task. Write a
response that appropriately completes the request.

### Instruction:
Select from the list below figures from The X-Files
that are losing their hair or bald: Walter Skinner,
John Fitzgerald Byers, Dana Scully, Melvin Frohike,
Darius Michaud, Peter Watts, Conrad Strughold, Queequeg

### Response:
```

**Perturbation 2: Instruction Reordering:**

```
Below is an instruction that describes a task. Write a
response that appropriately completes the request.

### Instruction:
From The X-Files, identify characters who are bald or balding
from the following list: Walter Skinner, John Fitzgerald
Byers, Dana Scully, Melvin Frohike, Darius Michaud,
Peter Watts, Conrad Strughold, Queequeg

### Response:
```

## L.2. Selection of Input Variables

For this experiment, we employed the same input variable selection strategy as detailed in Appendix K.2. This approach combines Selective Input Variable Analysis and Phrase-level Aggregation to manage the computational complexity associated with longer, open-ended instructions while preserving the core semantic elements for interaction analysis.

## L.3. Experimental Results

The experimental outcomes from this more challenging setup verify the main conclusions of our paper. Despite the increased complexity of the perturbations, we consistently observed that the four identified factors (supervised fine-tuning, increased

model scale, dense architectures, and few-shot learning) reduce prompt sensitivity, primarily by stabilizing low-order interactions. This replication demonstrates that our conclusions are not confined to simple template variations but hold true in scenarios that more closely mirror authentic user interactions, thereby strengthening the generalizability of our work.

Here are the results and $\tau$ is set to 0.1, we can observe the same conclusions in this experiment.

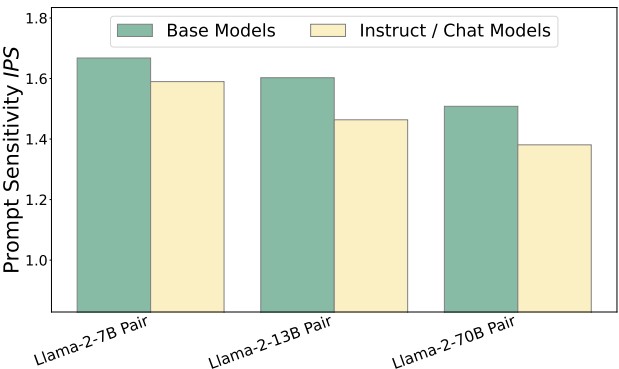

*Figure 52.* A comparison of the prompt sensitivity between instruct/chat models and base models. Results show that instruct/chat models are less sensitive than corresponding base models.

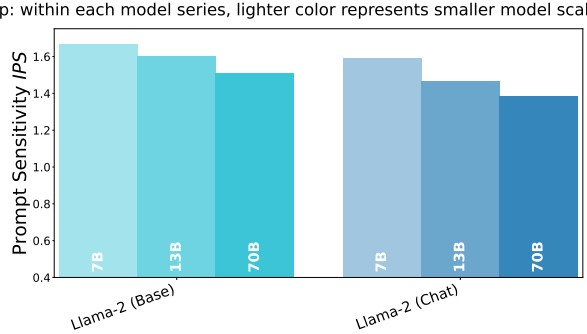

*Figure 53.* A comparison of prompt sensitivity across different model scales. As the model scale increases, the prompt sensitivity within a model series systematically decreases.

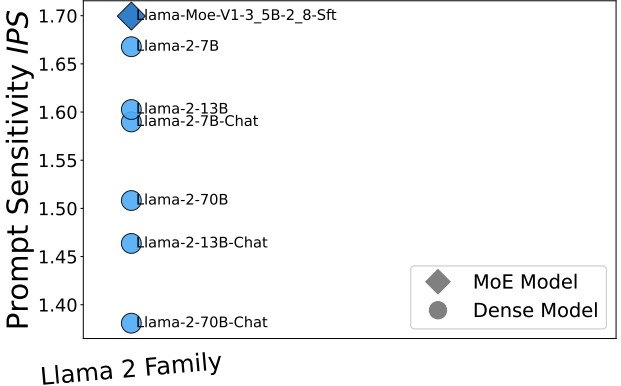

*Figure 54.* A Comparison of prompt sensitivity between MoE models and dense models. Generally, MoE models tend to be more sensitive than dense models in the same model family.

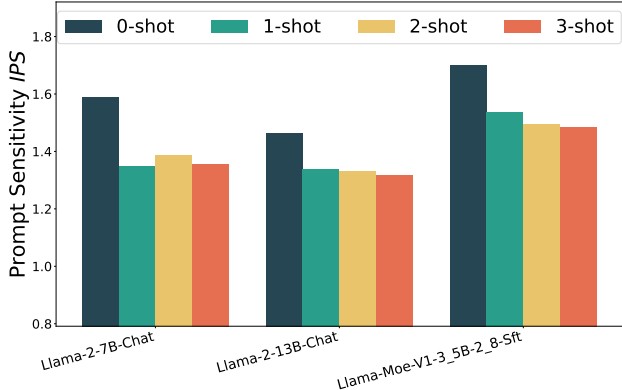

*Figure 55.* A comparison of prompt sensitivity between 0-shot learning and few-shot learning. The drop in prompt sensitivity is substantial from 0-shot to 1-shot.

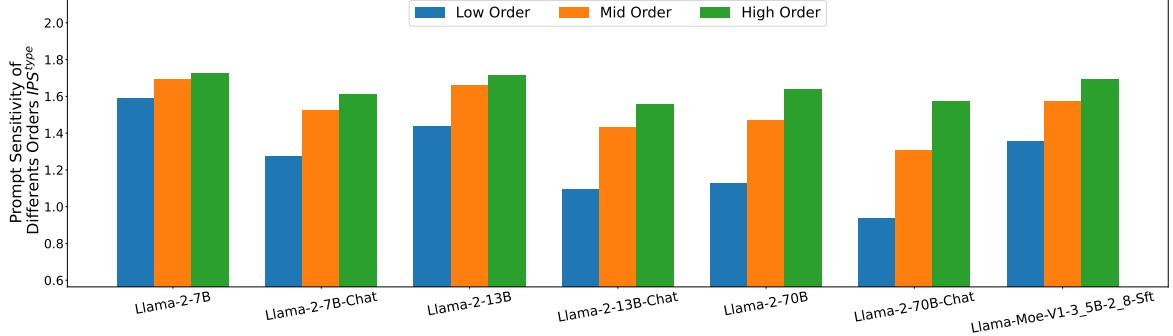

*Figure 56.* A comparison of the prompt sensitivity of three order types. Results show that low-order interactions are the least sensitive, while high-order interactions are the most sensitive.

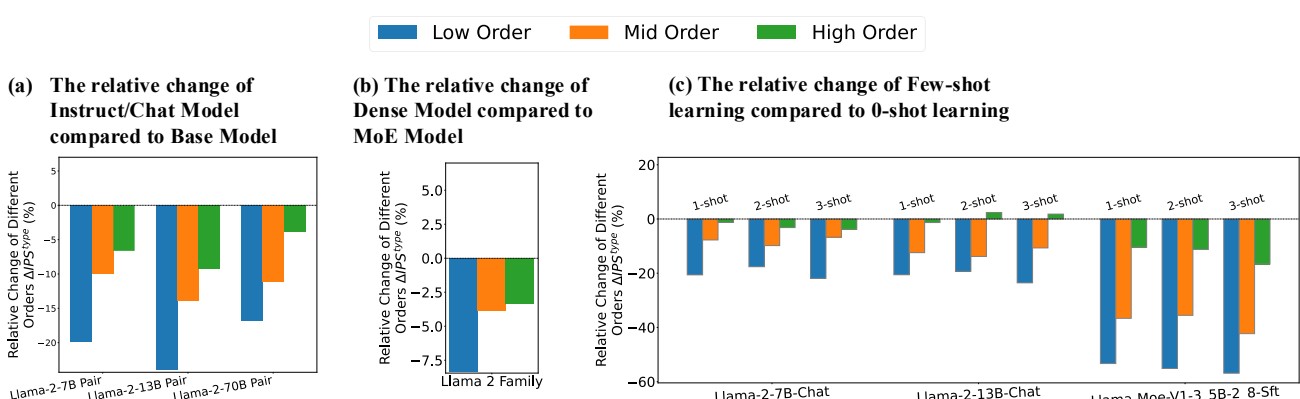

*Figure 57.* Comparing the relative change in the prompt sensitivity of low-, mid-, and high-order interactions for different factors.

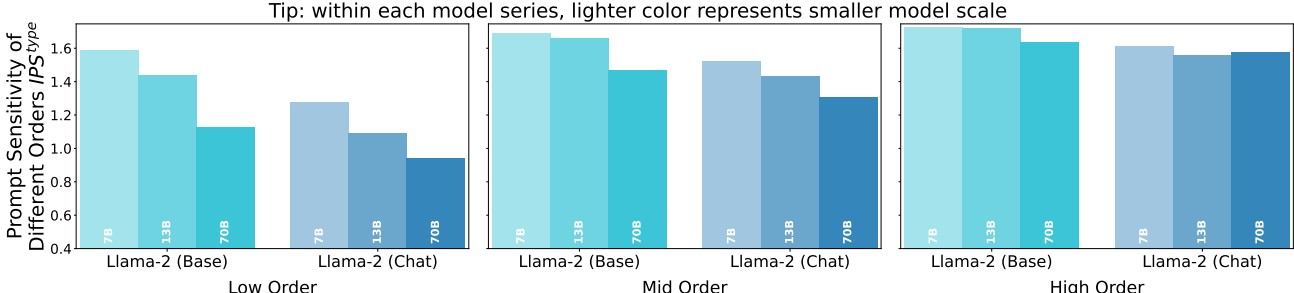

*Figure 58.* A comparison of prompt sensitivity at the order-level across different model scales.

