# OpenReview forum: "Evaluating and Explaining Prompt Sensitivity of LLMs Using Interactions"
_ICML.cc/2026/Conference — ICML 2026 regular_

### Official Review · Reviewer_2y3c · 2026-03-02

**Soundness:** 3
**Presentation:** 3
**Significance:** 2
**Originality:** 2
**Overall Recommendation:** 4
**Confidence:** 3

**Summary:**

This paper leverages a game-theoretical interaction framework to propose a fine-grained metric, IPS, for evaluating the prompt sensitivity of LLMs. The authors apply IPS to a broad set of 50 open-source LLMs and further summarize several potential factors that contribute to prompt sensitivity based on their experimental findings.

**Compliance With Llm Reviewing Policy:**

Affirmed.

**Final Justification:**

My main concerns have been resolved by the rebuttle.

**Key Questions For Authors:**

1. Could the authors elaborate further on the open-ended experiments? In particular, how does the method handle the exponentially large output space in datasets such as Databricks Dolly-15k?

2. Could the authors provide more details on the inference cost of IPS compared with other prompt sensitivity metrics, for example in terms of the number of forward passes required?

**Limitations:**

yes

**Strengths And Weaknesses:**

**Strengths**

1. The proposed method is theoretically grounded rather than relying on a heuristic metric.

2. The experiments cover multiple model families, which makes the empirical evaluation relatively comprehensive.


**Weaknesses**

1. The novelty appears limited, as the main contribution is applying an existing interaction framework to LLM prompt sensitivity analysis.

2. This metric appears to have limited practical usefulness. The current formulation relies on a scalar score defined by the log-odds of a single ground-truth token against all other tokens. In realistic open-ended settings, however, correct responses are often not unique, and multiple semantically equivalent answers may all be valid. For example, for the question “What is the capital of France?”, both “Paris” and “The place is Paris” should be considered correct. Because the method is tied to a single reference token rather than semantic equivalence at the response level, its applicability to real open-ended generation settings seems quite limited.

3. The practical value of the proposed metric remains unclear. The paper mainly introduces IPS, evaluates a range of model families on this metric, and reports several empirical trends. However, it does not sufficiently demonstrate why practitioners should care about IPS in real applications, or how this metric could meaningfully inform the deployment, prompting, or evaluation of LLMs in practice. As a result, the connection between the metric and real-world LLM usage is not yet convincing.

---

> ### Author Rebuttal · Authors · 2026-03-30
>
> We sincerely appreciate your efforts in providing constructive suggestions. We will try our best to answer all your questions.
>
> ---
>
> **Q1: Details on open-ended experiments**
>
> **A:** Thank you for your question. In open-ended experiments, we apply **selective input variable analysis and phrase-level aggregation** to limit the number of input variables $n$ (typically $n \le 15$), controlling the computational cost while preserving core semantic interactions.
>
> (1) **Selective Input Variable Analysis**. We analyze only a subset of informative input variables while treating uninformative ones (e.g., stop words) as fixed background context. Previous research [1] demonstrates that this selection does not significantly impair the faithfulness of the interaction framework.
>
> (2) **Phrase-level Aggregation**. Instead of analyzing individual words, we merge related words into combined phrasal units. This reduces the total number of input variables while preserving key semantic meaning.
>
> For more details, please see examples in Appendix K.2.
>
> [1] Defining and Extracting generalizable interaction primitives from DNNs, ICLR 2024
>
> ---
>
> **W2: Concern about practical usefulness**
>
> **A:** A good question. Actually we have addressed this issue by **defining a sequence-level reward function**. For text generation tasks (Dolly-15k), given the ground truth sequence $y^\ast = (y^\ast_1, \ldots, y^\ast_L)$, we define $\bar{p}$ as the geometric mean of the probabilities of each ground truth token in $y^\ast$, *i.e.*, $\bar{p} = \left( \prod_{l=1}^{L} p(y_l^\ast \vert x, y_{<l}^\ast) \right)^{1/L}$. The scalar output $v(x)$ is defined as $v(x) = \log \frac{\bar{p}}{1 - \bar{p}} \in \mathbb{R}$. By directly measuring the model's internal confidence in the exact target $y^\ast$, IPS bypasses the confusion when multiple prompts produce semantically equivalent answers. We will include these technical details in the revised paper.
>
> ---
>
> **W3: Practical value of IPS**
>
> **A:** Thank you for your insightful view. In practice, IPS has more practical value than output-level metrics. Most notably, IPS has **strong predictive power: a high IPS indicates unstable internal interactions**, effectively predicting **potential failure on unseen prompt variations or OOD data**. We demonstrate this through two aspects:
>
> **1. Correlation with Harder Samples and OOD dataset:** We calculate the **IPS** of 50 models on the ARC dataset, and then evaluate their **accuracy** on the **OOD** dataset MMLU-Pro. The results show a Pearson correlation coefficient of **-0.799** and a Spearman coefficient of **-0.816**. This indicates that **a lower IPS strongly correlates with better performance on harder OOD datasets**.
>
> **2. Predicting Failure on Harder Variations:** We select samples that models answer ***correctly*** across all current prompt templates. We divide them into high-IPS (highest 30%) and low-IPS (lowest 30%) groups. When **evaluated on 5 new, more complex templates**, the low-IPS group maintains higher accuracy than the high-IPS group (Table 1). This directly confirms that **a higher IPS predicts worse performance on harder prompt variations.**
>
> **Table 1: Average Accuracy on New Templates Based on IPS Group**
>
> | **Model Name**           | **Acc. of Low-IPS Group** | **Acc. of High-IPS Groups** |
> | ------------------------ | ------------------------- | --------------------------- |
> | Qwen3-8B-Instruct        | **0.67**                  | 0.55                        |
> | Qwen2.5-7B-Instruct      | **0.56**                  | 0.48                        |
> | Llama-3-8B-Instruct      | **0.61**                  | 0.50                        |
> | Llama-2-7B-Chat          | **0.35**                  | 0.27                        |
> | Mistral-7B-v0.3-Instruct | **0.67**                  | 0.61                        |
> | Olmo-2-7B-Instruct       | **0.62**                  | 0.54                        |
>
> ---
>
> **W1: Regarding originality**
>
> **A:** Thank you. Although the interaction framework exists, our core originality lies in discovering the **underlying mechanism** behind LLM prompt sensitivity for the first time: we discover that the four influencing factors (e.g., model scale, SFT)  **primarily stabilize low-order interactions**. This advances the field's understanding from **"what works" to "how it works"**, which can provide guidance for reducing sensitivity of LLMs.
>
> ---
>
> **Q2: On inference cost**
>
> **A:** Thank you. Regarding inference cost, calculating IPS requires $O(2^n)$ forward passes per sample. However, **techniques specifically designed for efficiently computing sparse interactions to bypass exhaustive $O(2^n)$ evaluations**, such as [2, 3], are applicable and **can largely reduce the inference cost**. We will discuss this extension in the revised paper.
>
> [2] SPEX: Scaling Feature Interaction Explanations for LLMs, ICML 2024
>
> [3] ProxySPEX: Inference-Efficient Interpretability via Sparse Feature Interactions in LLMs, NeurIPS 2025

---

> > ### Author Rebuttal · Reviewer_2y3c · 2026-04-01
> >
> > Thank you for your response, my questions have been resolved.

---

> > > ### Author Response · Authors · 2026-04-02
> > >
> > > We sincerely thank you for the valuable comment and for raising your score! We are delighted that our rebuttal successfully resolved your questions.
> > >
> > > Best regards,
> > > Authors

---

### Official Review · Reviewer_A7ms · 2026-03-05

**Soundness:** 3
**Presentation:** 3
**Significance:** 3
**Originality:** 2
**Overall Recommendation:** 4
**Confidence:** 3

**Summary:**

The paper focuses on the well-known problem - prompt sensitivity in LLMs. The authors argue that existing metrics are coarse-grained because they only measure final output changes and consistency without explaining why it occurs under the hood. To address the issue, the paper proposed analyzing prompt sensitivity using Interaction Effects among input variables. Based on the work on game-theoretic interaction decomposition, the authors represent the model's output as a sum of interaction terms among subsets of input variables. A new metric is created called IPS (Interaction-based Prompt Sensitivity), which measures the change of interaction effects when similar prompt are used.

The paper evaluate IPS on~50 open source LLMs, covering 6 model families. Exps are conducted on MCQ benchmarks such as ARC/MMLU. The study reports main findings as follows:
- Instruction-tuned/chat models exhibit lower prompt sensitivity than base ones.
- Larger models tend to be less sensitive to input perturbations.
- Dense model are more stable than MoEs.
- Few-shot prompting reduces prompt sensitivity compared to zero-shot prompting. (first shot example is the key.)

**Compliance With Llm Reviewing Policy:**

Affirmed.

**Final Justification:**

The rebuttal addressed most of my concerns. I maintain the score.

**Key Questions For Authors:**

- The open-ended task experiments (Appendices K, L) should be promoted to the main paper, even in condensed form, as they address one of my critical concern about task generality!
- In Fig. 1, the notation v(B|x, T) is used but Section 3.1 defines v(x) without conditioning on T.

- Q1: Can you provide evidence that IPS’s information beyond output consistency is actionable? Does higher IPS predict failure on unseen prompt perturbations?
- Q2: Have you experimented with alternative order boundaries (e.g., absolute thresholds |S| ≤ 3 instead of relative floor(n/3))?

**Limitations:**

- W1:Potential biases from the mask-token strategy are not discussed. LLMs are using different special tokens, and the OOD nature of masked inputs could systematically affect interaction estimates in model-specific ways, confounding cross-model comparisons.

- W2: Generalizability to closed-source or very large models (GPT series, Claude, Gemini) is unaddressed. The practical value for the most commonly deployed production models is unclear.

- W3: The computational cost of interaction analysis may limit scalability for longer prompts or larger models.

**Strengths And Weaknesses:**

Strengths:
- Comprehensive empirical scope. The experiments covers 50 open-source LLMs spanning 6 model families. Such breadth is commendable and lends statistical weight to claimed trends. Model size spans from 0.5B to 72B with mix of base/instruct variant, and dense vs MoE architectures, enabling a multi factorial analysis for prompt sensitivity.
- Rigorous math grounding. The interaction framework is well-founded in game theory. The theorem 3.1 provides a formal guarantee that the interaction decomposition recovers the LLM's output for masked inputs. Besides, the sparsity property, proven in prior work and verified in Figure 2, adds the practical legitimacy.
- The interesting observation: model improvements mainly stabilize low-order interactions rather than high-order interactions is an interesting and refreshing hypothesis which could motivate future research on model robustness.
- Thorough sensitivity analyses. 16 threshold values with near-perfect rank correlations, replication on MMLU, extension to open-ended tasks, and perturbations in different designs.
---

Weaknesses:
- Clarify the interpretation of interaction analysis. The paper should clearly distinguish between post-hoc attribution as a diagnostic tool and claims about internal reasoning mechanisms. interaction decomposition is a mathematical reparameterization of the output function. It decomposes what changes but how it lead to WHY change.
- Unsurprising findings. First two points listed in summary section of  paper findings are consistent with prior understanding and may not represent fundamentally new insights.
- Task diversity is limited. The eval is conducted all on MCQ benchmarks, which may not reflect the behavior of llms in open-ended generation tasks.
- The interaction framework requires evaluating the LLM on 2^n masked inputs for n input variables. The paper restricts analysis to MCQ tasks with ~13 words (yielding 2^14 AND-OR interactions). For real-world inputs of hundreds or thousands of tokens, the method is computationally intractable. The open-ended experiments rely on aggressive word selection and phrase merging, introducing subjective choices. The authors acknowledge the Fast Möbius Transform is inapplicable, and no concrete scalability path is offered. This fundamentally limits practical utility.

---

> ### Author Rebuttal · Authors · 2026-03-30
>
> Thank you for your constructive comments. We will try our best to answer all your questions.
>
> ---
>
> **W1: The interpretation of interaction analysis**
>
> **A**: Thank you for your insightful view. We agree that our framework serves as a rigorous diagnostic tool rather than discovering traditional internal reasoning mechanisms. However, unlike tracing causal pathways (e.g., specific attention heads or neurons), our approach aligns with Shapley-value-based explanations to offer conceptual interpretability. As [1] demonstrates, interactions represent meaningful semantic concepts encoded inside models. We will introduce more about the explanatory power of interactions in the paper.
>
> [1] Does a Neural Network Really Encode Symbolic Concepts? ICML 2023
>
> ------
>
> **Q1: Predictive capability of IPS**
>
> **A:** A good question. Beyond output consistency, **a high IPS predicts potential failure on unseen prompt variations or OOD data**. We demonstrate this in two ways:
>
> **1. Correlation with Accuracy:** We calculate the **IPS** of 50 models on the ARC dataset and evaluate their **accuracy** on the **OOD** dataset MMLU-Pro. The strong negative correlation (Pearson: -0.799, Spearman: -0.816) indicates that **a lower IPS strongly correlates with better OOD performance**.
>
> **2. Predicting Failure on Unseen Perturbations:** We divide **correctly** answered samples into high-IPS (top 30%) and low-IPS (bottom 30%) groups. When **evaluated on 5 new, complex templates**, the low-IPS group maintains higher accuracy than the high-IPS group (Table 1). This confirms that **a higher IPS predicts worse performance on unseen prompt perturbations.**
>
> **Table 1: Average Accuracy on New Templates**
>
> | **Model Name**           | **Acc. of Low-IPS Group** | **Acc. of High-IPS Groups** |
> | ------------------------ | ------------------------- | --------------------------- |
> | Qwen3-8B-Instruct        | **0.67**                  | 0.55                        |
> | Qwen2.5-7B-Instruct      | **0.56**                  | 0.48                        |
> | Llama-3-8B-Instruct      | **0.61**                  | 0.50                        |
> | Llama-2-7B-Chat          | **0.35**                  | 0.27                        |
> | Mistral-7B-v0.3-Instruct | **0.67**                  | 0.61                        |
> | Olmo-2-7B-Instruct       | **0.62**                  | 0.54                        |
>
> ------
>
> **Lim1: Potential Bias of Mask-Tokens**
>
> **A:** Following your suggestions, we recalculate IPS using new mask tokens  `<eos>` instead of `<unk>` or `<pad>`. Table 2 shows that the rank of new IPS scores across models **remains highly consistent with the original ranking**, confirming that our method is **robust to the choice of mask tokens**.
>
> **Table 2: IPS Values with Different Mask Tokens**
>
> | **Model Name**           | **Original IPS (Rank)** | **New IPS (Rank)** |
> | ------------------------ | ----------------------- | ------------------ |
> | Llama-2-7B-Chat          | 1.691 (1)               | 1.776 (1)          |
> | Mistral-7B-v0.3-Instruct | 1.631 (2)               | 1.762 (2)          |
> | Olmo-2-7B-Instruct       | 1.629 (3)               | 1.749 (3)          |
> | Qwen2.5-7B-Instruct      | 1.564 (4)               | 1.717 (4)          |
> | Qwen3-8B-Instruct        | 1.499 (5)               | 1.670 (5)          |
> | Llama-3-8B-Instruct      | 1.472 (6)               | 1.542 (6)          |
>
> ------
>
> **W3 & Q3: Open-ended tasks**
>
> **A:** We understand your concern. Due to space constraints, we place the experimental results of open-ended tasks in Appendices K and L. Adopting your suggestion, we will move them to the main text.
>
> ------
>
> **W4 & Lim3: Computational Cost**
>
> **A:** Thank you. For extremely long texts, **techniques specifically designed for efficiently computing sparse interactions to bypass exhaustive $O(2^n)$ evaluations**, such as [2, 3], are applicable and **can compute interactions at the scale of 1000 input variables**. We will include these methods in the revised paper.
>
> [2] SPEX: Scaling Feature Interaction Explanations for LLMs, ICML 2024
>
> [3] ProxySPEX: Inference-Efficient Interpretability via Sparse Feature Interactions in LLMs, NeurIPS 2025
>
> ------
>
> **Q2: The choice of thresholds**
>
> **A:** Following your suggestion, we conduct experiments using **fixed thresholds** (low-order $|S| \le 3$, mid-order $4 \le |S| \le 7$, high-order $|S| > 7$) while controlling total variables $n \in [10, 15]$. **Our core conclusions remain the same.** However, when $n$ is larger or smaller, these fixed thresholds become unreasonable. This justifies our reason for using relative thresholds.
>
> ------
>
> **Lim2: Generalizability to closed-source models**
>
> **A:** Thank you. This method cannot analyze closed-source LLMs. We will add this point to the limitations section.
>
> ------
>
> **Q4: Notation Issues**
>
> **A:** Thank you for your correction. We will correct the notation in Section 3.1 to ensure consistency with Figure 1.

---

> > ### Author Rebuttal · Reviewer_A7ms · 2026-04-03
> >
> > Thank you for your response, my questions have been resolved.

---

> > > ### Author Response · Authors · 2026-04-03
> > >
> > > We sincerely thank you for your time and the constructive comment!
> > > We are glad our rebuttal effectively addressed your questions.
> > >
> > > Best regards, Authors

---

### Official Review · Reviewer_SHTA · 2026-03-07

**Soundness:** 2
**Presentation:** 3
**Significance:** 2
**Originality:** 2
**Overall Recommendation:** 4
**Confidence:** 3

**Summary:**

The paper examines prompt sensitivity in large language models (LLMs), where minor variations in prompts can lead to different model predictions. Rather than relying on output-level metrics, the authors propose a fine-grained interaction-based framework to analyze this phenomenon. In this framework, the output score of an LLM is decomposed into a set of interaction effects among input variables, based on an interaction-based logical model (Sec. 3.1, Eq. (1)–(2)). Building on this formulation, the authors introduce Interaction-based Prompt Sensitivity (IPS), a metric designed to measure the instability of interaction patterns across semantically equivalent prompts (Sec. 4.1, Eq. (3)).

The approach is evaluated on 50 open-source LLMs from multiple model families using the ARC and MMLU datasets (Sec. 4.1, Fig. 4). The empirical analysis identifies four factors that help reduce prompt sensitivity: supervised fine-tuning, larger model scale, dense architectures compared with mixture-of-experts (MoE), and few-shot prompting (Sec. 4.2, Fig. 5–8). Furthermore, the authors argue that these factors primarily stabilize low-order interactions, while having a more limited effect on high-order interactions (Sec. 4.3, Fig. 9–11).

claim 1: Previous studies typically evaluate prompt sensitivity by comparing the LLM’s final outputs when prompts change. However, such coarse-grained metrics fail to explain the internal reasons for prompt sensitivity.

claim 2: Discover that subtle changes to prompts can trigger severe instability in interactions, even when the outputs of the LLM remain the same.

claim 3: Propose a fine-grained metric to evaluate the prompt sensitivity of LLMs, which is termed Interaction-based Prompt Sensitivity (IPS).

**Compliance With Llm Reviewing Policy:**

Affirmed.

**Final Justification:**

Questions are addressed. And no major issues

**Key Questions For Authors:**

1. How does the method scale to long prompts with dozens or hundreds of tokens?
2. How does the IPS metric behave when multiple prompts produce semantically different but valid answers?
3. Are the conclusions about interaction orders consistent across other NLP tasks beyond QA?
4. How stable are IPS measurements when evaluated with different prompt perturbation strategies?

Other questions can be found in the Strengths And Weaknesses

If the authors can address some of these weaknesses or questions, I will revise my evaluation accordingly.

**Limitations:**

Found in weaknesses.

**Strengths And Weaknesses:**

## Soundness
### Strengths
- The paper provides a formal definition of interactions and uses an interaction-based logical model to decompose the LLM output score. The universal matching property (Theorem 3.1) states that the logical model can reproduce the DNN output for masked inputs.
- The empirical evaluation covers 50 open-source LLMs across several model families, including Llama, Qwen, Mistral, and OLMo. Experiments use established benchmarks such as ARC and MMLU (Sec. 4.1). The IPS metric is evaluated under multiple prompt templates and averaged over dataset samples.
- Additional analyses examine factors such as instruction tuning, model scale, architecture type, and few-shot prompting.

### Weaknesses
- 1. Experiments focus on multiple-choice QA tasks (Sec. 4.1), potentially limiting generality.

## Presentation
### Strengths
- The motivation for studying prompt sensitivity is clearly introduced in the introduction.
- The distinction between coarse-grained output metrics and fine-grained interaction analysis is explained clearly.
- The experimental section systematically analyzes different contributing factors.
- The paper structure is easy to follow.

### Weaknesses
- 1. A limitations or discussion part is missing, which should discuss the constraints and weaknesses of the proposed IPS and provide guidance for its use in future research.
- 2. There are concerns regarding the identification of low-, mid-, and high-order samples. The paper seems to partition them using the 1/3n and 2/3n thresholds (Section 4.3). Is there a theoretical or empirical justification for this choice?

## Significance
### Strengths
- The paper addresses an important and timely problem in large language models: prompt sensitivity and robustness. Prompt sensitivity can cause large performance fluctuations from small prompt changes. This issue directly affects the reliability of LLM-based systems.
- The work proposes a fine-grained framework that examines the internal reasoning patterns of models rather than only final outputs. This perspective contributes to interpretability and robustness analysis of LLMs.
- The paper identifies several factors associated with reduced prompt sensitivity: 1. supervised fine-tuning; 2. larger model scale; 3. dense architectures; 4. few-shot prompting.

### weakness:
- Could the authors clarify the meaning of I() and explain how IPS can be used to improve prompt design? Although the authors propose a fine-grained metric (IPS) to evaluate the prompt sensitivity of LLMs, it is still unclear how practitioners can use this metric to design better prompts in practice.


## Originality
### Strengths
- The paper introduces Interaction-based Prompt Sensitivity (IPS) as a new metric for analyzing prompt sensitivity (Sec. 4.1; Eq. (3)). Unlike prior work focusing on output differences, IPS measures instability of internal interaction patterns.

### weakness:
- It is unsurprising that the authors identify four factors that reduce the prompt sensitivity of LLMs. These factors have been proposed in some previous work [1,2,3,4].

[1] Wei, J., Bosma, M., Zhao, V. Y., Guu, K., Yu, A. W., Lester, B., ... & Le, Q. V. (2021). Finetuned language models are zero-shot learners. arXiv preprint arXiv:2109.01652.

[2] Fedus, W., Zoph, B., & Shazeer, N. (2022). Switch transformers: Scaling to trillion parameter models with simple and efficient sparsity. Journal of Machine Learning Research, 23(120), 1-39.

[3] Chung, H. W., Hou, L., Longpre, S., Zoph, B., Tay, Y., Fedus, W., ... & Wei, J. (2024). Scaling instruction-finetuned language models. Journal of Machine Learning Research, 25(70), 1-53.

[4] Min, S., Lyu, X., Holtzman, A., Artetxe, M., Lewis, M., Hajishirzi, H., & Zettlemoyer, L. (2022, December). Rethinking the role of demonstrations: What makes in-context learning work?. In Proceedings of the 2022 conference on empirical methods in natural language processing (pp. 11048-11064).

---

> ### Author Rebuttal · Authors · 2026-03-30
>
> We appreciate your constructive suggestions and address your questions below.
>
> ---
>
> **W1 & Q3: Generalizability beyond multiple-choice QA tasks**
>
> **A**: We understand your concern. Due to space limits, results of open-ended tasks are placed in Appendices K and L. **We have already conducted open-ended experiments on Dolly-15k** (Lines 396-406 in the main text). Results (Appendices K.3 and L.3) show that **conclusions about interaction orders are the same as MCQ experiments**. We will move them to the main text.
>
> ---
>
> **Q2: Regarding semantically different but valid answers**
>
> **A:** A good question. In open-ended tasks, it's very possible that multiple prompts produce semantically different but valid answers. Actually we have addressed this issue in Dolly-15k by **defining a sequence-level reward function**. Given the ground truth sequence $y^\ast = (y^\ast_1, \ldots, y^\ast_L)$, we define $\bar{p}$ as the geometric mean of each token in $y^\ast$, i.e., $\bar{p} = \left( \prod_{l=1}^{L} p(y_l^\ast \vert x, y_{<l}^\ast) \right)^{1/L}$. The scalar output $v(x)$ is defined as $v(x) = \log \frac{\bar{p}}{1 - \bar{p}} \in \mathbb{R}$. By directly measuring the model's internal confidence in the exact target $y^\ast$, IPS bypasses the problem of generating semantically different but valid answers.
>
> ------
>
> **Q4: Regarding diverse perturbation strategies**
>
> **A:**  Following your suggestion, we design 5 new, complex prompt templates and recalculate new IPS scores. Table 1 shows that the rank of new IPS scores across models **remains highly consistent with the original ranking**. This demonstrates IPS's **robustness** to diverse perturbations.
>
> **Table 1: Comparison of IPS between original and new templates**
>
> | **Model Name**           | **Original IPS (Rank)** | **New IPS (Rank)** |
> | ------------------------ | ----------------------- | ------------------ |
> | Llama-2-7B-Chat          | 1.691 (1)               | 1.801 (1)          |
> | InternLM2-Chat-7B        | 1.650 (2)               | 1.782 (2)          |
> | Mistral-7B-v0.3-Instruct | 1.631 (3)               | 1.768 (3)          |
> | Qwen2.5-7B-Instruct      | 1.564 (4)               | 1.701 (4)          |
> | Qwen3-8B-Instruct        | 1.499 (5)               | 1.653 (5)          |
> | Llama-3-8B-Instruct      | 1.472 (6)               | 1.560 (6)          |
>
> ------
>
> **Q1: Regarding long texts**
>
> **A:** Thank you for your question. For moderate-length inputs (e.g., $\le 200$ tokens), we have successfully applied **selective input variable analysis and phrase-level aggregation** on Dolly-15k (Appendix K.2).
>
> For extremely long texts, **techniques specifically designed for efficiently computing sparse interactions to bypass exhaustive $O(2^n)$ evaluations**, such as [1, 2], are applicable and **can compute interactions at the scale of 1000 input variables**. We will discuss this extension in the revised paper.
>
> [1] SPEX: Scaling Feature Interaction Explanations for LLMs, ICML 2024
>
> [2] ProxySPEX: Inference-Efficient Interpretability via Sparse Feature Interactions in LLMs, NeurIPS 2025
>
> ------
>
> **W4 & W5: Theoretical explanations and originality**
>
> **A:** Thank you. $I_S$ measures the **joint effect of specific input variables on the model's output**. Consider the input sentence $x =$ "1 plus 2 equals", where the LLM outputs "3". $I_{\lbrace \text{1, 2} \rbrace}$ represents the effect of the set $S = \lbrace \text{1, 2} \rbrace$ on generating the output "3".
>
> **Actionable Guidance for Prompt Design**: We can use IPS to select the most stable template from a candidate pool. By calculating a template's average IPS against other templates, the template with the lowest average IPS represents the most robust one.
>
> Previous studies proposed the four factors, but we are the first to analyze the impact of these factors in the **field of prompt sensitivity**. More crucially, we reveal the **underlying mechanisms** of *how* they work: they primarily **stabilize low-order interactions**.
>
> ------
>
> **W3: Regarding the threshold selection**
>
> **A:** A good question. In the field of interpreting DNNs via interactions, using relative thresholds to categorize interaction orders is a common practice. Following [3], we choose $1/3n$ and $2/3n$ to **evenly classify interaction orders for varying input lengths**.
>
> Moreover, we conduct experiments using **fixed thresholds** (low-order $|S| \le 3$, mid-order $4 \le |S| \le 7$, high-order $|S| > 7$) while controlling total variables $n \in [10, 15]$. **Our core conclusions remain the same.** However, when the total number of variables $n$ is larger or smaller, these fixed partitions become unreasonable. This justifies our reason for using relative thresholds.
>
> [3] A Unified Approach to Interpreting Self-supervised Pre-training Methods for 3D Point Clouds via Interactions, CVPR 2025
>
> ------
>
> **W2: Regarding the limitations part**
>
> **A:** Thank you for your suggestion. We will add a section to discuss the limitations of our method.

---

> > ### Author Rebuttal · Reviewer_SHTA · 2026-04-02
> >
> > Thanks for the response. I have raised my score.

---

> > > ### Author Response · Authors · 2026-04-02
> > >
> > > Thank you for your valuable insights and for raising your score! We are glad our rebuttal effectively addressed your questions.
> > >
> > > Best regards, Authors

---

### Official Review · Reviewer_EHuq · 2026-03-12

**Soundness:** 3
**Presentation:** 3
**Significance:** 2
**Originality:** 3
**Overall Recommendation:** 4
**Confidence:** 4

**Summary:**

The paper proposes using game-theoretic interactions (AND/OR interaction patterns between input variables) as a fine-grained tool to analyze prompt sensitivity in LLMs. They decompose an LLM's output score into a sum of interaction effects, classify interactions as stable vs. unstable when prompts change, and propose a new metric called Interaction-based Prompt Sensitivity (IPS). They evaluate 50 open-source LLMs across 6 families on ARC and MMLU benchmarks.

**Compliance With Llm Reviewing Policy:**

Affirmed.

**Final Justification:**

The rebuttal addressed most of my concerns.

**Key Questions For Authors:**

Several questions:
Q1: Could you provide the total GPU-hour cost of the experiments? Also how does it scale with increased input length?
Q2: How could the metric add more practical value? Could the metric providing some prediction capability? For example, the metric on easy samples could predict the failure rate of harder samples? Or if a model is internally unstable on sample X but happens to get the right answer, does that predict it will fail on a slightly harder variant of X?
Q3: Any correlation study of IPS with other existing benchmarks? If a model ranks well on IPS, would it show better performance in certain type of benchmarks, for example OOD generalization?

**Strengths And Weaknesses:**

Strength of the paper:
S1.  This paper analyzes the prompt sensitivity problem through the lens of interaction decomposition, which is new and interesting. The idea that interactions can be unstable even when final outputs are unchanged is a very interesting finding that output-level metrics miss.
S2. The experiments are thorough. Experiments cover 50 LLMs across 6 families (Llama, Mistral, Qwen, OLMo, InternLM) is quite comprehensive. The controlled comparisons across the four factors are well-designed — matching families for base vs. instruct, dense vs. MoE, etc.
S3. The paper shows strong theoretical grounding. The framework was designed with formal guarantees.
S4. The finding that all four factors converge on the same mechanism (stabilizing low-order interactions) is a very interesting result. It provides a unifying lens rather than four disconnected observations.

Weakness of the paper
W1: The practical value of fine-grained interaction-level analysis is not established. The paper's central claim is that output-level metrics are insufficient and that interaction-based analysis reveals "hidden instability" invisible to traditional metrics. However, the paper did not demonstrates why this hidden instability matters in practice.
W2. The experiments mainly focus on multi-choice questions and the extension to open-ended tasks uses semantic similarity as a proxy and feels not deeply investigated. Real world prompt sensitivity concerns are more complex and critical compared to multi-choice questions.
W3. The paper claimed that IPS provides explanation to prompt sensitivity. The experiment findings are more descriptive other than explanatory. In addition, is it not clear if the analysis is actionable and provide guidance in possible improvements. Besides, the computation cost of conducting such evaluation would be high.

---

> ### Author Rebuttal · Authors · 2026-03-30
>
> Thank you very much. We find many of your comments to be deeply insightful. We are glad to answer all your questions.
>
> ---
>
> **W1 & Q2 & Q3: Concern about the practical value**
>
> **A:** Thank you for your question. The "hidden instability" revealed by our interaction-based analysis has more practical value than output-level metrics. Most notably, IPS has **strong predictive power**: a high IPS indicates unstable internal interactions, effectively predicting **potential failure on harder prompt variations or OOD data**. We demonstrate this through two aspects:
>
> **1. Predicting Failure on Harder Variations:** We select samples that models answer ***correctly*** across all current prompt templates. We divide them into high-IPS (highest 30%) and low-IPS (lowest 30%) groups. When **evaluated on 5 new, more complex templates**, the low-IPS group maintains higher accuracy than the high-IPS group (Table 1). This directly confirms that **a higher IPS predicts worse performance on harder prompt variations.**
>
> **Table 1: Average Accuracy on New Templates Based on IPS Group**
>
> | **Model Name**           | **Acc. of Low-IPS Group** | **Acc. of High-IPS Groups** |
> | ------------------------ | ------------------------- | --------------------------- |
> | Qwen3-8B-Instruct        | **0.67**                  | 0.55                        |
> | Qwen2.5-7B-Instruct      | **0.56**                  | 0.48                        |
> | Llama-3-8B-Instruct      | **0.61**                  | 0.50                        |
> | Llama-2-7B-Chat          | **0.35**                  | 0.27                        |
> | Mistral-7B-v0.3-Instruct | **0.67**                  | 0.61                        |
> | Olmo-2-7B-Instruct       | **0.62**                  | 0.54                        |
>
> **2. Correlation with Harder Samples and OOD Generalization:** We calculate the **IPS** of 50 models on the ARC dataset, and then evaluate their **accuracy** on the **OOD** dataset MMLU-Pro, which is **harder** for models to answer. The results show a Pearson correlation coefficient of **-0.799** and a Spearman coefficient of **-0.816**. This indicates that **a lower IPS strongly correlates with better performance on harder OOD datasets**.
>
> ---
>
> **Q1: GPU-hour cost**
>
> **A:** Thank you. Evaluating all the 50 models on four Tesla V100-DGXS-32G takes approximately 16 hours. When the input length increases, we apply key-entity extraction and phrase-level aggregation (Lines 407-420)  to limit the number of variables $n$ (typically $n \le 15$). This ensures the total computational time remains manageable and does not scale infinitely with text length.
>
> ---
>
> **W2: Experiments on open-ended tasks**
>
> **A:** We understand your concern. Due to space constraints, we place the details of the open-ended tasks in Appendices K and L. In fact, **we have already conducted open-ended generation experiments on the Dolly-15k dataset** (Lines 396-406 in the main text), evaluating more complex perturbations (such as **semantic paraphrasing and instruction reordering**, examples in Appendix L.1). We will promote this part to the main text.
>
> For your concern about complex real-world scenarios, we agree that they are critical. However, our **primary objective** in this work is to demonstrate a more **counter-intuitive** phenomenon: substantial changes in interactions occur **even with minor modifications to prompt templates**. While largely altering the prompt structure would undoubtedly impact the LLM's outputs and interaction patterns, doing so makes it difficult to decouple the effects of semantic shifts from inherent prompt sensitivity. We will include this clarification in the discussion part.
>
> ---
>
> **W3: Explanation vs. Description and Actionability**
>
> **A:** A good question. Our core explanatory contribution lies in revealing *how* the four influencing factors work: **they systematically reduce the prompt sensitivity of low-order interactions.** We will revise the paper to clearly distinguish our explanatory contributions from our descriptive findings.
>
> **Actionable Guidance for Prompt Design**: We can use IPS to select the most stable template from a candidate pool. By calculating a template's average IPS against other templates, the template with the lowest average IPS represents the most robust one.
>
> As for the computational cost of the evaluation, **techniques specifically designed for efficiently computing interactions over sparse structures to bypass exhaustive $O(2^n)$ evaluations**, such as [1, 2], are directly applicable and represent a clear path for reducing inference cost. We will discuss this extension in the revised paper.
>
> [1] SPEX: Scaling Feature Interaction Explanations for LLMs, ICML 2024
>
> [2] ProxySPEX: Inference-Efficient Interpretability via Sparse Feature Interactions in LLMs, NeurIPS 2025

---

> > ### Author Rebuttal · Reviewer_EHuq · 2026-04-02
> >
> > Thanks for the response. I have raised my score.

---

> > > ### Author Response · Authors · 2026-04-02
> > >
> > > We sincerely thank you for the thoughtful comment and for raising your score! We are grateful that our rebuttal addressed your concerns.
> > >
> > > Best regards, Authors

---

### Decision · Program_Chairs · 2026-04-30

**Decision:**

Accept (regular)

**Comment:**

The paper proposes to look at the "interactions" the effects of having subsets of words together has on the output, beyond just the overall sentence effect. There are many robustness, brittleness, consistency and sensitivity papers lately, but this take is unique and well executed.
This is new, interesting and well justified with both theory and experiments. There were several questions by reviewers, which were addressed in the rebuttal and might be worth addressing for the camera ready, for example, giving some intuitions on I and technical terms,

A note about your finding, reading it through it seems like if I replace the word sensitivity with lower performance it is almost always also true, model scale being the clearest. I am surprised no correlation with the average score is discussed (I might be only thinking about it because of [this](https://arxiv.org/abs/2602.03344) recent finding, but it seems like you do have all the numbers to know already, and it would change the conclusion a lot if everything you say is predicting performance rather than specifically sensitivity). This is also relevant when you show predictability, without a baseline (it is known, for instance, that some examples are easier or harder [see here https://arxiv.org/abs/2503.01622](https://arxiv.org/abs/2503.01622) but in many others as well)